# Multi-omics analysis identifies therapeutic vulnerabilities in triple-negative breast cancer subtypes

Brian D. Lehmann [1,2,10]✉, Antonio Colaprico[3,4,10], Tiago C. Silva [3], Jianjiao Chen [3], Hanbing An[5], Yuguang Ban [3,4], Hanchen Huang[3], Lily Wang[3,4], Jamaal L. James[1], Justin M. Balko [1,2,6], Paula I. Gonzalez-Ericsson [2], Melinda E. Sanders[2,6], Bing Zhang [7,8], Jennifer A. Pietenpol[2,9] & X. Steven Chen[3,4]✉

Triple-negative breast cancer (TNBC) is a collection of biologically diverse cancers characterized by distinct transcriptional patterns, biology, and immune composition. TNBCs subtypes include two basal-like (BL1, BL2), a mesenchymal (M) and a luminal androgen receptor (LAR) subtype. Through a comprehensive analysis of mutation, copy number, transcriptomic, epigenetic, proteomic, and phospho-proteomic patterns we describe the genomic landscape of TNBC subtypes. Mesenchymal subtype tumors display high mutation loads, genomic instability, absence of immune cells, low PD-L1 expression, decreased global DNA methylation, and transcriptional repression of antigen presentation genes. We demonstrate that major histocompatibility complex I (MHC-I) is transcriptionally suppressed by H3K27me3 modifications by the polycomb repressor complex 2 (PRC2). Pharmacological inhibition of PRC2 subunits EZH2 or EED restores MHC-I expression and enhances chemotherapy efficacy in murine tumor models, providing a rationale for using PRC2 inhibitors in PD-L1 negative mesenchymal tumors. Subtype-specific differences in immune cell composition and differential genetic/pharmacological vulnerabilities suggest additional treatment strategies for TNBC.

[1] Department of Medicine, Vanderbilt University Medical Center, Nashville, TN 37232, USA. [2] Vanderbilt-Ingram Cancer Center, Vanderbilt University Medical Center, Nashville, TN 37232, USA. [3] Department of Public Health Sciences, University of Miami Miller School of Medicine, Miami, FL 33136, USA. [4] Sylvester Comprehensive Cancer Center, University of Miami Miller School of Medicine, Miami, FL 33136, USA. [5] Department of Otolaryngology, Vanderbilt University Medical Center, Nashville, TN 37232, USA. [6] Department of Pathology, Microbiology and Immunology, Vanderbilt University Medical Center, Nashville, TN 37232, USA. [7] Lester and Sue Smith Breast Center, Baylor College of Medicine, Houston, TX 77030, USA. [8] Department of Molecular and Human Genetics, Baylor College of Medicine, Houston, TX 77030, USA. [9] Department of Biochemistry, Vanderbilt University, Nashville, TN 37232, USA. [10]These authors contributed equally: Brian D. Lehmann, Antonio Colaprico. ✉email: brian.d.lehmann@vumc.org; steven.chen@med.miami.edu

Triple-negative breast cancer (TNBC) is a heterogeneous disease defined by the absence of estrogen receptor (ER) and progesterone receptor (PR) expression and human epithelial growth factor receptor 2 (HER2) amplification. The lack of expression of these receptors and high-frequency "driver" alterations has impeded the development of targeted therapies for TNBC[1]. Currently, chemotherapy and immune checkpoint blockade are the leading treatment options for the majority of patients with TNBC[2]. TNBCs display transcriptional diversity with at least four tumor-intrinsic subtypes that include two basal-like (BL1, BL2), a mesenchymal (M), and a luminal androgen receptor (LAR) subtype[3–5]. Each subtype displays unique biology and differentially responds to standard-of-care chemotherapy[6,7]. Androgen receptor (AR), PI3K/AKT, EGFR, Ras/MAPK, JAK/STAT, and the NOTCH pathways are altered in a small percentage of TNBCs and have undergone clinical investigation[8,9]. Despite advances in tumor characterization, current biological insights have yet to be translated to specific treatments, with the exception of PARP inhibitors or platinum agents in germline BRCA1/2 carriers[10]. Recently, immune checkpoint inhibitors targeting PD1/PD-L1 were demonstrated to have efficacy for TNBC and have received FDA-approval for the treatment of metastatic TNBC in combination with nab-paclitaxel[2]. However, only a fraction of patients respond to immune checkpoint inhibition and these responses are correlated to both PD-L1 expression and tumor mutational burden (TMB)[2,11,12]. TNBC subtypes are associated with different immune cell compositions; particularly striking is the absence of immune cells in the mesenchymal subtype (M-subtype)[6]. These data suggest mesenchymal TNBCs likely have developed mechanisms to escape immune surveillance.

Here, we show a comprehensive subtype-specific analysis of mutation, copy number, gene expression, DNA methylation, and proteomic data of TNBC patients in The Cancer Genome Atlas (TCGA)[13] and Clinical Proteomic Tumor Analysis Consortium (CPTAC)[14] that identifies a potential mechanism of immune escape and provides further understand the biology of TNBC subtypes. TNBC cell line and animal models identify genetic and pharmacological vulnerabilities and identify potential therapeutic strategies for TNBC patients.

## Results

**Pathology-guided multi-omic identification of TNBC and clinicopathological differences among TNBC subtypes.** TNBC patients were identified from TCGA[13], CPTAC[14], the Molecular Taxonomy of Breast Cancer International Consortium (METABRIC)[15], and from the MET500[16] using genomic data guided by the expression distribution of clinically defined ER, PR, and HER2 tumors (Supplementary Fig. 1a and "Methods"). We identified 192 TNBC patients within TCGA (17.5%), 27 in CPTAC (25.7%), 348 in METABRIC (17.6%) and 40 metastatic TNBC patients from the MET500 (43.5%) cohorts (Supplementary Data 1). TNBC tumors were subtyped using RNA expression (Supplementary Data 2). Primary TNBCs showed similar distributions as previous studies[3,6] and distinct subtype-specific patterns (Table 1). Patients with LAR-subtype tumors were diagnosed at an older age and were more frequently invasive lobular pathology (13.5% vs. < 5%, $p = 8.88e−6$). Invasive ductal carcinoma was the most common histology, however special histological subtypes were significantly (chi-squared test) enriched in individual subtypes, with medullary carcinomas in the BL1 subtype ($p = 0.0041$), malignant phyllodes tumors in the M-subtype ($p = 0.0026$), and metaplastic carcinomas in the BL2 subtype ($p = 0.011$), highlighting the diversity within these transcriptional subtypes. We next examined whether prognosis

varied by subtype. The TNBC subtypes displayed similar trends as previously published[3] with the BL1 subtype trending to have the lowest (hazard ratio, HR 0.81) and the BL2-subtype displaying the highest (HR 1.87) risk for progression (Supplementary Fig. 1b, c). The presence of immune cells has been associated with survival[17] and TNBCs with lower immune cell estimates[18] trended (log-rank $p$-value = 0.11) towards a shorter progression-free interval (PFI), while tumors with stromal immune cells displayed the lowest risk of recurrence (HR, 0.59) (Supplementary Fig. 1d, e).

**Unsupervised clustering and single-cell RNA analysis uncovers intra-tumor heterogeneity.** We and others have previously identified between four to six distinct transcriptional TNBC subtypes[3–6]. Therefore, we performed unsupervised $k$-means consensus clustering of TGCA TNBC tumors. The optimal number of clusters were determined to be five based on the area under the curve of the consensus distribution function (Supplementary Fig. 2a, b). Annotation of the consensus cluster with subtype correlation strength showed that cluster 1 was mostly M-subtype, cluster 2 consisted of tumors with mixed M- and BL1 subtype correlations, cluster 3 was mostly BL1- and immunomodulatory (IM)-subtypes, cluster 4 was primarily BL2, and cluster 5 was nearly all LAR (Supplementary Fig. 2c). Interestingly, both mesenchymal clusters (1 and 2) lacked IM-subtype calls, consistent with the absence of non-tumor immune infiltrates in bulk tumors of this subtype[6].

Since some tumors correlated to two subtypes, we selected those tumors with low consensus clustering correlation (<0.5) as potential mixed subtype tumors. Tumors with mixed subtypes displayed significantly ($p = 0.049$, log-rank) decreased overall survival stratified by TNBC subtype and consensus clustering (Supplementary Fig. 2d, e). To determine how individual cells contribute to a mixed subtype we evaluated single-cell RNA sequencing (scRNA-seq) from six primary TNBC patients (Supplementary Fig. 3a)[19]. Using lymphocyte, monocyte, myoepithelial, endothelial, and epithelial cell markers we identified tumor epithelial cells (Supplementary Fig. 3b). We performed TNBC subtyping on individual cells and aggregated expression of all cells for each tumor representing a pseudo-bulk analysis. UMAP plots showed distinct clusters of cells that varied by the four TNBC subtypes (Supplementary Fig. 3c). The subtype composition of individual tumors varied between patients, however, the subtype correlation strength in the pseudo-bulk analysis was associated with the composition of individual cell subtypes (Supplementary Fig. 3d). These data provide evidence that tumors with multiple correlations are composed of mixed subtypes and may reflect tumor cell plasticity that allows transition between cell states.

**Identification of TNBC subtype features through integrative genomic characterization analyses.** To identify additional subtype-specific characteristics and potential therapeutic strategies, we evaluated DNA mutations, copy number, gene expression, protein and phosphoprotein expression, DNA methylation, and chromatin accessibility in TNBC tumors from the TCGA stratified by subtype (Fig. 1a). Since TNBC tumors are rarely pure, but rather a mixture or part of a continuum, we performed all differential testing using subtype correlation strength rather than binary subtype assignment. The majority of TNBC samples were the basal-like by PAM50, except for HER2 subtype enrichment in LAR tumors (Fig. 1b). The M-subtype displayed lower stromal and immune cell estimates[18] suggesting an absence of immune cells within these tumors. To better understand the cellular composition of TNBC tumors, we deconvoluted bulk

**Table 1 Clinical attributes of TNBC patients from CPTAC, TCGA, METABRIC and MET500 by TNBC subtype.**

| | | | TNBC | Basal-like 1 | Basal-like 2 | Mesenchy-mal | Luminal AR |
|---|---|---|---|---|---|---|---|
| Primary | Datasets | CPTAC | 27 | 11 (40.7%) | 3 (11.1%) | 9 (33.3%) | 4 (14.8%) |
| | | TCGA | 183 | 64 (35.0%) | 37 (20.1%) | 54 (29.5%) | 28 (15.3%) |
| | | METABRIC | 348 | 124 (35.6%) | 70 (20.1%) | 71 (20.4%) | 83 (23.9%) |
| | Age | | 55.7 | 52.7 | 57.9 | 54.8 | 59.9 |
| | Stage | I | 120 (26.0%) | 36 (22.6%) | 22 (24.1%) | 30 (27.0%) | 32 (33.3%) |
| | | II | 252 (52.7%) | 101 (63.5%) | 46 (50.5%) | 59 (53.2%) | 46 (47.9%) |
| | | III | 75 (16.2%) | 20 (12.6%) | 23 (25.3%) | 20 (18.0%) | 12 (12.5%) |
| | | IV | 3 (0.7%) | 1 (0.6%) | 0 (0.0%) | 2 (1.8%) | 0 (0.0%) |
| | Histology | IDC | 476 (85.8%) | 174 (87.9%) | 89 (81.7%) | 116 (87.2%) | 97 (82.9%) |
| | | ILC | 26 (4.9%) | 4 (2.1%) | 3 (2.8%) | 4 (3.2%) | **15 (13.5%)** $p = 8.88e-6$ |
| | | Mixed | 1 (0.1%) | 1 (1.6%) | 0 (0.0%) | 0 (0.0%) | 0 (0.0%) |
| | | Mucinous | 1 (0.1%) | 1 (1.6%) | 0 (0.0%) | 0 (0.0%) | 0 (0.0%) |
| | | Medullary | 29 (5.6%) | **18 (9.6%)** $p = 0.0041$ | 7 (6.5%) | 1 (1.8%) | 3 (2.7%) |
| | | Metaplastic | 14 (2.6%) | 0 (0.0%) | **7 (13.5%)** $p = 0.011$ | 6 (5.6%) | 1 (0.8%) |
| | | Phyllodes tumor | 4 (0.8%) | 0 (0.0%) | 0 (0.0%) | **4 (3.2%)** $p = 0.0026$ | 0 (0.0%) |
| | | Neuroendocrine | 1 (0.1%) | 0 (0.0%) | 0 (0.0%) | 1 (1.8%) | 0 (0.0%) |
| | | Secretory | 1 (0.1%) | 0 (0.0%) | 1 (0.9%) | 0 (0.0%) | 0 (0.0%) |
| | | Papillary | 1 (0.1%) | 0 (0.0%) | 1 (0.9%) | 0 (0.0%) | 0 (0.0%) |
| | | Other | 2 (0.3%) | 0 (0.0%) | 1 (0.9%) | 1 (0.8%) | 0 (0.0%) |
| | | NA | 4 (0.8%) | 1 (1.6%) | 1 (0.9%) | 0 (0.0%) | 2 (1.7%) |
| Metastatic | Datasets | MET500 | 40 | 11 (27.5%) | 10 (25.0%) | 8 (20.0%) | 10 (25.0%) |

Parentheses indicate the percentage of tumors within each column of primary or metastatic disease. Chi-squared testing was performed with statistically significant comparisons indicated in bold text.
Abbreviations: *IDC* invasive ductal carcinoma, *ILC* invasive lobular carcinoma.

RNA signals[20] using scRNA-seq data derived from primary TNBC[19] and normal human breast epithelial cells[21] (Supplementary Data 2). All subtypes were predominantly composed of epithelial cells with exception of myoepithelial and stromal cell enrichment in the BL2 subtype (Fig. 1b). The M-subtype lacked both lymphocyte and monocyte signatures. Normal epithelial cell estimation further supported myoepithelial cell origin for BL2-subtype and luminal progenitor (L1.2) origins for BL1- and LAR-subtypes[21]. Hormone-responsive L2-type cells were nearly exclusive to the LAR-subtype, supporting a more differentiated androgen receptor (AR) driven cell type in LAR tumors. Similar deconvolution methods were used to determine immune cell composition and supported an absence of antigen-presenting and effector immune cell classes in the M-subtype[22]. TNBC tumors displayed distinct patterns of gene expression. Since the mesenchymal tumors were associated with decreased immune cell signatures, we evaluated known immune cell markers, immune checkpoint expression, and antigen presentation expression and observed lower expression in the mesenchymal subtype (Fig. 1b). Similar patterns of tumor profiles were observed in primary tumors in METABRIC and metastatic tumors in MET500 (Supplementary Fig. 4a–f).

Since alterations in DNA can contribute to transcriptional diversity, we examined DNA mutations and copy number alterations (CNAs) within TNBC subtypes (Supplementary Data 4). Similar to other cancer types and consistent with a prior report[23], TNBC patients with a higher TMB (>1.5mut/Mb) had significantly ($p = 0.017$, log-rank) better progression-free interval (PFI) than patients with lower mutation loads (Supplementary Fig. 5a). Stratified by subtype, the BL1 and M-subtypes had more mutations per tumor (average 2.1 mut/Mb and 2.3 mut/Mb, respectively) compared to the BL2 and LAR subtypes (1.2 and 1.8 mut/Mb, respectively) (Supplementary Fig. 5b). However, despite both BL1 and M tumors displaying higher mutational burdens, only BL1 tumors were significantly ($p = 0.021$ log-rank test) associated with better survival, suggesting differences in subtype-specific long-term outcomes following standard chemotherapy (Supplementary Fig. 5c).

Consistent with prior reports, TP53 alterations were frequent (95%mutated/CNA) and distributed among all subtypes, while activating PIK3CA and ERBB2 mutations were enriched[5,6,24], DNA repair and cell cycle alterations largely absent from the LAR subtype (Fig. 1b). There were relatively few activating MAPK

pathway mutations, however, they were significantly enriched ($p$-value $= 0.01075$, Fisher's exact test) in the BL2 subtype (Supplementary Data 4). About 19.7% of TNBC tumors had a loss-of-function mutation in a gene that regulates epigenetic modifications. Compared to other subtypes, M-subtype tumors have a significantly higher percentage of epigenetic modifier mutations (29.6% vs. 15.6%, $p = 0.040$, Fisher's exact test) and are significantly enriched in *ASXL* gene family mutations (23.4% vs. 5.1%, $p = 0.0013$, Fisher's exact test) that regulate the polycomb repressive complex 2 (PRC2)[25]. Furthermore, the M-subtype was enriched with loss-of-function mutations in members of the BAF SWI/SNF nucleosome remodeling complex (Fig. 1b), known to negatively regulate PRC2 function[26]. These observations suggest functional loss of these chromatin remodeling genes could result in increased PRC2 activity in M-subtype tumors[27].

Mutational patterns in DNA are derived from the mutational processes leading to distinct biological changes occurring during tumorigenesis. Therefore, we examined the patterns of single base substitution signatures across the TNBC subtypes[28]. Four prominent DNA mutational signatures (APOBEC cytidine deaminase, spontaneous deamination of 5-methylcytosine, defective DNA mismatch repair, and defective DNA repair by homologous recombination) were identified in TNBC (Supplementary Fig. 5d). These signatures have previously been characterized in breast cancer without subtype associations[29]; however, our data demonstrate the defective DNA repair by homologous recombination signature was distinctly higher in the BL1- and M-subtypes (Supplementary Fig. 5e).

To identify recurrent focal chromosomal CNAs associated with each subtype, we applied GISTIC2[30] to TCGA TNBC tumors (Supplementary Fig. 5f and Supplementary Data 4). Overall, M-subtype tumors displayed the greatest degree of CNAs, as evidenced by the highest fraction of the genome altered, consistent with a prior study[24] (Supplementary Fig. 5g, h). Deletions of DNA repair genes (BRCA1, BRCA2, and ATM) and members of the BAF SWI/SNF complex (SMARCAD1, ARID1A, ARID1B, KDM6A, and BAP1) were more frequently observed in M-subtype tumors (30% vs. 5%, $p = 0.0003$ and 35% vs. 10%, $p = 0.0015$, Fisher's exact test). While amplifications in PD-L1 occurred across all subtypes, deletions in beta-2-microglobulin (B2M) were more frequent (17.8% vs. 3.7%) in the M-subtype compared to other subtypes ($p = 0.0061$, Fisher's exact test), potentially decreasing antigen presentation and impacting the efficacy of immune checkpoint therapies[31].

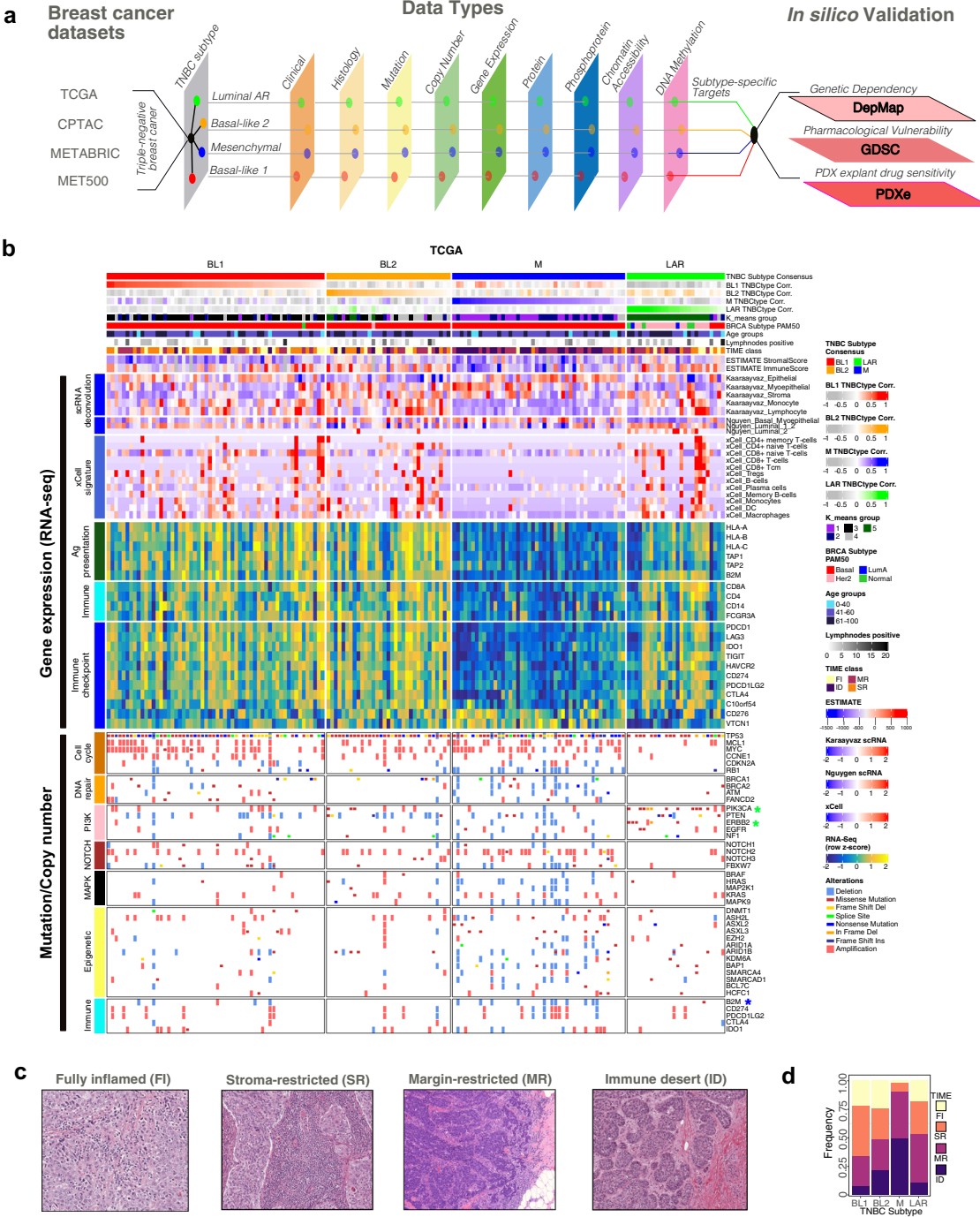

**Fig. 1 Identification of TNBC subtype features through integrative genomic analyses. a** Summary of datasets and workflow used in this study. **b** Heatmap shows TNBC samples from TGCA stratified by subtype correlation strength and annotated for k-means group, PAM50 subtype, age, positive lymph nodes, and tumor microenvironment (TIME) classification. Gene expression heatmaps show immune cell abundance (ESTIMATE), scRNA deconvolution of normal mammary cells and immune cell lineages, relative RNA expression for immune markers, and antigen presentation and immune checkpoint genes. Mutation and copy number alterations are displayed for individual tumors and stratified by pathway. * indicates significant ($p < 0.05$ two-sided Fisher's exact test, raw $p$-values in source file) differences in mutation/CNA in one subtype (colored) compared to all others. **c** Representative H&E images showing TIME classification of TCGA into fully inflamed (FI), stromal-restricted (SR), margin-restricted (MR), or immune desert (ID). These images have no scale bar because they were obtained from the TCGA Digital Slide Archive. **d** Barplot shows TIME quantification of images by TNBC subtype. See also Supplementary Fig. 1, and Supplementary Data 1, 2, and 3.

The presence of immune cells and their spatial resolution within TNBC has been shown to predict prognostic outcomes[32]. To determine if immune cells differ spatially within the tumor microenvironment, we scored archived H&E images according to previously defined criteria and binned tumors into the following

groups: (1) fully inflamed (FI) tumors with high tumor-infiltrating lymphocytes (TILs) in the stromal and carcinoma, (2) stromal restricted (SR) tumors with TILs limited to the intratumoral stroma, (3) margin restricted (MR) defined by low to absent TILs in the tumor core, but present at the interface of

the tumor margin, or (4) immune desert (ID) defined as with TILs absent from the tumor core and surrounding tissue (Supplementary Data 2). M-subtype tumors were more likely to be characterized as ID or MR (<90%), consistent with the absence of immune cell transcriptional signatures in the RNA-seq data, and together with decreased antigen presentation expression, suggest a tumor immune recognition deficiency in this subtype (Fig. 1c, d).

**Analysis of gene expression and phosphoproteomic analysis data identifies unique subtype-specific targetable pathways.** To better understand the transcriptional pathways driving TNBC subtypes, we performed pathway single-sample gene set variation analysis (GSVA) for differentially expressed genes across TNBC subtypes (Supplementary Fig. 6a and Supplementary Data 3). The BL1-subtype displayed enrichment in *MYC* targets and cell cycle checkpoint pathways. The LAR-subtype exhibited enrichment in androgen response, fatty acid metabolism, and oxidative phosphorylation. Both the BL2- and M-subtypes were enriched in EMT and TGF-β pathways. All subtypes, with the exception of the M-subtype, showed higher immune cell pathway enrichment (antigen processing and presentation, interferon-gamma response, and T cell signaling).

To determine if these subtype-specific transcriptional pathways lead to active protein signaling, we analyzed RNA and protein levels from TNBC tumors in CPTAC (Supplementary Fig. 6b, c). Overall, TNBC subtypes displayed similar protein pathway enrichment as RNA, with cell cycle enrichment in BL1 tumors, EMT and integrin pathways in BL2 and M tumors, and metabolic pathways enriched in LAR tumors (Fig. 2a). There was a consistent absence of immune cell markers and antigen presentation in M tumors (Supplementary Fig. 6c). Together these analyses highlight the diversity of underlying biology and varying immune cell states in TNBC subtypes.

To better understand protein signaling differences we examined phosphorylation levels of key residues in proteins involved in DNA repair/cell cycle, PI3K, MAPK, and antigen presentation pathways. The DNA repair and cell cycle signaling pathways displayed interesting differences among the TNBC subtypes (Fig. 2b). While not observed on RNA expression, the M-subtype displayed increased DNA repair signaling, evidenced by elevated phosphorylation of ATR (T1989), ATRIP (S224/S239), and ATM (S1981). Increased spindle checkpoint was observed in both BL1- and M-subtypes with increased protein and phosphorylation of PLK1 (T210). The BL1-subtype displayed higher CCNB2 protein and both the BL1- and M-subtypes displayed elevated phosphorylation CDK1 (T14/T15), suggesting these subtypes may benefit greater from a CDK1/2 inhibitor. In contrast, the BL2-subtype displayed higher CDK6 protein levels. In the BL2- and LAR-subtypes, phosphorylation of RB (S807/T826/S795) was higher while E2F protein levels were lower, suggesting an intact G1S checkpoint and potential sensitivity to CDK4/6 inhibition.

The PI3K/mTOR pathway was predominantly activated in the BL2- and LAR-subtype (Fig. 2c), demonstrated by increased phosphorylated AKT1(T308/S473) and AKT2 (S474). Potentially contributing to AKT activation are elevated PDK1 protein and activated ERBB2 (T1240) signaling. However, downstream AKT signaling differed between LAR- and BL2-subtypes with higher phosphorylation of GSK3B (S9/S21), AKT1S1(T266), and FOXO3 (S294) in the BL2 subtype (Fig. 2c). Both LAR and BL2 tumors also displayed downstream activation of mTOR (S2478/S2481), EIF4EBP1 (S65), EIF4B (S422), and RPS6KB1 (T421). However, the LAR-subtype may be more dependent on protein translation, as this subtype displayed the highest 40S subunit phosphorylation, RPS6 (S236), suggesting that both PI3K and

mTORC inhibitors may both be effective in targeting LAR tumors.

The EGFR/MAPK signaling pathway displayed higher activity within the BL2-subtype (Fig. 2d) as evidenced by activated MEK (pS222/S226), ERK1 (T185/Y187), and ERK2 (T202/Y204). This subtype also displayed higher KRAS protein levels and active RAF1 (S29/S220). EGFR signaling likely contributes to some of the active MAPK, evidenced by phosphorylated EGFR (T693/S1166) and adapter proteins (SHC1 and SOS1). Due to the elevated MAPK signaling, the BL2-subtype may be sensitive to EGFR, MEK or ERK targeted therapies.

The expression of antigen processing and presentation proteins were lower in the M-subtype (Fig. 2e), including immunoproteasome (PSMB8/9) components, antigen transport (TAP1/2), and MHC-I antigen presentation (HLA-A/B/C and B2M). Since immune-activated TNBC tends to respond better to chemotherapy[33,34], we hypothesized that decreased antigen presentation could provide a mechanism of immune escape for this subtype, and reactivation of this pathway could increase tumor efficacy of standard chemotherapies.

**In silico analysis of genetic and pharmacologic screens identify few targetable vulnerabilities in M-subtype TNBCs.** To identify potential subtype-specific genetic and pharmacological vulnerabilities, we identified TNBC cell line models that were screened for viability after genetic knockdown (RNAi and CRISPR) from the Broad DepMap[35], cell lines treated with 250 pharmacological inhibitors in the Genomics of Drug Sensitivity in Cancer (GDSC)[36], and patient-derived tumor xenograft (PDTX) explant models treated with 96 compounds[37] (Fig. 3a and Supplementary Fig. 7a). To account for the variation of cell line models between labs and subsequent differences in correlation to TNBC subtype (Supplementary Fig. 7b), we performed a regression analysis associating subtype correlation strength with viability after knockdown across all cell lines (Fig. 3a and Supplementary Data 6). Genes that decrease viability after knockdown or drug treatment in a subtype-dependent manner will display a more negative slope and thus a lower T-value. The RNAi and CRIPSR screens identified 3129 and 2245 genes, respectively, that differentially affected the viability of one subtype (Supplementary Fig. 7c) and identified reproducible genetic dependencies specific to each subtype (Fig. 3b and Supplementary Fig. 7d).

The LAR-subtype displayed dependencies in AR signaling (*AR, SPDEF,* and *FOXA1*), the PI3K/Akt/mTOR pathway (*PIK3CA, AKT1, EIF4A2,* and *RPS6*), and *ERBB2*, matching previously known sensitives to AR[3] and *PIK3CA*[38] (Fig. 3b). LAR-subtype cell lines were uniquely sensitive to the AR antagonist bicalutamide and the PI3K/mTOR pathway inhibitors GSK690693 (AKT), ZSTK474 (PI3K), omipalsib (PI3K/mTORC1), OSI-027 (mTORC1/2), and AZD6482 (PI3K) (Fig. 3c). LAR-subtype PDTX-derived tumor cells (PDTCs) models displayed similar preferential sensitivity to bicalutamide and several PI3K/mTOR inhibitors (Fig. 3d). The additional genetic dependencies on *CCND1* and *CDK4* provide further evidence for the effectiveness of CDK4/6 inhibitors in LAR-subtype[39] and suggest several compounds may have efficacy in LAR tumors (Supplementary Fig. 7d).

The BL1-subtype displayed genetic dependencies in the cell cycle and DNA repair genes (Fig. 3b). Cell line models were more sensitive to cell cycle inhibitors ZM447439 (AURKA/B) and PHA-793887 (CDK2/5). Cells were also sensitive to KU-559333 (ATM) and NU7441 (DNAPK), targeting the DNA repair pathway (Fig. 3c). Similarly, the PDTC models displayed more sensitivity to cell cycle inhibitors vinblastine (microtubules), MLN8237(AURKA), BI-2536 (PLK1/2), and bortezomib, in

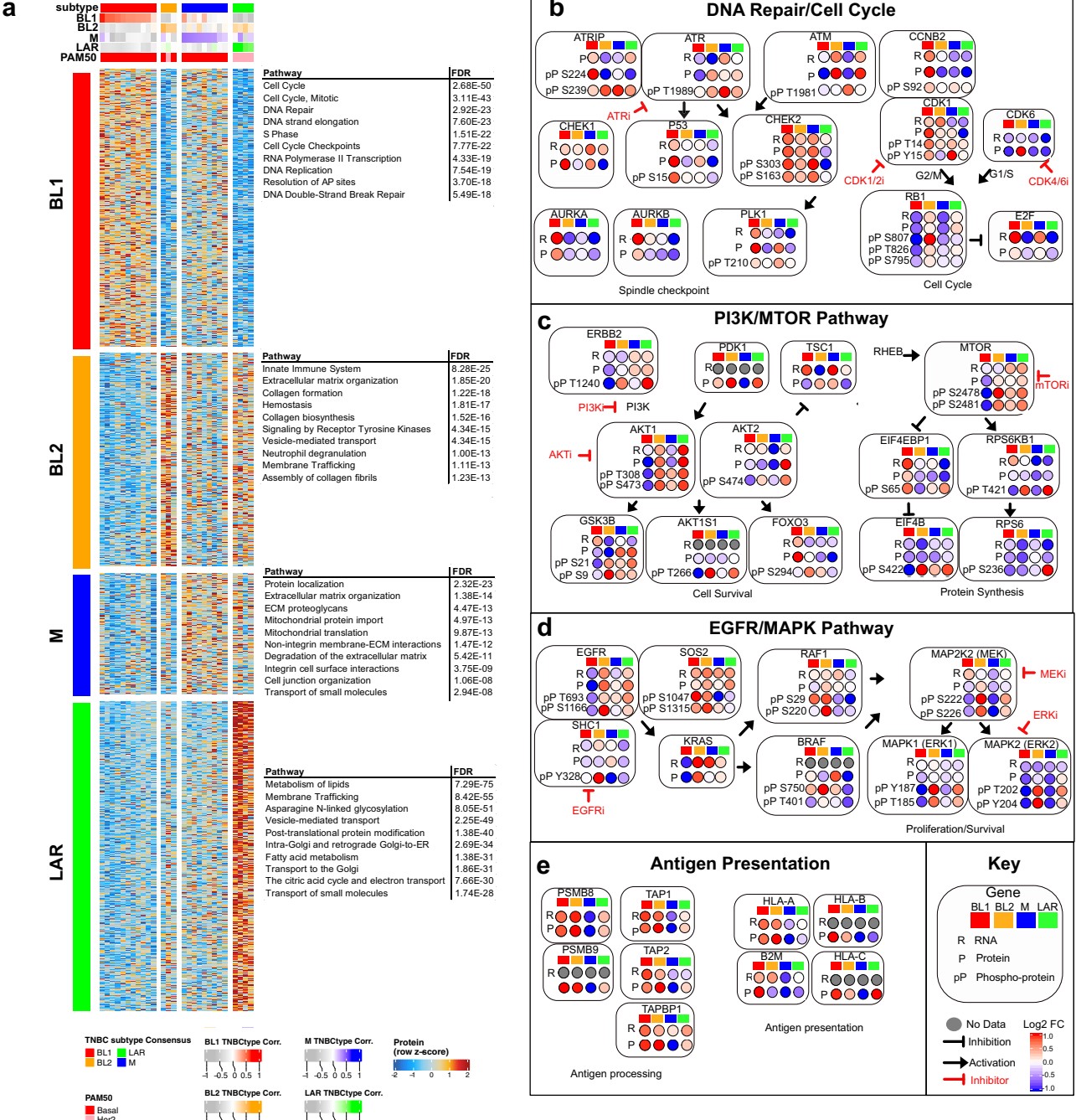

**Fig. 2 Gene expression and phosphoproteomic analyses identify unique subtype-specific targetable pathways. a** Heatmap shows differentially expressed proteins ($p < 0.05$ and Log2 FC >1) and significantly enriched pathways (Reactome) by subtype for TNBC tumors in CPTAC. Balloon plots show integrated pathway analysis of RNA, protein, and phosphoprotein log fold change values for genes/proteins from CPTAC TNBC tumors demonstrating subtype-specific differences in **b** DNA repair/cell cycle, **c** PI3K/mTOR, **d** EGFR/MAPK, and **e** antigen presentation pathways. FDR represents the false discovery rate that the normalized enrichment score represents a false-positive finding.

addition agents targeting DNA repair (BMN-673, gemcitabine, and camptothecin) (Fig. 3d).

The BL2-subtype displayed unique genetic dependencies on developmental pathways, with additional dependencies on DNA repair (*RAD50* and *TERF1*) and developmental genes *(WNT3, JAG1, NODAL, BMPR1A,* and *RSPO2)*, in addition to ETV5, *CDK6,* and ERK1/2 pathway (Fig. 3b and Supplementary Fig. 7e). BL2-subtype cell lines were uniquely sensitive to agents targeting DNA repair (olaparib and CP466722) and the MAPK pathway

inhibitors trametinib, PD0325901, refametinib, selumetinib, and CI-1040 (Fig. 3b and Supplemental Fig. 7f). In PDTX models, BL2-subtype was more sensitive to DNA alkylating agents (carboplatin, temozolomide, and cyclophosphamide) (Fig. 3d).

The M-subtype displayed dependencies on adhesion/motility genes, growth factor genes, and several transcription factors (Fig. 3b and Supplementary Fig. 7e). M-subtype cells were uniquely sensitive to the kinase inhibitors midostaurin, BX796 (PDK1), SL0101 (RSK), and ponatinib (RTK). M-subtype TNBC

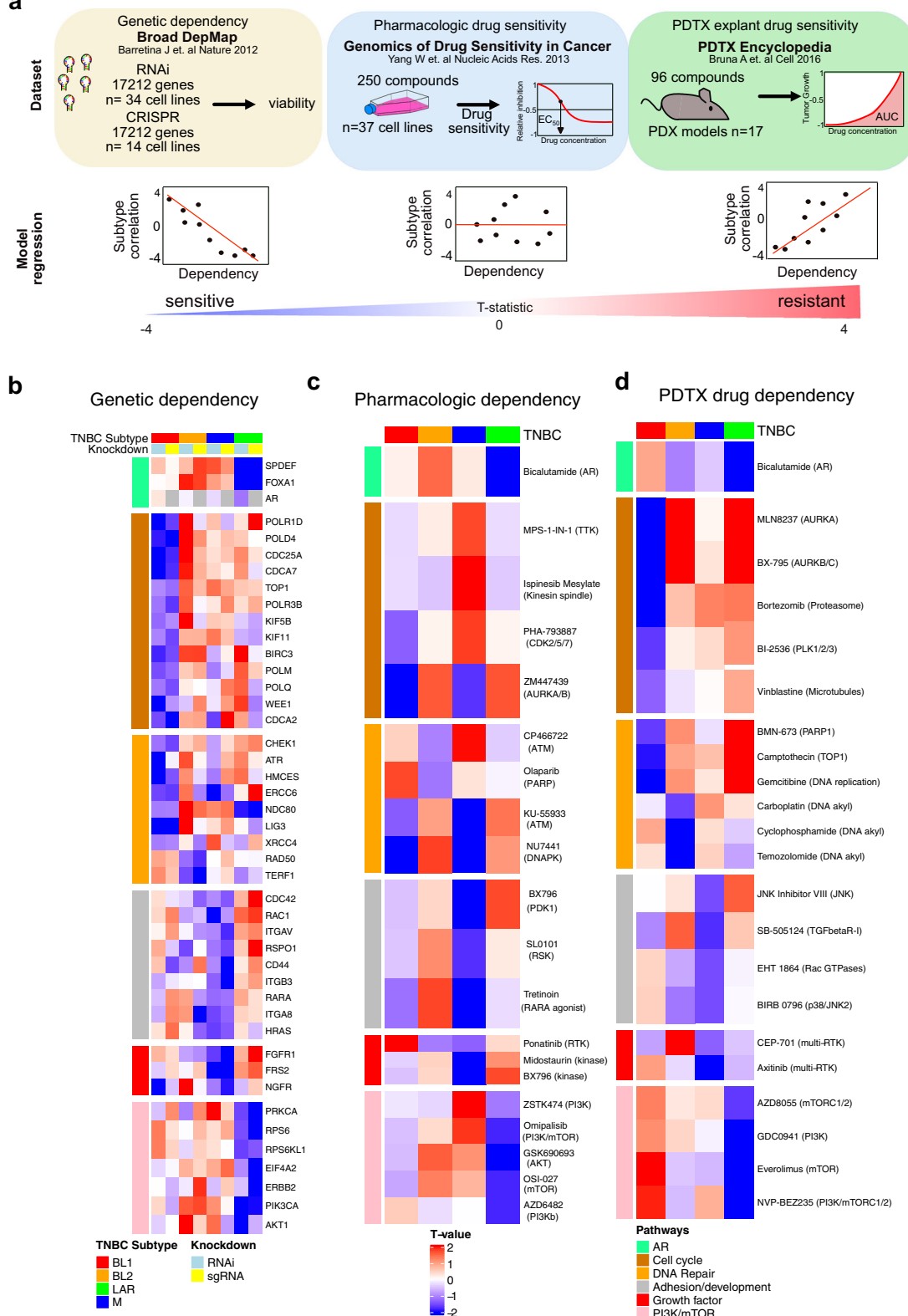

**Fig. 3 In silico analyses of datasets from genetic and pharmacologic screens identifies subtype-specific vulnerabilities in TNBC. a** Datasets and workflow for identifying subtype-specific genetic dependencies and pharmacologic sensitivities. TNBC cell lines identified from datasets and subtype correlation of all cell line models with values to yield a single subtype T-value for each drug sensitivity/genetic dependency. **b** Subtype-specific genetic dependencies from Broad DepMap whole-genome RNAi and CRISPR screen. Overlapping genetic dependencies and drug sensitivities from both screens are highlighted in the heatmap by pathway. Heatmaps show significant subtype-specific pharmacological dependencies determined with a modified T-test corrected for multiple hypothesis testing (T-value, FDR <0.1) in **c** TNBC cell lines screened in the Genomics of Drug Sensitivity in Cancer or **d** PDX explant drug sensitivity colored by similar pathway and subtype. See also Supplementary Data 6.

cell lines were also dependent on retinoic acid receptor alpha (*RARA*) and more sensitive to the retinoic acid receptor agonist tretinoin (Supplementary Fig. 7e and Fig. 3c). Retinoic acid derivatives have had success in differentiating acute promyelocytic leukemia[40], suggesting there may be differences in the epigenetic landscape between subtypes. Interestingly, M-subtype PDTX-derived models were resistant to most compounds but displayed some sensitivity to TGFßRI inhibitor SB-505124, two p38/JNK inhibitors (BIRB0796 and JNKi VII), the Rac inhibitor EHT1864, and the multi-RTK inhibitors axitinib and CEP-701 (Fig. 3d).

**TNBC subtypes differ in global DNA methylation patterns with specific methylation of EZH2 targets and expression antigen presentation genes in mesenchymal TNBC.** Relatively few potential therapeutic strategies were validated for the M-subtype and because this subtype is characterized by an absence of immune cells and LOF mutations in epigenetic modifiers, we examined the epigenetic landscape of each subtype by evaluating global DNA methylation patterns for TCGA TNBC tumors. Through this analysis, we identified differentially methylated CpGs between the TNBC subtypes (Fig. 4a and Supplementary Data 5). The LAR-subtype displayed the greatest amount of differentially hypermethylated CpGs, while the M-subtype displayed the most hypomethylated CpGs (Fig. 4b). The M-subtype displayed approximately double the number of differentially hypomethylated CpGs (74,288) as the BL1- (37,011), BL2- (23,033), or LAR-subtypes (30,020) and were dispersed evenly throughout the chromosomes (Fig. 4c). Global differences in DNA methylation primarily occurred in regions <3kb from promoters. Specifically, the majority of the hypermethylated regions of LAR-subtype tumors occurred in promoters, suggesting a more differentiated tumor (Fig. 4d).

Differentially methylated CpGs in promoter regions (<3kb from TSS) were used to identify genes with concordant gene expression unique to each TNBC subtype (Fig. 4e and Supplementary Fig. 8a). Despite overall decreased methylation, mesenchymal tumors displayed increased methylation and decreased expression in interferon-gamma (*IFNG*), immune checkpoint genes (*CD274*, *LAG3*, and *TIGIT*), and MHC-mediated antigen processing presentation (NLRC5, *CIITA*, *HLA-A*, *HLA-B*, and *TAP1*). Methylation near promoters of antigen presentation genes displayed lowered gene expression across M-subtype tumors (Supplementary Fig. 8b). Pathway analysis was performed on concordant methylation/expression differences to identify subtype-specific regulation (Fig. 4f). The M-subtype was hypermethylated in regions encompassing immune signaling (interferon-gamma signaling, IFNγ response, differentiating T lymphocyte, TNF response, and immune cytokine signaling) and antigen processing and presentation. Furthermore, EZH2 targets were suppressed in hypermethylated regions, suggesting a deregulated polycomb repressive complex 2 in this subtype (Fig. 4f).

To better understand the effects of DNA methylation changes on gene expression and chromatin structure, we analyzed distal methylation probes that overlapped with ATAC-seq peaks with predicted target gene expression identified by ELMER[41] (Supplementary Data 7). ATAC-seq alone was able to separate TNBC subtypes, providing evidence that chromatin accessibility may contribute to expression differences between subtypes (Supplementary Fig. 8c). Notably, the M-subtype displayed more hypomethylated regions that corresponded with increased chromatin accessibility and enhancer regions (Supplementary Fig. 8d). Interferon-gamma can upregulate the expression of MHC-I[42] and the transcription factors *NLRC5* and *CIITA* drive

MHC class I and II expression, respectively. Therefore, we evaluated the chromatin accessibility within these genomic regions to evaluate the potential differences between subtypes. ATAC-seq tracks showed decreased accessibility near the promoters of *IFNG, NLRC5,* and *CIITA* (Supplementary Fig. 8e). These findings suggest epigenetic DNA methylation as a mechanism to suppress antitumor immune function in M-subtype tumors.

**Inhibition of polycomb repressive complex 2 restores MHC-I expression in mesenchymal TNBC models.** Due to global epigenetic differences displayed by mesenchymal TNBC tumors, we next analyzed DNA methylation and trimethylation of lysine 27 on histone H3 (H3K27me3), indicative of PRC2 activity, in TNBC cell lines from the Broad Cancer Cell Line Encyclopedia (CCLE). Compared to the other subtypes, the M-subtype TNBC cell models displayed the lowest median DNA methylation (beta-value) and highest median H3K27me3 levels (Supplementary Figs. 9a, b). To evaluate levels of antigen presentation, we analyzed protein expression using a pan MHC-I antibody (HLA-A/B/C). M-subtype TNBC cell lines displayed the lowest levels of total MHC-I by immunoblot (Fig. 5a). Cellular MHC-I was then evaluated by IHC on a tissue microarray composed of fixed cell lines (Fig. 5b). MHC-I expression was localized to the cell membranes and the total number of MHC-I positive cells were lower in M-subtype cells (BT549: 0.1%, CAL51: 0.6%, HS578T: 4.7% and CAL120: 18.8%) compared to other subtypes (HCC1143: 29.7%, HDQP-1: 57.6%, MDA-MB-468: 65.7%, MDA-MB-436: 82.0% and HCC1937: 85.3%) (Supplementary Fig. 9c). Furthermore, cell surface MHC-I expression by flow cytometry revealed low expression in M-subtype cell lines that partially overlapped with unstained controls (Fig. 5c).

To evaluate the effects of PRC2 inhibition on viability/proliferation, TNBC cells were treated with increasing doses of the PRC2 inhibitors and viability was determined, relative to control, after four days. While very low doses tended to increase cell numbers, TNBC cells were only mildly sensitive to PRC2 inhibitors, and no compound reduced viability more than 50% even at 20 μM (Fig. 5d). To identify genes that are repressed by the PRC2 complex, we treated M-subtype cell lines with two different EZH2 inhibitors (tazemetostat or CPI-1205) or the EED inhibitor MAK-683, and then evaluated changes in gene expression five days later (Fig. 5e and Supplementary Fig. 9d). We identified 1663, 1463, and 2048 differentially regulated transcripts common to all inhibitors in cell lines CAL51, CAL120, and BT549, respectively (Fig. 5f). The vast majority of differentially expressed transcripts increased in expression with PRC2 inhibitor treatment (CAL51, 96.6%; CAL120, 91.2%; BT549, 93.8%). Gene ontology analysis of those transcripts that increased and shared between each of the cell lines ($n = 275$, union) were in pathways associated with polycomb repressor complex 2 and H3K27me3, indicative of decreased repression of PRC2 repressed genes (Supplementary Fig. 9e, f). Many of the additional elevated genes were involved in immune signaling, antigen presentation, and in interferon-gamma and inflammatory response pathways (Fig. 5f, g). Expression of MHC-I, and to some degree MHC-II, increased in M-subtype cell lines with PRC2 inhibitors and was correlated with increased expression of the transactivator of MHC-I (*NLRC5*) or MHC-II (*CIITA*) (Fig. 5h). To determine if MHC-I protein levels changed after PRC2 inhibition, we treated CAL51, CAL120, and BT549 cell lines with a single dose of either tazemetostat, CPI-1205, or MAK-683 and evaluated protein expression at 1, 3, 5, or 7 days. PRC2 activity was inhibited by each of the compounds as evidenced by steady decreases in H3K27me3 from day 3 to 7 (Fig. 5i). MHC-I protein

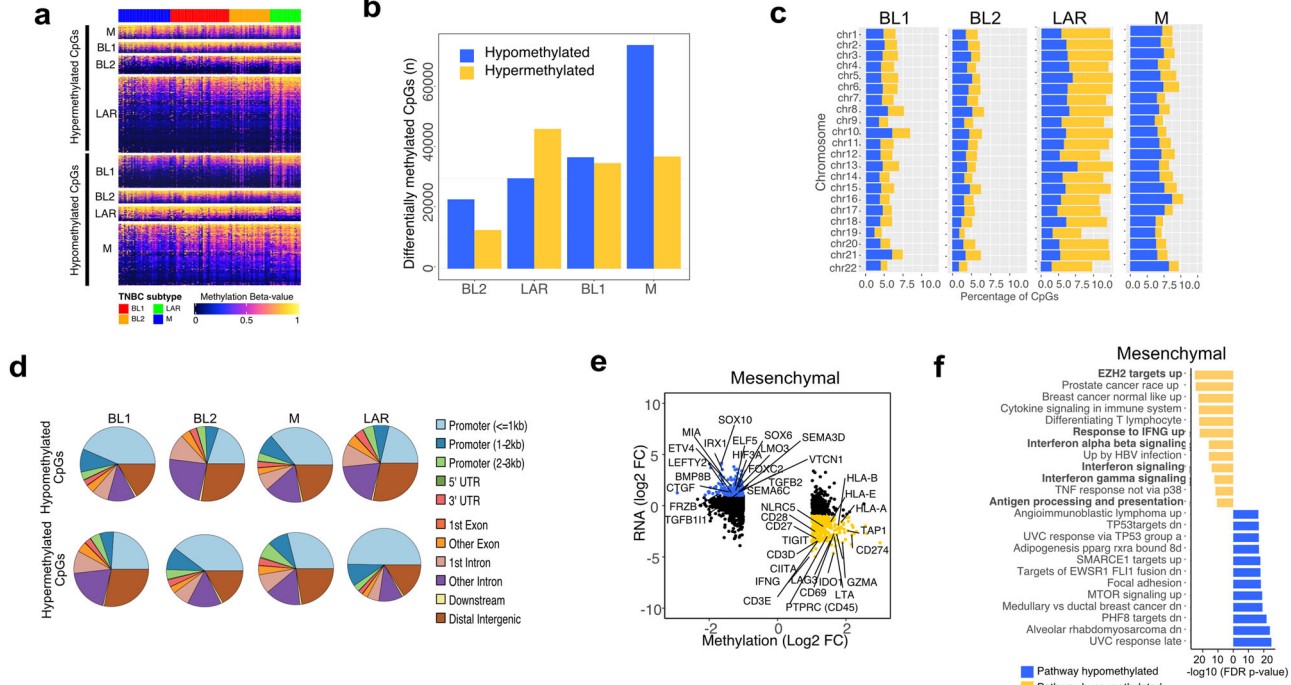

**Fig. 4 TNBC subtypes differ in global methylation patterns and mesenchymal TNBCs show specific methylation for EZH2 targets and antigen presentation genes. a** Heatmap shows all differentially (FDR <0.05 and β-value>0.1) methylated CpGs by subtype. **b** Graph shows the number of differentially) hypomethylated and hypermethylated CpGs by TNBC subtype. **c** Barplots show the percentage of hypomethylated (blue) and hypermethylated (yellow) CpGs within each chromosome by TNBC subtype. **d** Pie charts show the distribution of differentially methylated CpG sites in each subtype by genomic location. **e** Starburst plots show gene expression and DNA methylation <3kb from promoter regions of corresponding genes. Significantly (FDR <0.05) hypo- and hyper-methylated promoter regions are colored in blue and yellow respectively, with genes of interest labeled in the plot. **f** Gene set enrichment analysis of genes negatively correlated with hypomethylated (blue) or hypermethylated (yellow) promoter methylation probes.

expression increased with PRC2 inhibition as H3K27me3 protein levels decreased. Cell surface MHC-I protein expression also increased around two-fold for each of the inhibitors (Fig. 5j). The increased MHC-I expression was specific to M-subtype cells, as PRC2 inhibitors decreased H3K27me3 without altering MHC-I protein levels in other TNBC subtypes (Supplementary Fig. 10a–c). The increased MHC-I expression in mesenchymal TNBC cells treated with PRC2 inhibitors has the potential to increase the antitumor immune response.

**Decreased H3K27me3 repressive marks at MHC-I locus with EZH2 inhibition.** To determine whether re-expression of HLA genes with EZH2 treatment corresponds to a demethylation of H3K27me3, we performed H3K27me3 chromatin immunoprecipitation sequencing (ChIP-Seq) of mesenchymal TNBC cell lines 4 days after treatment with either DMSO or 1 μM of the EZH2 inhibitor tazemetostat (Supplementary Fig. 11a). ChIP-seq analysis revealed decreased genome-wide H3K27me3 deposition in BT549 ($n = 221,518$), CAL120 ($n = 54,339$) and CAL51 ($n = 58,967$) cells treated with tazemetostat (Fig. 6a). Furthermore, H3K27me3 was decreased in EZH2 targets[43] and H3K27 bound genes[44]. Significantly decreased (FDR < 0.05) peaks with EZH2 inhibition were similar between cell lines with ~47% of all peaks shared by at least two cell lines and 16% shared between all three cell lines (Supplementary Fig 11b). Gene set enrichment analysis of peaks within 3 kb of promoters common to all three cell lines were enriched in H3K27me3, EED, and SUZ12 targets (Supplementary Fig. 11c). To identify gene expression changes related to decreased H3K27me3, we examined RNA expression levels relative to promoter H3K27me3 peaks that decreased with EZH2 inhibition. It was noted that many genes with increased

expression and decreased promoter H3K27me3 occurred in HOX-associated genes (Fig. 6b). The PRC2 is known for silencing homeobox (HOX) gene loci[45] and may also contribute to the repression of MHC-I expression. In addition, we observed decreased H3K27me3 and increased expression of MHC-I genes, as well as transactivation NLRC5, in mesenchymal cells treated with tazemetostat (Fig. 6b). Genomic snapshots of MHC-I loci confirmed decreased H3K27me3 in promoter regions of HLA-A (BT549), HLA-B (BT549 and CAL120), HLA-C (CAL120 and CAL51), and at several locations corresponding to enhancer regions within the MHC-I transactivator NLRC5 (Supplementary Fig. 11d).

**EZH2 inhibition increases paclitaxel efficacy in a syngeneic xenograft model.** To determine if PRC2 inhibition could be therapeutically beneficial to M-subtype, we identified an "immune cold", mesenchymal syngeneic murine xenograft TNBC tumor model. Similar to human cells, PRC2 inhibitors effectively reduced H3K27me3 in murine 4T1 cells (Fig. 7a). 4T1 cells also displayed low levels of MHC-I expression at the cell surface, which could be increased > two-fold after treatment with PRC2 inhibitor (Fig. 7b, c). Syngeneic 4T1 xenografts were established in mice and treated with either vehicle, bi-weekly paclitaxel, twice daily tazemetostat, or the combination (Fig. 7d). Treatment with either paclitaxel (average 382 mm³) or tazemetostat (average 433 mm³) alone resulted in moderate reductions in final tumor volume compared to vehicle-treated mice (average 643 mm³), while the combination (average 237 mm³) was significantly better than each agent alone (tazemetostat $p = 0.015$, paclitaxel $p = 0.049$) (Fig. 7e). Paclitaxel or tazemetostat alone did not reduce final tumor weight, however, the combination resulted in

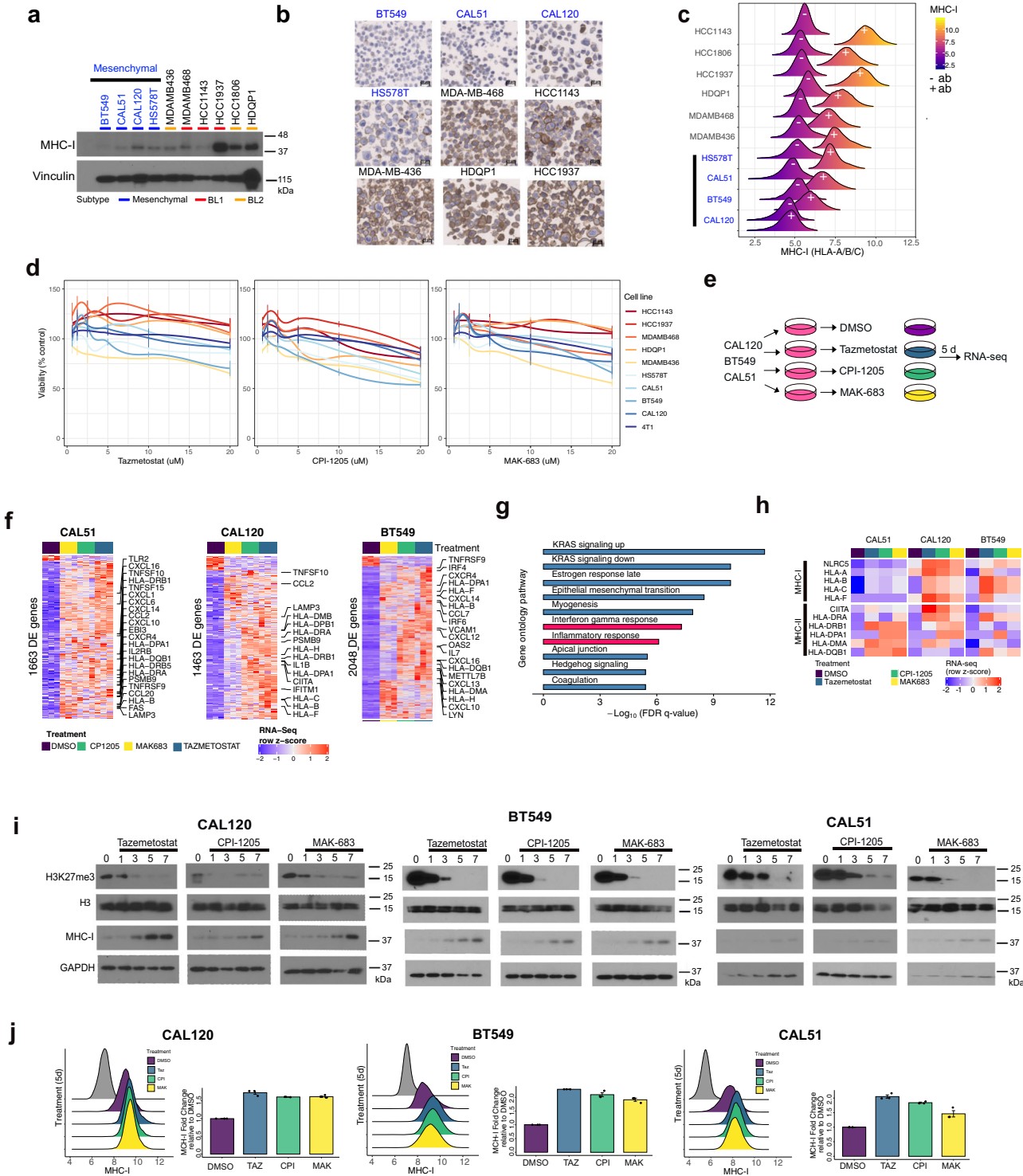

significantly ($p < 0.05$) reduced tumor size compared to control mice (Fig. 7f). We performed serial IHC staining for H3K27me3, Ki67, and cleaved caspase-3 on resected tumors. Vehicle-treated tumors were larger with proliferating (Ki-67+) cells along the periphery of the tumor and cleaved caspase-3 positive cells confined to the necrotic core (Supplementary Fig. 12a). H3K27me3 expressing cells were confined to proliferating cells and staining was less intense in tazemetostat-treated tumors (Supplementary Fig. 12b, d). In contrast to vehicle-treated mice, cleaved caspase-3 was present outside of the necrotic core in tumors treated with paclitaxel or tazemetostat (Supplementary Fig. 12a–d). IHC for

CD3+ T-cells was performed to determine if treatment altered tumor-infiltrating lymphocytes. Intratumor CD3+ T-cells increased with each treatment (paclitaxel 1.68-fold, tazemetostat 1.85-fold) and with the combination (2.05-fold) displaying the greatest increase (Fig. 7g), potentially contributing to the decreased tumor size.

## Discussion
TNBC is currently treated as a single disease, however, our analyses revealed additional subtype-specific features within

**Fig. 5 Inhibition of polycomb repressive complex 2 restores MHC-I expression in mesenchymal TNBC models. a** Immunoblot shows levels of MHC-I (HLA-A/B/C) across TNBC cell lines. Blots are representative of two independent experiments. **b** Representative images show immunohistochemistry for MHC-I expression on individual cells in TNBC cell lines. **c** Histogram plots show the distribution of membrane-bound MHC-I protein (+) in TNBC models compared to unstained controls (−). Images are representative of three independent cores from a tissue microarray. **d** Plots show relative viability of TNBC cell lines treated with increasing concentrations (1.25, 2.5, 5, 10, and 20 μM) of tazemetostat, CPI-1205, or MAK-683. Error bars represent the standard deviation of three independent experiments. **e** Diagram shows experimental workflow and **f** heatmaps show differential gene expression of M-subtype TNBC cell line models after 5 days of a single 10 μM treatment with inhibitors of the PRC2 complex, tazemetostat (EZH2 inhibitor), CPI-1205 (EZH2 inhibitor) or MAK-683 (EED inhibitor). **g** Gene ontology analysis (Hallmark) of the gene in the union (Supplementary Fig. 9e) between CAL51, CAL-120, and BT549 cell lines that were significantly (FDR p-value <0.05, FC>2) upregulated between treatment with PRC2 inhibitors (n = 9) compared to DMSO treatment (n = 3). Differential genes identified by modified T-test corrected for multiple hypothesis testing. **h** Heatmap shows an expression of MHC-I and MHC-II genes in M-subtype TNBC cell lines treated with control (DMSO) or PRC2 inhibitors. **i** Immunoblots show H3K27me3 and MHC-I protein expression at 1, 3, 5, 7 days after a single 10 μM treatment with either tazemetostat, CPI-1205, or MAK-683. Immunoblots are representative of two experiments. **j** Histograms show the distribution of cell-surface MHC-I protein expression 5 days after a 10 μM treatment with the indicated PRC2 inhibitors. Error bars were determined from three independent experiments.

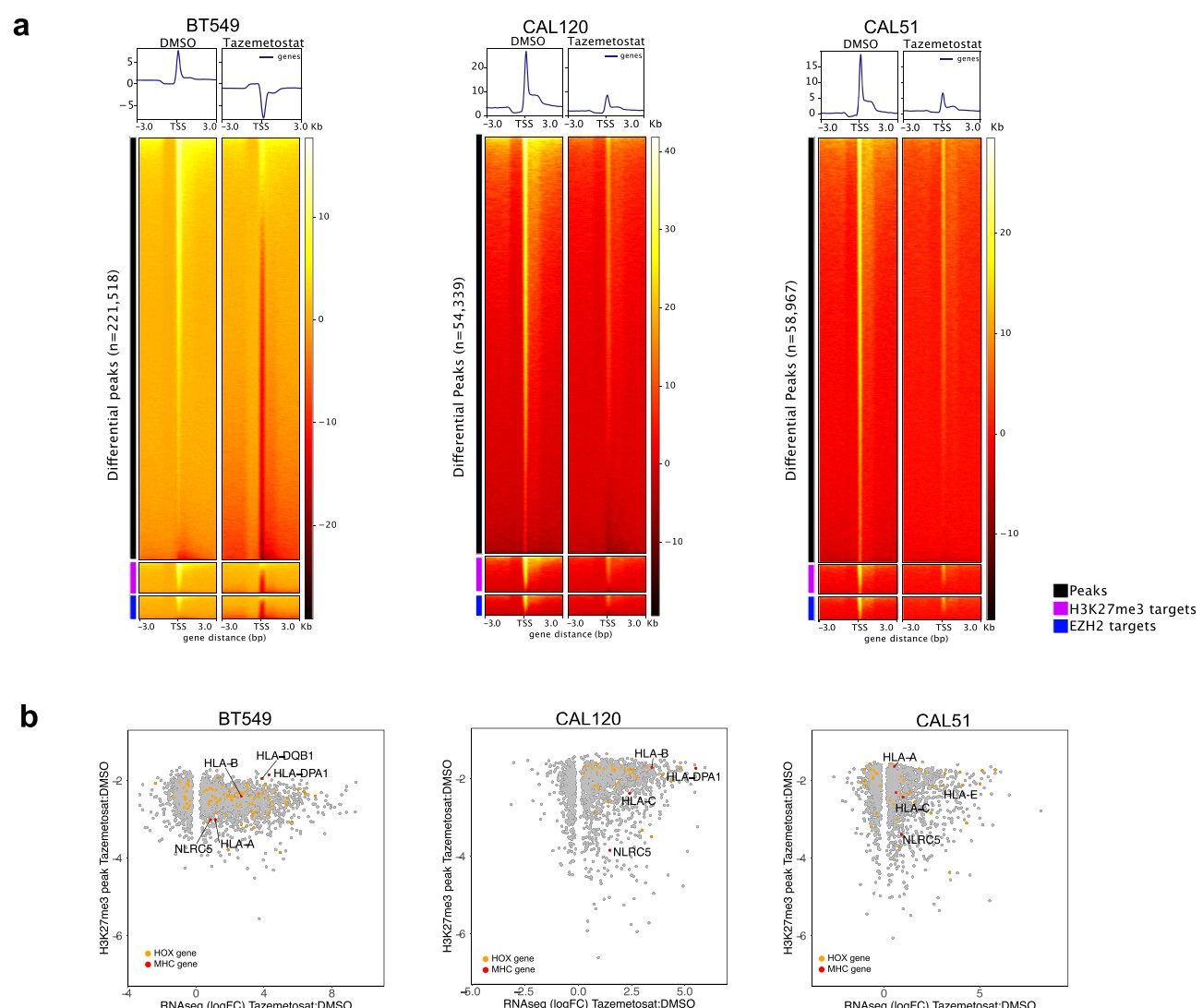

**Fig. 6 EZH2 inhibition decreases global H3K27me3 and repressive marks at MHC-I locus. a** Profile plot and heatmap for H3K27me3 ChIP-seq signal for differential peak, H3K27me3 targets (BENPORATH_ES_WITH_H3K27ME3), or EZH2 targets (NUYTTEN_EZH2_TARGETS_UP) centered on transcriptional-start sites (TSS) for BT549, CAL-120, and CAL-51 cells treated for 4 days with either DMSO or 1 μM tazemetostat. Sequencing reads were normalized to reads per genomic content. **b** Scatterplots show differential RNA expression (Log2 FC, FDR <0.05) and differential H3K27me3 promoter occupancy (FDR <0.0.5) in tazemetostat treated cells relative to DMSO treatment.

mutational patterns, CNAs, protein pathway activation, epigenetic backgrounds, and immune microenvironments. These extensive genomic and epigenomic differences suggest that TNBC subtypes could arise from different cells of origin. Supporting this

hypothesis is the differential correlation to scRNA signatures derived from normal breast epithelium cells[21]. BL1-subtype tumors were closely related to the bi-potent L1.2 luminal progenitor cells, aligning with prior studies in BRCA-mutated mouse

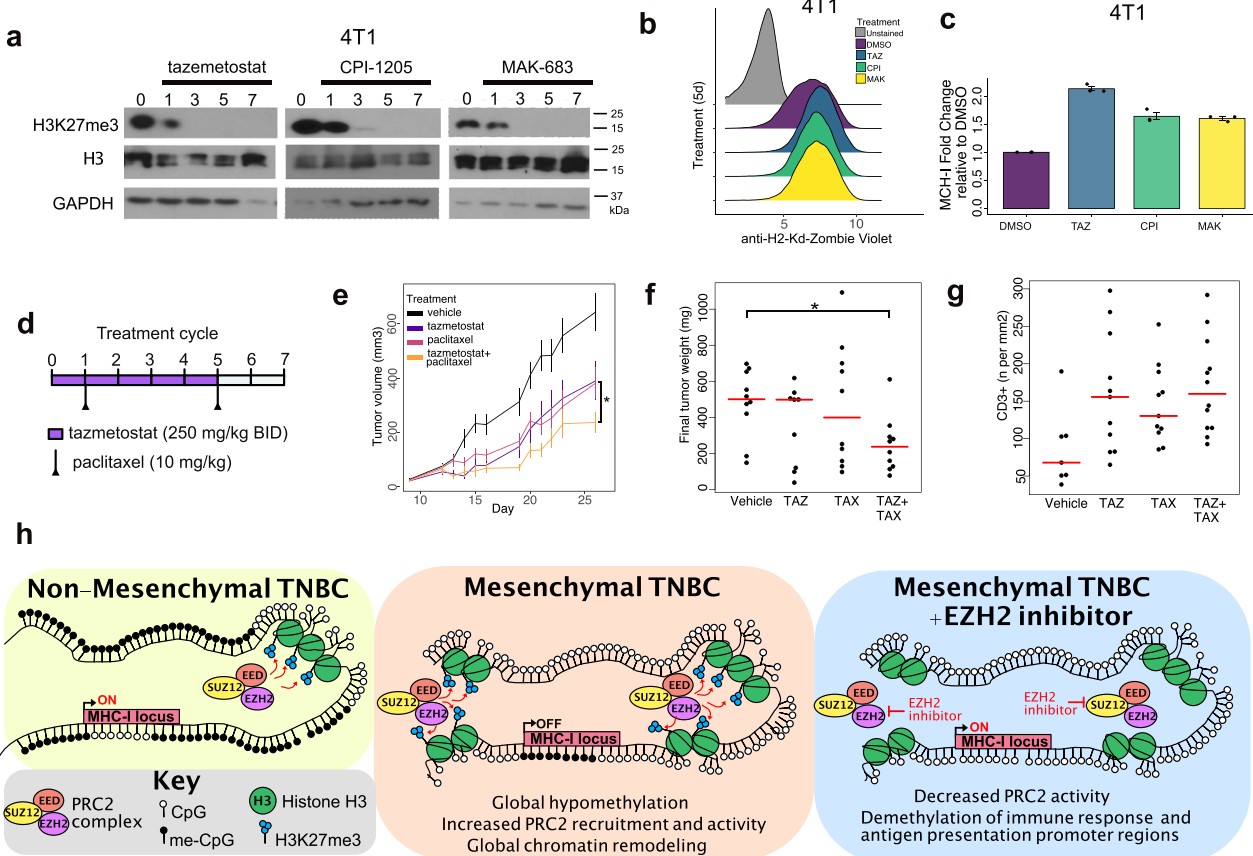

**Fig. 7 EZH2 inhibition increases the efficacy of paclitaxel chemotherapy in a syngeneic murine TNBC model. a** Immunoblots show H3K27me3 and MHC-I protein expression at 1, 3, 5, 7 days after a single 10 μM treatment with tazemetostat, CPI-1205, or MAK-683. **b** Histograms show the distribution and **c** quantification of cell-surface MHC-I protein expression 5 days after a 10 μM treatment with the indicated PRC2 inhibitors. Results are representative of three experiments. Error bars represent the standard error of the mean. **d** Treatment schedule for mice bearing syngeneic 4T1 xenograft tumors. Mice were treated with vehicle, twice daily (BID) with 250 mg/kg tazemetostat, twice a week with 10 mg/kg paclitaxel, or the combination of tazemetostat and paclitaxel. Results are representative of ten tumors. **e** Graphs show 4T1 tumor volume (mm³) across time of mice treated with vehicle, tazemetostat, paclitaxel, or the combination. Error bars represent the standard error of the mean. Significance determined by two-tailed Student's t-tests, *$p = 0.0353$. **f** Barplot shows the distribution of final tumor weight (mg) from mice treated with vehicle, tazemetostat (TAZ), paclitaxel (TAX), or the combination (TAZ + TAX). Significance determined using Dunnett's multiple comparison test. *$p = 0.0423$. **g** Plot shows IHC quantification of intratumor CD3+ T-cells in 4T1 xenograft tumors by treatment group. **h** Schematic shows CpG methylation and H3K27me3 epigenetic states in non-mesenchymal TNBC or mesenchymal TNBC with or without EZH2 inhibition.

models, suggesting luminal cells are the cell-of-origin for basal-type breast cancer[46]. The LAR-subtype was closely related to the L2 hormone-responsive cells, consistent with the differentiated luminal state and the observed dependency on hormone signaling. Both the BL2- and M-subtypes correlated to myoepithelial/basal cells. However, the global DNA hypomethylation observed in the M-subtype may be more indicative of a de-differentiated cell. The similar mutation and CNAs between M- and BL1-subtypes suggest that the M-subtype may be derived from the BL1-subtype. The consensus clustering and scRNA analyses support the possibility of a transition state between the BL1- and M-subtypes, in which some individual tumors are composed of both subtypes. These data suggest the distinct TNBC subtypes may arise from different cells-of-origin or transitions between cell states, which likely lead to differential sensitivity to therapeutic agents.

Apart from BRCA1/2-mutated tumors, combination chemotherapy alone or with immune checkpoint therapy is the standard of care for TNBC. Our genomic analysis demonstrates that LAR-subtype tumors are more stable, carry lower mutational burdens, and display decreased cell cycle pathway activation by protein analysis, providing mechanisms explaining the lower pathological complete response rates to neoadjuvant chemotherapy reported in

other retrospective studies[47]. LAR-subtype tumors are enriched in activating *PIK3CA* and *ERBB2* mutations and display increased protein and phospho-protein for AR, ErBb2, and AKT signaling. Cell line models displayed genetic dependencies on *AR*, *FOXA1*, *ERBB2*, *PIK3CA*, and *AKT*, and sensitivity to the AR inhibitor bicalutamide and a variety of PI3K, AKT, and mTOR inhibitors. The next-generation AR antagonist enzalutamide is currently being evaluated in AR-positive (AR+) TNBC in combination with paclitaxel (NCT02689427), and a recent clinical trial demonstrated increased clinical benefit rate in metastatic AR+ TNBC receiving the combination of enzalutamide and the PI3K inhibitor taselisib[9]. While not amplified, the presence of *ERBB2* mutations, elevated protein, and genetic and pharmacologic dependencies of cell line models, suggest that ERBB2 inhibitors may be another approach for treating patients with LAR-subtype tumors.

Similar to the LAR-subtype, BL2-subtype tumors displayed lower mutational levels, copy number complexity, and proteomic/phosphoproteomic data suggesting an intact G1/S checkpoint, whereas M- and BL1-subtypes displayed copy number loss and low Rb protein levels. While LAR-subtype cell lines were genetically dependent on *CDK4* and *CCND1*, BL2-subtype tumors displayed elevated CDK6 protein, and cell lines were

uniquely sensitive to *CDK6* knockdown, suggesting that CDK4/6 inhibitors may be more beneficial in these subtypes. The CDK4/6 inhibitor palbociclib is currently under investigation in combination with bicalutamide in AR+ TNBC (NCT03090165). In addition, we found that BL2-subtype tumors were uniquely enriched with activating MAPK pathway mutations, *KRAS* amplification, and activated MAPK protein signaling. Furthermore, BL2-subtype cell lines models were more sensitive to MEK inhibitors. Although combined MEK and PI3K inhibition have shown promising results in preclinical PDX metaplastic TNBC models[48], the combination of GSK1120212 with BKM120 resulted in significant toxicities in TNBC patients (NCT01155453)[49]. Perhaps another look should be taken at MEK inhibitors in the BL2-subtype or TNBCs that harbor genomic mutations/amplifications in the MAPK pathway.

In contrast to the LAR- and BL2-subtypes, both the BL1- and M-subtypes displayed higher mutational burdens, evidence of DNA homologous repair deficiency, increased mutational heterogeneity, and increased genomic instability. We identified *MYC* amplification and overexpression in the M- and BL1-subtypes, *CCNE1* amplification and overexpression in the BL1-subtype, and NOTCH3 amplification and overexpression in the M-subtype, similar to a prior subtype analyses in both TCGA and METABRIC[24]. BL1- and M-subtypes also displayed increased G2/M and DNA repair protein expression, and cell models displayed sensitivity to CDK1/2, AURKA/B, ATM, PARP, and DNA-PK inhibitors. *NOTCH1/2/3* mutations primarily occurred in BL1- and M-subtypes. *NOTCH* PEST domain mutations have been shown to be oncogenic and sensitive to γ-secretase inhibitors in TNBC[50], and may be a therapeutic option for BL1- and M-subtype tumors with mutated or amplified *NOTCH*.

TMB typically predicts better survival in breast cancer to standard chemotherapy[23], however, this is not the case for the M-subtype. Despite similarities with the BL1-subtype, the M-subtype is characterized by a lack of immune cells and exhibits lower global DNA methylation. The histological evaluation confirmed the M-subtype is largely devoid of immune cells or cells are constrained to tumor margins. The M-subtype displayed decreased antigen presentation and B2M deletions. Mutations, loss of heterozygosity, and deletions in B2M were previously identified in non-responding melanoma patients to immunotherapy[31]. This suggests the absence of MHC-I antigen presentation may be an effective means of immune escape. The increased mutational burden and global hypomethylation detected in the M-subtype are consistent with previous observations that genomic hypomethylation often correlates with immune escape signatures in cancers with high mutational burden[51]. In melanomas, hypomethylation results in epithelial-mesenchymal changes and increased PD-L1 expression[52]. However, this is not the case for M-subtype TNBC, in which the majority of tumors have low PD-L1 expression, rendering metastatic patients' ineligible for combinations of immune checkpoint inhibitors and chemotherapy.

Although M-subtype tumors displayed global DNA hypomethylation, there were distinct regions of DNA hypermethylation and concordant decreased gene expression in interferon gamma-induced genes, antigen presentation, and EZH2 gene targets. M-subtype cell line models displayed the lowest levels of cell surface MHC-I expression. Using cell line models, we demonstrate that MHC-I expression is epigenetically silenced by the polycomb repressor complex and restored by EZH2 inhibitors in mesenchymal cell lines. ChIP-seq analysis after EZH2 inhibition demonstrated that increased MHC-I expression is, in part, related to decreased promoter H3K27me3 occupancy at the MHC-I loci and the transactivator NLRC5. Several recent studies have also demonstrated EZH2 inhibition can enhance tumor cell

antigen presentation in head and neck squamous cell carcinoma[53], diffuse large B-cell lymphoma[54], and melanoma[55]. EZH2, the catalytic subunit of the polycomb repressor complex, methylates histones and represses gene expression through modifying chromatin structure. Interestingly, we observed increased DNA methylation in PRC2-regulated genes in M-subtype tumors. Previous studies have demonstrated that EZH2 can recruit DNA methyltransferases, resulting in DNA methylation and subsequent gene silencing[56].

EZH2 expression has been shown to promote TNBC tumors in the mammary glands of transgenic mice expressing a phospho-mimicking (T416D) EZH2 mutant[57]. While non-Hodgkin's lymphoma and melanomas have activating EZH2 mutations[58], and overexpression of EZH2 has been reported in solid tumors[59], we did not observe these alterations in mesenchymal TNBCs. This suggests PRC2 activation may result from indirect regulation rather than alterations in the complex itself. Epigenetic antagonism has previously been observed between the polycomb repressor complex and SWI/SNF BAF complex, in which polycomb targets are H3K27-trimethylated in SNF5 (SMARCB1 homolog) deficient cells[27]. The increased loss-of-function mutations and deletions in SWI/SNF genes in M-subtype tumors suggest this may be a potential mechanism.

EZH2 inhibitor anti-tumor activity was previously demonstrated in refractory B-cell non-Hodgkin's lymphoma and epithelioid sarcomas[60] and are currently in clinical development with tazemetostat in refractory B cell non-Hodgkin's lymphoma (NCT03456726), CPI-1205 in advanced solid tumors (NCT03525795), and the EED inhibitor MAK-683 in diffuse large B-cell lymphoma (NCT02900651). We did not observe substantial cytotoxicity of PRC2 inhibitors in TNBC cells, and therefore PRC2 inhibitors will unlikely be effective as a monotherapy in TNBC. The EZH2 inhibitor tazemetostat moderately impacted the syngeneic xenografts tumor growth and enhanced the efficacy of taxane chemotherapy significantly reducing overall tumor size and increasing intratumoral T cells.

Using integrative analyses of TNBC multi-omics we have identified additional characteristics of TNBC subtypes and subtype-specific genetic/pharmacologic vulnerabilities for future investigation. Our pre-clinical data provide a strong rationale for the use of EZH2-inhibitory agents as a strategy for restoring MHC-I expression in immune cold, PD-L1 negative, M-subtype tumors.

## Methods

**Datasets**. We performed a bioinformatic analysis on publicly available transcriptomic and genomic data from the following breast cancer datasets. TCGA study is hosted by the NCI's Genomic Data Commons (GDC) https://portal.gdc.cancer.gov/ and contains RNA-seq, DNA methylation, ATAC-seq, copy number, mutation, and clinical data. The CPTAC dataset contains mass spectrometry-based proteomics with next-generation DNA and RNA sequencing profiles on 122 treatment-naive primary breast cancers[14]. The METABRIC dataset consists of normalized RNA microarray profiling on 1981 fresh-frozen primary breast cancer samples on the Illumina HT-12 v3 array[15]. The MET500 dataset contains expression profiles from 500 metastatic samples obtained from 22 different organs were processed as described in Robinson et al.[16].

**Genomic-guided identification of TNBC specimens**. TNBC samples were identified by evaluating the distribution of ER, PGR, and HER2 using RNA, protein (RPPA/mass spec), and DNA copy number annotated with clinical assessment (IHC and FISH) provided by TCGA and CPTAC (Supplementary Fig. 1). The empirical cutoffs for hormone receptor status and *ERBB2* amplification were manually defined from the bimodal distribution of genomic data and distribution of known clinical annotations (IHC for all markers and FISH for HER2) when available. For datasets with several genomic datasets (i.e., mRNA expression, protein expression, and CN levels for *ERBB2*) two lines of genomic evidence required to override a clinical call or infer status when no clinical information was available (Supplementary Figs. 1a, 1b and Supplementary Data 1).

For TCGA and CPTAC, RNA-seq mRNA expression and RPPA data were used to define ER and PR cutoffs, while *ERBB2* copy number data was also included to

define HER2 cutoffs. There was a substantial correlation between clinical and genomic calls for ER (95.1%) PR (90.0%) and HER2 immunohistochemistry (89.3%) and HER2 fluorescence in situ hybridization (91.0%) in TCGA (Supplementary Data 1). In total, we identified 192 (17.5%) TNBC tumors from 1097 patients in TCGA and 28 (23.0%) TNBC from 122 CPTAC BRCA tumors. For METABRIC, mRNA expression distributions for ER, PR, and HER2 with clinical annotations for ER and HER2 were used to infer hormone status (Supplementary Fig. 4a). In METABRIC, we identified 348 (17.6%) TNBC patients from 1981 BRCA patients. The distribution of inferred clinical subtypes was remarkably similar between TCGA and METABRIC with TNBC (17.5% vs. 17.6%), ER+ (71.4% vs. 70.9%), and HER2+ (10.6% vs. 11.6%). Using mRNA expression cutoffs for the metastatic MET500 dataset, we identified 40 unique TNBC patients (Supplementary Fig. 4b). There were less ER+ (46.7%) and HER2+ (9.8%) tumors in the MET500 dataset and enrichment in TNBC tumors (43.5%), consistent with the increased frequency of metastasis[61].

**RNA expression data analysis.** TCGA BRCA RNA-seq RSEM gene-level counts of 1211 cases, including 1097 primary solid tumor tissues and 114 solid tissue normal, aligned to the hg19 reference genome were downloaded from GDC's legacy archive. The 192 TNBC samples identified in the previous section were extracted and normalized (TCGAanalyze_Normalization) using the R/Bioconductor package TCGAbiolinks (ver.2.9.5)[62]. Next, we performed sample normalization adjusting for GC-content and upper-quantile between-lane by applying and implementing the EDASeq protocol[63]. For the 122 prospective CPTAC BRCA samples[64], we obtained the median-normalized gene expression data (log2 FPKM) from linkedomics (http://linkedomics.org/data_download/CPTAC-BRCA/). RNA expression for the 28 TNBC tumors identified using the "genomic-guided identification of TNBC specimens" section were normalized and subtyped as detailed in methods section "TNBC subtyping" below. MET500 (FPKM) RNAseq data (n = 868) was retrieved from https://xenabrowser.net/[16]. We identified 92 unique breast tumors within the MET500 dataset, among them 40 were classified as TNBC samples that underwent TNBC subtyping. Complete normalized expression data for 1981 METABRIC BRCA samples profiled with Illumina HT 12 arrays were obtained through Synapse (https://www.synapse.org/#!Synapse:syn1757063).

**TNBC subtyping.** The TNBCtype web-based tool (http://cbc.mc.vanderbilt.edu/tnbc/) was used to classify each TNBC tumor into four TNBC subtypes using centroid correlation to the BL1, BL2, LAR, and M subtypes using the highest positive centroid correlation[65]. For the TCGA cohort, TNBC subtyping was performed on normalized mRNA expression from 192 TNBC tumor samples identified in the "genomic-guided identification of TNBC specimens" section. TNBC subtypes were assigned to either BL1, BL2, M, or LAR based on the subtype with the highest centroid correlation and p-value. Nine samples could not be assigned to a definitive subtype due to low correlations to all subtypes, so these were deemed unclassified (UNC). The remaining 183 TGCA TNBC tumors were classified into 64 BL1, 37 BL2, 27 LAR, and 55 M TNBC subtypes. For CPTAC, we identified 27 tumor samples and performed TNBC subtyping using normalized mRNA expression data. The CPTAC TNBC cohort was classified into 11 BL1, 3 BL2, 4 LAR, and 9 M TNBC subtypes. For the MET500 dataset, we identified 40 TNBC tumor samples and performed TNBC subtyping using normalized mRNA expression data. The MET500 TNBC cohort was classified into 11 BL1, 14 BL2, 9 LAR, and 11 M TNBC subtypes. TNBC subtyping was performed on normalized mRNA expression for the 348 METABRIC TNBC tumor samples. The METABRIC TNBC cohort was classified into 124 BL1, 70 BL2, 83 LAR, and 71 M TNBC subtypes.

**Consensus clustering.** We performed unsupervised k-means consensus clustering using the 5,000 genes with the highest standard deviation (SD), 500 repetitions, 80% of subsampling for each repetition, and k-varying from a minimum of 2 clusters up to 10 clusters. The optimal number of clusters was determined from the cumulative distribution function (CDF), which plots the corresponding empirical cumulative distribution. The number of clusters is decided when any further increase in cluster number (k) does not lead to a corresponding marked increase in the CDF area. The R packages ConsensusCluterPlus (ver.1.54.0) and CancerSubtypes (ver.1.16.0) were used to perform this analysis. Code to reproduce the analysis is available at https://github.com/TransBioInfoLab/TNBC_analysis/blob/master/analysis/TCGA/consensusCluster.R.

**TNBC scRNA analysis.** RSEM counts for scRNA from six TNBC tumors from GSE118389[19] were downloaded and a normalized (NormalizeData) Seruat object was created (CreateSeuratObject, min.cells = 3, min.features = 200) with the R package Seruat (ver. 3.2.3)[66]. For clustering, the following parameters were used: RunPCA; RunUMAP, dims = 1:30; FindNeighbors, dims = 1:30; and FindClusters. UMAP plots were generated and colored by expression levels of cell lineage markers to identify cell populations. The clusters showing expression of immune cell (CD3E, CD79A, and CD14), endothelial cell (PECAM1), and myoepithelial (TP63) markers were removed. Additional Seruat objects were created from the remaining epithelial cells expressing EPCAM for all the epithelial cells together or within each tumor. Pseudobulk expression was derived from the sum of the

normalized RSEM expression for all epithelial cells within a given tumor. TNBC subtyping was performed on normalized expression from each individual epithelial cell and the pseudobulk composite for each tumor.

**TNBC subtype association testing for omics data.** To identify significantly enriched subtype-specific associations with genomic data, and account for tumors with multiple subtype correlations, we performed testing using the correlation strength for each subtype individual to each genomic dataset. For TCGA RNA-seq, DNA methylation data, CPTAC RNA-seq, proteomics, phosphorylation data, limma voom[67] (RNASeq only), and limma[68] were applied to test the association of omics features with subtype correlation coefficients of each subtype from TNBCtype. Clinical variables, including patient age and tumor stage, were adjusted in the limma voom/limma regression models. For hidden batch effects, svaseq and sva[68] implemented in Bioconductor package sva (ver.3.40.0) were used to estimate the surrogate batch effect variables, which were also adjusted in limma (ver.3.48.3) voom/limma models. To adjust for multiple comparisons, the false discovery rate was estimated using the Benjamini-Hochberg procedure. The full tables can be found in Supplementary Data 5. Chi-squared testing was used to determine significant clinical associates with subtype in Table 1.

**Single sample gene expression pathway analysis.** To further investigate gene programs enriched by each TCGA TNBC sample, we employed a single sample gene set enrichment analysis (ssGSEA) method from the GSVA R package (ver.1.40.1)[69]. We then used ten collections from the Molecular Signatures Database as follows: H: hallmark gene sets, C2: KEGG pathway database, C2: REACTOME pathway database. For this analysis we considered all 192 TCGA TNBC normalized gene expression data. The heatmap for the collections (Hallmark, KEGG, and Reactome) has been generated using the function Heatmap from the package ComplexHeatmap (ver.2.8.0), reported in Supplementary Fig. 6a. For each pathway, we calculated normalized enrichment scores (NES) of cancer-relevant gene sets by projecting the matrix of signed multi-omic feature weights onto hallmark pathway gene sets[70] using ssGSEA[71] available on https://github.com/broadinstitute/ssGSEA2.0 (parameters: gene.set.database = "h.all.v6.2.symbols.gmt", sample.norm.type = "rank", weight = 1, statistic = "area.under.RES", output.score.type = "NES", nperm = 1000, global.fdr = TRUE, min.overlap = 5 correl.type = "z.score"). Significantly (FDR <0.003) enriched Hallmark, KEGG and Reactome pathways are shown in Supplementary Fig. 6a.

**Estimation of stromal and immune scores.** ESTIMATE (ver.1.0.13)[18] was used to infer the presence of infiltrating stromal/immune cells in tumor tissues from RNA-seq gene expression in TNBC tumors in TCGA and METABRIC (Supplementary Data 2 and Fig. 1b).

**Single-cell RNAseq (scRNA) deconvolution analysis.** We performed cell type deconvolution using MuSiC (ver.0.2.0)[20], and six primary TNBC tumors[19]. For normal epithelial cells, normalized scRNA expression was retrieved from the publicly available data set (GSE113197)[21] and trained with the cell type composition for distinct epithelial cell populations (basal, luminal 1-1, luminal 1-2, basal myoepithelial, luminal-2). For primary TNBC tumors normalized scRNA expression data (GSE118390) was used to train cell type composition for individual cell types (epithelial, myoepithelial, stroma, monocyte, and lymphocyte). Reference scRNA for individual cell types were used to characterize cell-type proportions in bulk RNA-seq from TNBC tumors in TCGA.

**Deconvolution of immune cell-type composition.** The abundance of 64 different cell types, along with immune, stroma, and microenvironment scores, were computed via xCell (ver.1.1.0)[21]. For this analysis, normalized mRNA expression values were utilized, and therefore, this analysis was performed for 192 TCGA TNBC tumor samples (Fig. 1b), 22 CPTAC TNBC tumor samples, 74 MET500 TNBC tumor samples (Supplementary Fig. 4f) and 348 METABRIC samples (Supplementary Fig. 4f) with gene expression data. For the 22 CPTAC TNBC tumor samples (Fig. 2b) we also used the protein expression and gene-level phosphorylation data derived by taking the median of phosphorylation of the peptides for a given gene. Supplementary Data 2 contains the final cell type enrichment scores computed by xCell for all the samples.

**Mutation data.** Mutation calls for TCGA tumors were downloaded from TCGA Data Portal and imported into R (http://www.r-project.org) using TCGAbiolinks[62] (GDCquery, GDCdownload, and GDCprepare) and annotated consensus mutations (MAF file) obtained from the MC3 Working Group, and binned into high (>1.5mut/Mb) and low (<1.5mut/Mb) categories. Oncoprint plots were generated with Maftools (v2.8.0).

**Copy number variant calling.** Copy number variant (CNV) segment level data were obtained for 167 TCGA TNBC primary solid tumor tissue samples (57 BL1, 33 BL2, 31 LAR, and 46 M). Samples were downloaded and imported into R from GDC Data Portal using TCGAbiolinks (GDCquery, GDCdownload, and GDCprepare) using the following parameters: data.type = "Copy number

segmentation", platform = "Affymetrix SNP Array 6.0", file.type = "nocnv_hg19.seg", sample.type = "Primary solid Tumor". Copy number amplifications (>1) and deletions (<−0.7) are indicated for genes of interest.

We then used GISTIC2 (ver.2.0.22)[30] to independently identify genomic regions recurrently amplified or deleted in all TNBC samples or by each TNBC subtype (Supplementary Data 4). The top significantly (FDR < 0.25) amplified and deleted genes were plotted with ggplot2 (ver.2.2.1) using GISTIC q-values and genomic regions in Supplementary Fig. 5f. Known cancer oncogenes and tumor suppressors[72] were annotated (findOverlaps) within each broad peak using package GenomicRanges (ver.1.42.0)

As a measure of genomic instability, we evaluated the fraction of genome altered (FGA) calculated from segment level copy number data using the ratio of the sum of the lengths of all segments with signal above the threshold to the sum of all segment lengths[73]. FGA data ("brca_tcga_pan_can_atlas_2018_clinical_data.tsv") were obtained from cbioportal (www.cbioportal.org) and TNBC samples analyzed by subtype.

**Survival analysis.** We conducted a survival analysis of TNBC tumors from TCGA using Cox-proportional hazards model to study the association between overall survival (OS) and progression-free interval (PFI) within TNBC subtypes. For this analysis, the survival indices OS and PFI were retrieved from the latest publication[13]. Forrest plots were generated with ggforest and coxph functions from the R packages survminer (ver.0.4.9) and survival (ver.1.3.28), respectively. Forest plots were generated using the function ggforest from the R package survminer[74]. Proportional hazard ratios for the likelihood of a survival event for each TNBC subtype were adjusted for age and stage relative to all TNBC tumors ($n = 183$).

**Protein expression analysis**

*CPTAC protein expression data.* CPTAC protein expression data was downloaded from the LinkedOmics publicly available portal (http://www.linkedomics.org/data_download/CPTAC-BRCA/) for the 105 CPTAC BRCA prospective samples. Gene-level Log2ratio (normalized) proteomics data containing proteins ($N = 9733$) was used for downstream analysis[75].

*CPTAC phosphoproteomics data.* CPTAC phosphoproteomics data was downloaded from the LinkedOmics publicly available portal (http://www.linkedomics.org/data_download/CPTAC-BRCA/)) for the 105 CPTAC BRCA prospective samples. Peptide-level Log2ratio (normalized) phosphoproteomics data containing phosphosites ($N = 18806$) was used for downstream analysis.

*TCGA data.* The TCGA Protein expression dataset comprised of 192 TNBC cases was profiled for a panel of proteins (involving 281 protein features) by the antibody-based reverse-phase protein array (RPPA) platform. TCGA TNBC RPPA level 3 normalized data were downloaded from the GDC Data Portal using TCGAbiolinks functions GDCquery, GDCdownload, and GDCprepare in R software (http://www.r-project.org) for further analysis. We used function parameters platform = "MDA_RPPA_Core", data.type = "Protein expression quantification", and file.type = "expression" for these functions.

*TCGA DNA methylation data.* DNA methylation levels measured by Illumina HumanMethylation 450 (HM450) platform were available for 140 TCGA BRCA primary solid tumor (TP) tissue samples. Level 3 data were downloaded from TCGA Data Portal and imported into R software (http://www.r-project.org) using TCGA-biolinks functions GDCquery, GDCdownload and GDCprepare functions for further analysis[75]. DNA methylation level 3 data are β-values that were calculated from pre-processed raw data using the methylumi Bioconductor package[76]. Pre-processing steps included background correction, dye-bias normalization, and calculation of β-values and detection p-values. β-values range from zero to one, with zero indicating no DNA methylation and one indicating complete DNA methylation. A detection p-value compares the signal intensity difference between the analytical probes and a set of negative control probes on the array. Any data point with a corresponding p-value greater than 0.01 is deemed not statistically significantly different from the background and is thus masked as "NA" in TCGA level 3 data.

To identify genes with concordant gene expression and DNA methylation, average fold change for DNA methylation probes (<3 kb from TSS) were plotted against changes in gene expressions (fold changes, FC) by subtype (Fig. 4e). Pathway analysis (GSEA C2) was performed on genes with both hypermethylation (>1 log2 FC) and down-regulated gene expression (< 1 log2 FC) or genes with both hypomethylation (< 1 log2 FC) and up-regulated gene expression (> 1 log2 FC) (Fig. 4e, f).

*Enhancer linking by methylation/expression relationships (ELMER) analysis.* Using only TCGA primary tumor samples with both DNA methylation and gene expression ($n_{AllSubtypes} = 131$, stratified as the following TNBC subtypes: $n_{BL1} = 41$, $n_{BL2} = 30$, $n_{LAR} = 22$, $n_M = 34$) (Supplementary Data 7a), Enhancer Linking by Methylation/Expression Relationships analysis was performed using the R/Bioconductor ELMER package version 2.11.0[41]. For each analysis, we compared one

TNBC subtype vs. all the other remaining subtypes (e.g., BL1 vs BL2/LAR/M), in which we looked for distal probes that were differently methylated regulating any of the 10 upstream and 10 downstream genes. In detail, the following parameters were used: genome = "hg38", method = "supervised", get.diff.meth(sig.diff) = 0.2, get.diff.meth(p_value) = 0.01, get.diff.meth (minSubgroupFrac) = 0.2, get.-pair(Pe) = 10^−3, get.pair(raw.pvalue) = 10^−3, and get.pair(filter.probes) = TRUE, get.pair(permu.size) = 10,000, get.pair(minSubgroupFrac) = 0.4 (Supplementary Data 7b). To show the accessibility of some of those regions, ATAC-Seq TCGA-BRCA specific counts were downloaded from https://gdc.cancer.gov/about-data/publications/ATACseq-AWG and the subset of overlapping ATAC-Seq peaks/probes region (±250 bp) and predicted target genes were shown as a Heatmap in Supplementary Fig. 8d. (Supplementary Data 7c). The heatmap showing the anti-correlation between DNA methylation in distal regions and predicted target gene expression were created using ComplexHeatmap[77].

*Chromatin accessibility analysis.* ATAC-seq normalized bigWig track files for all the 18 TCGA/BRCA TNBC samples/technical replicates were downloaded GDC web portal (https://gdc.cancer.gov/about-data/publications/ATACseq-AWG). Genome browser screenshots of normalized ATAC-seq sequencing tracks of 18 different TNBC replicates for nine samples, shown across the same gene loci, were generated using UCSC Genome Browser (ver.376102).

*Tumor immune microenvironment (TIME) scoring.* H&E images for TCGA TNBC cases were accessed from (https://cancer.digitalslidearchive.org/) and immune cell spatial scoring of tumor microenvironment was performed by two pathologists (M.E.S and P.I.G-E) based on criteria from Gruosso et al. 2019. Tumors were classified as either: fully inflamed (FI), high TILs in tumor core stromal and intratumoral component (TILs ≥ 10%); stroma-restricted (SR), high TILs in tumor core limited to stromal component; margin-restricted (MR), low TILs in tumor core and inflammation limited to the tumor margin; or immune dessert (ID) low TILs in tumor core and tumor margin.

*Key oncogenic protein pathways and therapeutic opportunities.* We constructed an integrated multi-omic map involving four known oncogenic signaling pathways in TNBC. Specifically, we summarized the RNA, protein, and phospho-site levels per expression subtype. We characterized four pathways as: DNA Repair/Cell Cycle PI3K-MTOR signaling, MAPK, and antigen presentation pathway (Fig. 2b–e). For each gene annotated, in each pathway, we reported the logFC from differential testing analysis using CPTAC TNBC retrospective data. We have suggested specific inhibitors of up-regulated key events.

*In silico pharmacological and genetic dependency of TNBC cell lines.* Gene expression for breast cancer cell lines from DepMap (CCLE_depMap_18Q4_TPM_v2)[77], Genomics of Drug Sensitivity in Cancer (CCLE_DepMap_18q3_RNAseq_RPKM_20180718.gct)[35], and Breast Cancer PDTX Encyclopedia[36] were used to identify TNBC cell lines from the distribution of ER, PGR, and HER2 mRNA expression (Supplementary Fig. 7a). TNBC cell lines were then subtyped using the highest positive centroid correlation[65]. To identify-subtype specific genetic dependencies, Achilles gene effect scores and dependency scores were extracted for TNBC cell lines screened by RNAi (Combined Broad, Novartis, and Marcotte; Public 18Q4) and CRISPR (Avana; Public 18Q4). Linear-regression models of TNBC cell line correlation strength and viability were created for each subtype across all TNBC cell lines tested. T-statistic testing was used to evaluate association strength between subtype correlation strength and viability for each of the datasets. To identify subtype-specific pharmacological vulnerabilities, similar regression models were generated with subtype correlation strength and the area under the curve (AUC) from dose-response curves of cell lines in the GDSC or PDTX models treated with differing pharmacological inhibitors. In addition, for CCLE cell lines, DNA methylation (CCLE_RRBS_TSS1kb_20181022.txt.gz) and global chromatin modification of histone marks protein expression profiles (CCLE_GlobalChromatinProfiling_20181130.csv) were used to evaluate subtype differences among TNBC cell lines[65].

*Cell lines.* BT549 (ATCC, HTB-122), HCC1143 (ATCC, CRL-2321), HCC1937 (ATCC, CRL-2336), HCC1806(ATCC, CRL-2335), and 4T1 (ATCC, CRL-2539) were maintained in RPMI media. CAL-51 (DSMZ, ACC302,), CAL-120 cells (DSMZ, ACC459), HS578T (ATCC, HTB-126), MDAMB436 (ATCC, HTB-130), MDAMB468 (ATCC, HTB-132) and HDQP1 (DSMZ, ACC 494) were maintained in DMEM media supplemented with 5% fetal bovine serum. All cell lines were maintained in 100 U/mL penicillin and 100 μg/mL streptomycin (Gemini). Cell lines were authenticated by short-tandem repeat profiling (March, 2011) and were regularly tested and verified to be mycoplasma-negative by PCR analysis (Lonza). All cell lines were checked against the list of known misidentified cell lines maintained by the International Cell Line Authentication Committee (ICLAC).

*Cell viability assays.* TNBC cell lines were plated in quadruplicate at 2,000 cells/well into 96-well plates. Serial dilutions of tazmetostat (Selleck Chemicals, #S7128), CPI-1205 (Selleck Chemicals, #S8353), or MAK-683 (VWR, #B1972) were added to

wells after 24 h. Cell viability was determined after 4 days of treatment by measuring the fluorescence of metabolic reduction of AlamarBlue (ThermoFisher, #DAL1100), per the manufacturer's protocol. Average viability was determined relative to vehicle control from three independent experiments.

*Immunoblot.* TNBC cell lines were lysed in 100 mm plates 0, 1, 2, 3, or 5 days after treatment with a single 10 μM treatment of tazmetostat (Selleck Chemicals, #S7128), CPI-1205 (Selleck Chemicals, #S8353) or MAK-683 (VWR, #B1972). All cells were lysed in a RIPA buffer supplemented with protease and phosphatase inhibitors. Cell lysates (40 μg) were separated on polyacrylamide gels and transferred to polyvinyl difluoride membranes (Millipore). Immunoblotting was performed using anti-H3K27me3 (1:1000, Cell Signaling, 9733S), anti-histone 3 (1:5000, Abcam, ab1791), anti-HLA-A/B/C (1:1000, Santa Cruz, sc-52810), anti-GAPDH (1:5000, EMD Millapore, MAB374), and anti-vinculin (1:2000, Thermo-Fisher, 700062).

*Immunohistochemistry and quantification.* A tissue-microarray (TMA) including nine TNBC cell lines and tissue controls was used for IHC analyses. Formalin-fixed paraffin-embedded (FFPE) tissue sections were cut at 4 mm and deparaffinized. Antigen retrieval was performed with a citrate buffer (pH 6) in a decloaking chamber (Biocare). Endogen peroxidase blocked with 3% H2O2, protein block sol (Agilent), sections were then incubated with the primary HLA-A antibody (Santa Cruz Biotechnology, sc-365485; clone C6, dilution 1:300) overnight at 4 °C. Visualization system was Envision (Agilent), DAB as the chromogen (Agilent), and hematoxylin was applied as the counterstain. Whole sections were digitally acquired using an AxioScan Z1 slide scanner (Carl Zeiss) at 20x. Automated semiquantitative scoring was performed on exported scenes for each core by a pathologist blinded to the study hypothesis, using QuPath software (ver.0.3.0)[78]. Simple tissue detection was used to select cell line cores. Color deconvolution stains were set from ROIs for hematoxylin and DAB. Cell segmentation was determined on hematoxylin OD. Percentage of HLA-A+ cells over the total cell and per mm$^2$ were calculated with the positive cell detection algorithm according to the cell DAB OD mean. Each selected region was visually assessed for the correct performance of the quantification algorithm.

For Ki67, caspase-3, and H3K27me3 IHC, slides were placed on the Leica Bond Max IHC stainer. All steps besides dehydration, clearing, and coverslipping are performed on the Bond Max. Slides were deparaffinized and heat-induced antigen retrieval was performed using their Epitope Retrieval 2 solution for 20 min. Slides were placed in a Protein Block (Ref# x0909, DAKO, Carpinteria, CA) for 15 min. The sections were incubated with anti-Casp3 (1:300, Catalog # 9664S, Cell Signaling Technology, Danvers, MA), anti-Ki67 (1:250, Catalog #12202S, Cell Signaling Technology, Danvers, MA) or Tri-Methyl-Histone H3K27 (1:300, Catalog 9733, Cell Signaling Technology, Danvers, MA). The Bond Refine Polymer detection system was used for visualization.

*Flow cytometric analysis of MHC-I.* Flow cytometry was performed on an Attune NxT Flow Cytometer (Thermo Fisher). To visualize human MHC-I expression, human TNBC cell lines were stained with viability dye (Zombie Violet) and fluorophore-conjugated HLA-A/B/C (Biolegend, W6/32; A488, 5μl/million cells). To visualize murine MHC-I expression, mouse TNBC cell lines were stained with viability dye (Zombie Violet; Biolegend) and PE fluorophore-conjugated anti-H2-Kd (SF1-1.1, 1:1000, Biolegend). All antibodies were titrated to optimal concentrations prior to staining. Flow data were processed in R flowCore (ver.2.2.0) and flowStats (ver.4.2.0), corrected with compensation and plotted with R packages ggcyto (ver.1.18.0) and ggridges (ver. 0.5.3). An example of gating strategy located in Supplementary Fig. 13.

*RNA-sequencing.* TNBC cell lines were treated in triplicate with either DMSO or PRC2 inhibitors for five days and RNA was extracted using the Quick-RNA kit (Zymo Research, R1054). RNA concentration was quantified using a Qubit Fluorometer (Thermo FisherScientific). Library preparation was performed by BGI Americas and sequenced on the MGISEQ2000 using 100 bp paired-end chemistry at a depth >20M reads. Reads from RNA-Seq were mapped to reference genome GRCh38 using STAR (ver.2.5.0) aligner[79]. Raw counts were generated based on Ensembl gene models (GENCODE ver.21) with featureCounts (ver.1.5.0)[80].

*Chromatin Immunoprecipitation (ChIP).* ChIP-seq was performed using anti-H3K27me3 (Cell signaling #9733S) on 20 million CAL51, 10 million BT549 or 10 million CAL120 cells at 4 days post-treatment with 1μM TAZ. Cells were cross-linked with 1% formaldehyde for 8 min at room temperature and quenched with 125 mM Glycine, followed by an additional incubation for 5 min. Following cell lysis, chromatin was sonicated with a Bioruptor (Diagenode) to generate 200−500 bp chromatin fragments and immunoprecipitated with 5μl H3K27me3 antibody plus 30 μl Protein A magnetic beads (Invitrogen 10001D, 30 mg/ml), and incubated overnight at 4 °C. Beads were washed with low-salt buffer (20 mM Tris-HCl pH 8, 150 mM NaCl, 2 mM EDTA, 0.1% SDS, and 1% Triton X-100), high-salt buffer (20 mM Tris-HCl pH 8, 500 mM NaCl, 2 mM EDTA, 0.1% SDS, and 1% Triton X-100), LiCl wash buffer (10 mM Tris-HCl pH 8, 250mM LiCl, 1% NP-40, 1% sodium deoxycholate, and 1mM EDTA), and TE (pH 7.5), then re-suspended in de-cross slinking solution (50 mM Tris-HCl pH 8, 50mM EDTA, and 1% SDS).

DNA was purified with QIAquick PCR purification kit (Qiagen) according to the manufacturer's instructions and 150bp paired-end sequencing (Illumina NovaSeq) performed by Vanderbilt University Medical Center VANTAGE core.

*ChIP-seq data analysis.* Sequencing reads were aligned to the human genome (HG19) with bowtie (ver.1.2.2)[81] with filtering out duplicates. Broad peak calling was performed from aligned reads using MACS (ver.2.1.1) with the following options (macs2 callpeak --broad --gsize hs -B –SPMR)[82] ENCODE blacklist genomic regions were removed with bedtools (ver.2.30.0). Significant bound regions (false-discovery rate < 0.01) were filtered and annotated using the R package ChIPseeker (ver.1.28.3)[83]. ChIP-seq signal intensity plots over genomic regions were generated from reads per genomic content (RPGC) normalized bigwig files after input subtraction were generated using the function bamCompare from deepTools (ver.3.5.0) and genomic tracks were visualized using the IGV(ver.2.4.16) genome browser. Heatmaps were generated with input-subtracted bigwig files and significant peak bed file with the computeMatrix and plotHeatmap function from deepTools using the reference point ±3 kb distances upstream and downstream from the transcription start (TSS).

*Syngeneic murine xenograft.* For syngeneic xenograft tumors, $1 \times 10^5$ 4T1 cells in 200uL phosphate-buffered saline (PBS) were subcutaneously injected into the flanks of 6-week female BALB/c mice. Tumors were allowed to establish (50 mm$^3$), and mice were randomized to treatment arms. Mice were either treated with oral vehicle (2%DMSO, 0.5% sodium carboxymethyl cellulose, and 0.1% Tween-80) twice daily by orogastric gavage (o.g.) and intraperitoneal injection (i.p) of 200ul PBS twice per week, twice daily 250 mg/kg tazemetostat (o.g) in oral vehicle, twice per week intraperitoneal injections (i.p) of 10mg/kg paclitaxel (Hospira, NDC 61703-342-22) in PBS, or the combination of twice daily 250mg/kg tazemetostat (o.g) and twice weekly injections (i.p) of paclitaxel. Tumor sizes were measured by calipers and volume was calculated by the following formula: volume ($W^2 \times L$)/2. Mice were sacrificed after 4 weeks of treatment and tumors extracted, weighed, and fixed in 10% neutral-buffered formalin. Mice were housed and treated in accordance with NIH guidelines and protocols approved by the Institutional Animal Care and Use Committee at the Vanderbilt University Medical Center (IACUC; protocol M1900101-00).

*Quantification and statistical analysis.* R version 3.5.0 was used for all statistical analyses, unless specified otherwise. The statistical details of all experiments are reported in the figure legends and figures, including statistical analysis performed, statistical significance, and exact *n* values. The details of statistical analysis are presented within the text and the corresponding Method sections.

**Reporting summary.** Further information on research design is available in the Nature Research Reporting Summary linked to this article.

## Data availability

The raw RNA and ChIP sequencing files generated in this study are available at the NCBI Sequence Read Archive (SRA) under the accession PRJNA647007. The TCGA-omics publicly available data used in this study are available on the NIH Genomic Data Commons (https://gdc.cancer.gov) and the TCGA mutation publicly available data (mc3.v0.2.8.PUBLIC.maf.gz) is available at MC3 (https://api.gdc.cancer.gov/data/). The CPTAC publicly available protein datasets (CPTAC RNA, HS_CPTAC_BRCA_2018_RNA_GENE.cct.txt; CPTAC proteome, HS_CPTAC_BRCA_2018_Proteome_Ratio_Norm_gene_Median.cct.txt; CPTAC phosphoproteome, HS_CPTAC_BRCA_2018_Phosphoproteome_Ratio_Norm_Gene_median.cct.txt; CPTAC Clinical metadata, HS_CPTAC_BRCA_2018_CLI.tsi.txt) are available through the LinkedOmics data portal (http://www.linkedomics.org/data_download/CPTAC-BRCA/). The METABRIC publicly available data used in this study are available at the European Genome-Phenome Archive (http://www.ebi.ac.uk/ega/), which is hosted by the European Bioinformatics Institute, under accession number EGAS00000000083. Access can be obtained by contacting the EGA data access committee. ET500 publicly available sequencing data used in this study is available from dbGaP under accession number phs000673.v2.p1 (https://www.ncbi.nlm.nih.gov/projects/gap/cgi-bin/study.cgi?study_id=phs000673.v2.p1) and the MET500 web portal (http://met500.path.med.umich.edu). The single-cell RNA-seq publicly available datasets used in this study are available in Gene Expression Omnibus under accession code GSE113197 and GSE118390. The CCLE cell line gene expression publicly available data used in this study is available from the CCL (https://portals.broadinstitute.org/ccle). The DepMap cell line annotations (DepMap-2018q4-celllines.csv), RNAi (D2_combined_gene_dep_scores.csv), and CRISPR (CRISPR_gene_effect.csv) dependencies and gene expression (GDS_v17_TNBC_IC50.csv) publicly available data used in this study are available from the Broad DepMap (https://depmap.org/portal/download/). The Genomics of Drug Sensitivity in Cancer publicly available data used in the study are available in the GDS portal (https://www.cancerrxgene.org). The patient-derived tumor xenograft (PDTX) gene expression publicly available data used in the study are available from the Breast Cancer PDTX Encyclopedia (https://caldaslab.cruk.cam.ac.uk/bcape/). Full immunoblot scans are available in the source data. All other data supporting the findings of this study are available within the article, its supplementary information files, and the source data provided with this paper. Source data are provided with this paper.

## Code availability

All data, data acquisition, and analysis scripts are available at https://github.com/TransBioInfoLab/TNBC_analysis. The corresponding DOI is as follows: https://doi.org/10.5281/zenodo.5517463.

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

## Acknowledgements

This work was supported by the following grants: NIH grants, NCI R01CA200987, R01CA158472, P30CA068485, P50CA098131, and U24CA210954; Department of Defense BCRP grant BC201286 and, Susan G. Komen CCR13262005 and SAC110030. We thank the members of the scientific writing and editing for researchers at Vanderbilt (SWERV) core for manuscript editing.

## Author contributions

Conception and design, X.S.C. and B.D.L.; development of methodology, X.S.C., A.C., T.S., J.C., B.D.L. and L.W.; data acquisition; X.C., A.C., T.S., J.C., B.D.L., M.E.S., P.I.G.-E., J.L.J. and L.W.; analysis and interpretation: X.S.C., A.C., T.S., J.C., B.D.L., J.L.J., J.M.B., P.I.G-E, M.E.S, B.Z., Y.B., H.H. and L.W.; writing, review, and/or revision of the manuscript: X.S.C., A.C., T.S., J.C., B.D.L., J.A.P., B.Z., M.E.S., P.I.G.-E. and L.W.; study supervision: X.S.C. and B.D.L. All authors contributed to the interpretation of the results and read and approved the manuscript.

## Competing interests

X.C., B.D.L. and J.A.P. are inventors (PCT/US2012/065724) of intellectual property (TNBCtype) licensed by Insight Genetics Inc. J.M.B. receives research support from Genentech/Roche, Bristol Myers Squibb, and Incyte Corporation, has received consulting/expert witness fees from Novartis, and is an inventor on provisional patents regarding immunotherapy targets and biomarkers in cancer. The remaining authors declare no competing interests.

## Additional information

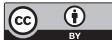

