## [Peer Review File · Nature Communications]

Multi-omics analysis identifies therapeutic vulnerabilities in triple-negative breast cancer subtypesREVIEWER COMMENTS

Reviewer #1, expert in proteogenomics and breast cancer (Remarks to the Author):

Lehmann et al present a potentially interesting manuscript to characterize triple negative breast cancer (TNBC) using publically available omics data. They go on and validate that PRC2 pharmacological inhibition can partially restore MHC-1 levels in subtype M cell lines. The interesting over-all question is if that is enough to elicit an immune response?

However, the reviewing of the content as well as interpretation and conclusions of the work is limited by some shortcomings presented below:

There is lots of polishing to do with the manuscript and this partly makes the reviewing the scientific content challenging:

In addition, the blurry quality of supplementary figures S1, S4, S5 makes it impossible to accurately review these results.

Since the majority of the paper revolves around data analysis of publicly available data it is important that the data analysis is clearly explained. The materials and methods section is lacking in many parts. To be transparent and provide a means for other researchers to reproduce the results, all scripts for the analyses should be included with the paper (or preferentially upload to github or similar). Since the authors use a lot of different data sources, the data that goes into the scripts should also be included so the analysis can be easily reproduced (or with alternative methods to load the data from within the scripts).

Example of above mentioned problem:

The authors present fig 1 as an overview of the TCGA data with RNA, RPPA and copy number alterations. It is unclear in the materials and methods how the different genes were selected for this heatmap. What is the FDR cutoff for inclusion? Other exclusion criteria? Same question goes for fig 2. Some genes that are mentioned in the text reads as they have different levels between subtypes, but do not look significant upon observing the quantitative pattern. Which genes are significantly up in which group? A boxplots of interesting genes is needed to visualize these.

Please use the standard color scheme for PAM50 in fig 1.

The color codes for alterations cannot be deduced by this reviewer. Please update, so the data can be interpreted.

The text needs read-through to check for consistency between text and figures. For example PTPRC, CD3E, KMT2A are mentioned in the text but I cannot find the data in figure 1b.

In this context, a lot of gene symbols are mentioned in the text which is good for the interested but takes up space and make it difficult to read. Consider cutting some text and gene symbols to make it easier to read.

Fig S1e is difficult to understand since it is blurry, but the p-values looks very borderline significant and I have difficulty to follow the reasoning in the result section due to this. Adjusted for multiple testing?

This analysis, as the one performed by Bareche et al, Annals of Oncology both use the TCGA dataset to characterize the TNBC subtypes, using the same online tool. How does your results compare to theirs? For example, they show high expression levels of EGFR, NOTCH1 and NOTCH3 in M subtype.

There is an ongoing discussion about the number of subtypes in TNBC. Why did the authors settle with 4 subtypes? The cancer literature suggests that the stroma surrounding the tumor is important for tumor cell response to therapy. Unbiased subtype grouping (for example by consensus clustering)

based on the TCGA and other data sets would provide a foundation for the validity of the number of different subtypes in TNBC and the continued characterization of the subtypes in the paper.

From text:

"To identify recurrent focal chromosomal copy number alterations (CNA) associated with each subtype, we applied GISTIC2.2 to TCGA TNBC tumors (Table S3). Overall, M-subtype tumors displayed the greatest degree of copy number alterations (Fig. S2f)."

M and BL1 seems to have similar amounts of CNAs according to fig S2f. What do you base this statement on?

From results section:

"To better understand the transcriptional pathways driving TNBC subtypes, we performed pathway gene set variation analysis (GSVA) analysis for differentially expressed genes across TNBC subtypes (Fig. 2a). The BL1 subtype displayed enrichment in cell cycle and DNA repair pathways."

To this reviewer, the majority of samples in BL1 does not seem to show enrichment in DNA repair genes. But in the text you discuss as they do. How did you determine which pathways to include in the plot? What are the statistical cutoffs used? Same questions go for fig 2b.

The dependency analysis in figure 3 is interesting. However, how the regression analysis was performed needs to be clarified in the m&m. For example, how is T actually calculated? Does it only reflect the slope?

How were the significance levels determined in figure 3, and thus how were genes and drugs selected for the figure? Same statistical cutoff for all 3 data types in the different panels? The display of T values across subtypes as in 3b is informative. Why did you use 3 different ways to display the data in 3b, c and d? It would be easier with one way to display the data for the reader. S4e is too blurry to read but consider changing its appearance also.

Genes and drugs are grouped into different pathways in fig 3. What is the overlap in pathway dependencies between the 3 different types of data? Do all different drugs that target the same gene have an effect in a subtype? An overview figure summarizing recurrent dependencies between the 3 data types would be useful.

Another suggestion would be to make a figure that relates the RNA/protein levels in TCGA to a dependency to identify potential biomarker candidates for a drug target. Now the results feel confusing and you have to manually look for potential biomarkers in for example fig 1.

Based on figure 4d, there seems to be a very weak correlation between methylation status and RNA expression. What additional value does the methylation status provide when RNA levels are closer to the phenotype? Is the enrichment of "EZH2 targets up" in M subtype also observed with only RNA data? Or is it because of the smaller number of genes with methylation status makes it significant? Why did you choose to validate drugs that target the PRC2 complex when you had all the drug dependency data that could be used to select candidates for further validation? Were the drugs targeting the PRC2 complex part of the screens, and if so, how was the response?

In fig 5, a panel of cell lines are used to compare the protein and RNA expression to M subtype. Which subtypes do the other cell lines reflect? Do they represent all the TNBC subtypes?

The concentrations of PRC2 complex inhibitors used in fig 5 are all over 1 μ M. The use of this high concentration suggests that the drugs are low affinity binders to their targets and very likely have multiple off target effects at the 10 μ M concentration used as standard treatment. What is known about the drug binding specificities?

From results:

"We identified 1663, 2048 and 1463 differentially regulated transcripts common to all inhibitors in cell lines CAL51, CAL120 and BT549, respectively (Fig. 5d)."

What do you mean by common? What is the overlap in transcript changes between the different cell lines?

I suggest that the authors use at least one person, with expertise in the field that have never read the paper to take substantial time to critically read it, check all statements made in it and provide feedback for updating the manuscript.

Proteogenomics is a relatively new term that have become a bite fashionable the last years. I do not fully agree on how the authors use the word proteogenomics. Please see the definition by Nesvizhskii 2014 Nat Methods.

The abbreviation ICI is commonly used for the drug fulvestrant in BC articles. Considering changing it.

From m&m:

"TNBC samples were identified by evaluating distribution of ER, PGR and HER2 using RNA, protein (RPPA/mass spec) and DNA copy number annotated with clinical assessment (IHC and FISH) provided by TCGA and CPTAC (Fig. S1). Genomic expression cutoffs were defined by separation of assigned pathology definitions and two lines of genomic evidence required to override a clinical call or infer status when no clinical information was available (Fig. S1a and S1B and Table S1)."

How were these cutoffs determined?

Further down in the same section:

"Similar methods were used to identify 348 primary TNBC patients from METABRIC and 45 metastatic TNBC patients from the MET500."

Similar does not mean the same, so how was it done?

Under m&m:

CPTAC phosphoproteomics data

"Similarly, gene-level phosphorylation data were derived by taking the median of phosphorylation levels of the peptides from a given gene, resulting in 4765 unique genes."

Does this mean that the median of all peptides belonging to one gene was used for the analysis? If so, why did you do it this way? Phosphorylation sites on a protein can have independent effects.

Similar statement later: For the 22 CPTAC TNBC tumor samples (Fig. 2B) we also used the protein expression and gene-level phosphorylation data derived by taking the median of phosphorylation of the peptides for a given gene.

Some refs are given as for example: Vasaikar et al. 2018 PMID: 29136207

How was the TNBC subtyping done and were there any filtering done? The text in m&m display some different numbers to those in table 1. What test did you use to look for significance in table 1?

In conclusion the manuscript is potentially interesting however the over all impact of the MHC-level and how that is validated could strengthen the paper by consolidating the main conclusions, as well as significant polishing.

Reviewer #3, expert in breast cancer subtypes and epigenetics (Remarks to the Author):

Lehman and al. integrate an impressive set of data to interrogate the complexity of TNBC biology. I have several comments regarding the methods and potential limitations of the study:

1/The entire study is based on bulk data analysis - the authors only use single-cell data to perform

deconvolution - which is in my view a strong limitation to appreciate the complexity of TNBC. Not only do we expect inter-tumoral heterogeneity, but also intra-tumor heterogeneity. A given tumor could display several subgroups of cells overlapping with several subtypes. Deconvolution is not sufficient to account for such complexity as signatures of some subtypes overlap. At least for expression studies, single-cell approaches (scRNA-seq) are mandatory to further dissect the complexity of TNBC.

2/ Regarding the integrative view of TNBC (Fig.1), sample order is forced to match known subgroups (BL1, BL2, M and LAR). I wondered why the authors have used this approach rather than a clustering-type of approach to discover co-occurring or exclusive features within this heterogeneous group of tumors. One objective of the study was to dive into the complexity of TNBC and reveal novel characteristics of these tumors - in this line it would be interesting to integrate data without a priori.

3/ Throughout the paper, a series of conclusions lack statistical testing, or at least no p-value is mentioned in the text even when the word 'significant' is used.

4/ Initially HLA genes are identified as displaying an anti-correlation between DNA methylation and RNA expression. Why did the authors use in a first instance agents targeting EZH2 rather than starting with agents targeting DNA methylation ?

5/ The study lacks H3K27me3 ChIP-seq analysis of some cell lines to show that indeed the re-expression of HLA genes after EZH2i treatment corresponds to a demethylation of H3K27me3 residues, i.e to 'epigenetic suppression' as indicated in the title of the manuscript.

Reviewer #4, expert in pharmacogenomics and in silico vulnerabilities (Remarks to the Author):

Lehmann et al provide important insights in potential therapeutic vulnerabilities for triple negative breast cancers (TNBCs) which represents a high clinical need. The strength of the paper is the combination of the classification of TNBCs in relation to the identification of targets for therapy by an integrated approach to link patient data to pharmacogenomic data.

In general the manuscript could improve by putting more focus on the strongest patterns that are observed and to leave all non-relevant relations, including most of the gene ontology analyses, out. The current version provides lists of gene ontologies and leaves the interpretation to the reader which is distracting. Initially, there is relative much emphasis on the classification based on the histology of TNBC tumors and explaining differences between these groups for expressed genes. It is not always clear why certain genes were selected and the patterns are not always convincing. The amount of detail in the text is, although clearly written, is distracting from the over-all message of the manuscript.

The not always convincing patterns as provided in Figure 1 become less convincing in Figure 2, also because there is insufficient explanation why certain gene ontology groups are preferred over others. Therefore, given this unclarity, these data might fit better in the supplementary information or, even better, should be selected only for the most convincing part which can be added to Figure 1. Figure 2C is quite difficult to understand because too much visual information (i.e. colors and classes) are provided in the figure which makes it is unclear how to figure should be interpreted. The gene ontology analysis of Figure 4e is also not convincing.

Most importantly, for the manuscript as a whole to become convincing, the therapeutic targets that are identified need evaluation in relevant in vivo models. An obvious focus point could be the therapeutics mentioned for the M-subgroup in Figure 3 in relation to the epigenetic drugs that induce MHC1 expression (Figure 5). These in vivo experiments should validate the findings and could be performed independently or as combinations to provide a rationale

for combination therapies. This validation is necessary because the depmap data might be flawed by noise. Since there are many MHC molecules, just the increased expression of MHC expression does not necessarily translate to increased immune activation. Therefore, to show that the results have a translational impact, this in vivo experiment is required.

Minor issues

The link of PRC2 to MHC has been made previously: PMID: 31564637; PMID: 30705065; PMID: 31562203, these references should be mentioned.

It could be informative to see how the histological classification relates to non-supervised clustering of the expression data and the mutation/cnv profiles, for instance by using K-means clustering. A non-supervised molecular classification could lead to a more refined classification of the tumors, although the mesenchymal subtype will probably still remain distinct given the difference in expression patterns in this subtype.

RESPONSE TO REFEREES

We appreciate the reviewer's time and thoughtful critique and have performed additional recommended experiments. As per the request of both reviewers 1 and 3 we have performed H3K27me3 ChIP-seq in three mesenchymal TNBC cell lines treated with EZH2 inhibitor tazemetostat, confirming a reduction in peaks near known EZH2 targets and MHC-I genes. We have included additional analysis of single cell RNA-seq data demonstrating intra-tumor heterogeneity and multiple subtype composition of bulk tumors. As suggested by all three reviewers, we have performed unbiased k-means consensus clustering to justify the use of four subtypes. Finally, we have added *in vivo* evaluation of the EZH2 inhibitor tazemetostat alone or in combination with paclitaxel chemotherapy in an immune competent syngeneic mouse xenograft model. These preclinical studies suggest a potential translational impact of adding an EZH2 inhibitor to chemotherapy in immune cold mesenchymal TNBC tumors. Please find a point-by-point rebuttal addressing each of the reviewers concerns below.

REVIEWER COMMENTS

Reviewer #1, expert in proteogenomics and breast cancer (Remarks to the Author):

Lehmann et al present a potentially interesting manuscript to characterize triple negative breast cancer (TNBC) using publicly available omics data. They go on and validate that PRC2 pharmacological inhibition can partially restore MHC-1 levels in subtype M cell lines. The interesting over-all question is if that is enough to elicit an immune response?

However, the reviewing of the content as well as interpretation and conclusions of the work is limited by some shortcomings presented below:

1. There is lots of polishing to do with the manuscript and this partly makes the reviewing the scientific content challenging:

In addition, the blurry quality of supplementary figures S1, S4, S5 makes it impossible to accurately review these results.

Authors' reply: We have performed significant polishing of the manuscript including independent editing by the scientific writing and editing for researchers at Vanderbilt (SWERV) core and hope these changes enhance the review of the manuscript. We also apologize for the poor quality of the supplemental figures during pdf conversion in the initial submission and have taken steps to ensure proper image resolution of all supplemental figures.

2. Since the majority of the paper revolves around data analysis of publicly available data it is important that the data analysis is clearly explained. The materials and methods section is lacking in many parts. To be transparent and provide a means for other researchers to reproduce the results, all scripts for the analyses should be included with the paper (or preferentially upload to github or similar). Since the authors use a lot of different data sources, the data that goes into the scripts should also be included so the analysis can be easily reproduced (or with alternative methods to load the data from within the scripts).

Authors' reply: We appreciate the reviewer's concern regarding the analysis and therefore we have made all of the data, data acquisition and analysis scripts available on github (https://github.com/TransBioInfoLab/TNBC_analysis) so that analysis can be easily reproduced. We have also modified the methods section to include more detail about identification of TNBC specimens in "Genomic-guided identification of TNBC specimens" (Page 37, Lines 964-967, 972-974 and 979-984), "TNBC subtype association testing for omics data" (Page 41, Lines 1055-1056), "RNA expression data analysis (Page 38, Lines 993-1004), "Copy number variant calling" (Page 43, Lines 115-1128) and "Single sample gene expression pathway analysis" (Page 42, Lines 1076-1077).

Example of above-mentioned problem: The authors present fig 1 as an overview of the TCGA data with RNA, RPPA and copy number alterations. It is unclear in the materials and methods how the different genes were selected for this heatmap. What is the FDR cutoff for inclusion? Other exclusion criteria? Same question goes for fig 2. Some genes that are mentioned in the text reads as they have different levels between subtypes, but do not look significant upon observing the quantitative pattern. Which genes are significantly up in which group? A boxplots of interesting genes is needed to visualize these.

The reviewer expressed concern over the criteria for the data displayed in Figure 1. The reviewer is correct that we included some non-significant genes in the pathway approach for the RPPA analysis in Figure 1. This was due to the limited number of proteins evaluated with this technology. However, we have taken the reviewer's suggestion and removed the RPPA data from Figure 1, as it does not significantly add to the overall conclusions of the manuscript. The genes selected for the gene expression heatmap were selected based on a biased curated list of known genes in antigen presentation, immune markers and immune checkpoint genes that were significantly (FDR p-value <1E-5) differentially expressed in the M subtype compared to other subtypes. Mutations were hand selected and grouped into similar pathways. Copy number amplifications and deletions were indicated when segment values were > 1 (amplification) or < -0.7 (deletion) and included for known oncogene and tumor suppressor genes. In Fig. 2 we selected genes with > 1FC and p-value<0.05 to performed unbiased gene ontology analysis and have now indicated this in the figure legend and methods. For Fig 2b-e, we performed a pathway analysis of several pathways and known activating phosphosites.

3. Please use the standard color scheme for PAM50 in fig 1.

Authors' reply: We appreciate the reviewer suggestions and we have modified the PAM50 colors to follow the same standard as other marker papers.

4. The color codes for alterations cannot be deduced by this reviewer. Please update, so the data can be interpreted.

Authors' reply: Thank you for pointing out this problem. The colors palette from the CNV/mutation alterations were altered to have visible and distinct colors.

5. The text needs read-through to check for consistency between text and figures. For example, PTPRC, CD3E, KMT2A are mentioned in the text but I cannot find the data in figure 1b. In this context, a lot of gene symbols are mentioned in the text which is good for the interested but takes up space and make it difficult to read. Consider cutting some text and gene symbols to make it easier to read.

Authors' reply: We thank the reviewer and have modified Fig. 1b to accurately match the text. We also have removed many gene symbols from the text to improve readability.

6. Fig S1e is difficult to understand since it is blurry, but the p-values looks very borderline significant and I have difficulty to follow the reasoning in the result section due to this. Adjusted for multiple testing?

Authors' reply: We apologize for the resolution issues and have updated the figures accordingly. We have modified the risk of recurrence analysis in the Forrest plot to adjust for multiple covariates of age and stage covariates. The new analysis is in Supplementary Fig. 1b, c.

7. This analysis, as the one performed by Bareche et al, Annals of Oncology both use the TCGA dataset to characterize the TNBC subtypes, using the same online tool. How does your results compare to theirs? For example, they show high expression levels of EGFR, NOTCH1 and NOTCH3 in M subtype.

Authors' reply: The reviewer is interested in the similarities with a prior analysis that evaluated gene expression and copy number changes in a combined METABRIC and TCGA cohort ¹. There were several similarities such as MYC amplification and overexpression in the M and BL1 subtypes, CCNE1 amplification and overexpression in the BL1 subtype, NOTCH3 amplification and overexpression in the M subtype. However, there are some important distinctions between the analyses, in which the Annals of Oncology manuscript included the IM and MSL subtypes and excluded the BL2 subtype and this analysis focused on BL1, BL2, M and LAR tumor intrinsic subtypes. We excluded the MSL and IM subtypes from the analysis as they likely reflect TNBC tumors of several subtypes with varying composition of normal stromal and immune cells within the tumors. In addition, we demonstrate tumors can be composed of several subtypes using scRNA data (see discussion from Reviewer 3 comment #1 and new Fig S3) and bulk tumors can be composed of multiple subtypes with differing correlation strength reflecting a subtype composition. Therefore another important distinction is in the methodology of this paper in which we performed a regression analysis associating the continuous variable for subtype correlation (i.e., estimated correlation score of each sample with centroid of each TNBC subtype) strength with genomic features (e.g., RNA-seq gene expressions, protein expressions, methylation levels) rather than treating subtypes as categories². Therefore, we have modified the discussion to include these comparisons (Page 23, Lines 579-582).

8. There is an ongoing discussion about the number of subtypes in TNBC. Why did the authors settle with 4 subtypes? The cancer literature suggests that the stroma surrounding the tumor is important for tumor cell response to therapy. Unbiased subtype grouping (for example by consensus clustering) based on the TCGA and other data sets would provide a foundation for the

validity of the number of different subtypes in TNBC and the continued characterization of the subtypes in the paper.

Authors' reply: We agree with the reviewer that several other investigators have performed independent analyses and identified 4-6 TNBC subtypes. We chose to focus on the four tumor intrinsic subtypes as the IM and MSL subtypes are likely tumors with high levels of immune and stroma. However, we have performed unbiased consensus clustering on the TCGA RNA-seq data and using the area under the CDF curve to determine that five clusters were the most optimal (Fig.1 below and Supplemental Fig. 2 in manuscript). Annotation of the clusters with both 4 and 6 TNBC subtypes along with correlation strength showed that these five clusters were composed of M-subtype (cluster 1) a mixture of BL1 and M (cluster 2), BL1 subtype (cluster 3), BL2 subtype (cluster 4) and a LAR subtype (cluster 5). Interestingly most of the IM subtype tumors were within the BL1 subtype, however they were also present in BL2 and LAR tumors with lower subtype correlations. These data support that the IM subtype is not a distinct subtype, but rather reflects

tumors of varying subtypes that include tumor infiltrating immune cells. While not tumor intrinsic, this classification likely identifies immune reactive tumors that have better prognosis and may be more amenable to immune checkpoint therapy, regardless of subtype. Cluster 3 that is composed of both M and BL1 tumors with correlations to both subtypes likely reflects a transition state between BL1 tumors that

Figure 1 (now Fig S2c): Consensus clustering of TNBC TCGA. Unbiased k-means consensus clustering was performed on the 192 TNBC samples. **a**, The empirical cumulative distribution function (CDF) plot shows the functions of the consensus matrix for each k^{1-10} . **b**, Relative change in area under the CDF curve comparing k and $k-1$. The number of clusters is decided when any further increase in cluster number (k) does not lead to a corresponding marked increase in the CDF area. **c**, Heatmap of TNBC samples similarity. Annotations show: 1) the continuous correlation with TNBCtype subtypes (BL1, BL2, M and LAR), 2) TNBC final assignment considering 4-class TNBC subtypes (BL1, BL2, M and LAR), and 6-class TNBC subtypes (BL1, BL2, M, LAR, MSL and IM) 3) TNBC Consensus clustering (CC) groups results from the unsupervised algorithm k-means ($k=5$) of the 5000 most variable genes.

are undergoing epithelial to mesenchymal transition. These data support the genomic similarities (mutation, copy number) between BL1 and M tumors, but differ in gene expression and global methylation patterns. Together with the scRNA data, these data suggest that binary subtyping may not accurately reflect the true tumor composition of individual cells of multiple subtypes and support the use of continuous modeling of subtypes using the correlation strength of each subtype. We have added additional text (Page 5-6 Lines 132-154).

9. From text: “To identify recurrent focal chromosomal copy number alterations (CNA) associated with each subtype, we applied GISTIC2 29 to TCGA TNBC tumors (Table S3). Overall, M-subtype tumors displayed the greatest degree of copy number alterations (Fig. S2f).” M and BL1 seems to have similar amounts of CNAs according to fig S2f. What do you base this statement on?

Authors’ reply: We thank the reviewer for pointing this out and the statement was only based on visual inspection of the GISTIC plots. Therefore, we performed an independent analysis of genomic instability using the fraction of the genome altered (FGA). FGA is calculated from segment level copy number data and reflects the ratio of the sum of the lengths of all segments with signal above the threshold to the sum of all segment lengths. We obtained the FGA data by downloading the clinical data file “brca_tcga_pan_can_atlas_2018_clinical_data.tsv” from www.cbioportal.org. FGA analysis showed the M-subtype had the highest median FGA followed by BL1, BL2 and LAR (Figure 2 below and Supplemental Fig. 5g in manuscript). Compared to all other subtypes, the M-subtype had significantly ($P=0.00046182$, unpaired T-test) greater FGA. These data are consistent with previously published data from Bareche Y et. al.³, in which their subtype analysis of the METABRIC dataset in which they showed (supplementary Figure S5D) the M and BL1 subtypes displayed significantly higher median chromosomal instability (CIN) scores, as defined by the percentage of the genome affected by CNAs. Therefore, we have added “Fraction Genome Altered” to Table S2, Supplemental Fig S5g and modified the text (Pages 43-44, Lines 1125-1129) as follows to include the prior observations of Bareche Y et. al.

Figure 2 (now Supplemental Fig. 5g): Mesenchymal TNBC subtype displays highest genomic instability. Boxplot shows a fraction of the genome altered (FGA) calculated from segment level copy number data stratified by subtype. Significance determined by unpaired T-test.

10. From results section: “To better understand the transcriptional pathways driving TNBC subtypes, we performed pathway gene set variation analysis (GSVA) analysis for differentially expressed genes across TNBC subtypes (Fig. 2a). The BL1 subtype displayed enrichment in cell cycle and DNA repair pathways.” To this reviewer, the majority of samples in BL1 does not seem to show enrichment in DNA repair genes. But in the text you discuss as the do. How did you determine which pathways to include in the plot? What are the statistical cutoffs used? Same questions goes for fig 2b.

Authors’ reply: The pathways included in Fig. 2a were selected from single-sample gene set enrichment analysis (ssGSEA) performed on each subtype compared to all others ($FDR < 0.05$, $NES > \text{abs } 1$). We have included a table of all of the testing results (Supplemental Table 3). The reviewer is correct that the DNA repair does not appear enriched in the Hallmark gene set, however there are several DNA repair pathways enriched in the BL1 subtype in the Reactome gene sets. We have modified Fig. 2a (now Supplemental Fig. S6a) to only reflect significant pathways. For Fig. 2A, we performed differential testing of protein expression between subtypes and have included and unbiased analysis of significant ($p\text{-value} < 0.05$ and $FC > 1$) for pathway

analysis. In this analysis, several DNA repair and cell cycle pathways are enriched in the BL1 subtype.

11. The dependency analysis in figure 3 is interesting. However, how the regression analysis was performed needs to be clarified in the m&m. For example, how is T actually calculated? Does it only reflect the slope?

Authors' reply: The T-value is the testing statistic for the regression coefficient between subtype strength and viability for each of the datasets. The T-value is the test statistic of the slope and naturally reflects the direction of the slope. To reduce confusion, we have modified Fig. 3a and removed "T-value" from the plots. We also have clarified how the regression analysis was performed in the results section (Pages 12-13 Lines 322-325).

12. How were the significance levels determined in figure 3, and thus how were genes and drugs selected for the figure? Same statistical cutoff for all 3 data types in the different panels? The display of T values across subtypes as in 3b is informative. Why did you use 3 different ways to display the data in 3b, c and d? It would be easier with one way to display the data for the reader. S4e is too blurry to read but consider changing its appearance also.

Authors' reply: For RNAi and CRISPR dependencies, genes were selected and curated into pathways from significant ($p < 0.05$) differential sensitivity (T-value < -1.5) for each subtype. We thank the reviewer for their appreciation of the T-value dependency and agree that displaying similar data with different methods may be more difficult for the reader. Therefore, we have modified Fig. 3b and c to display the results as heatmaps. However, we have displayed Fig. 3d differently as a balloon plot to stress the differences in the datasets, as Fig. 3b and c were performed in cell lines and 3d was generated from PDX explants. We agree with the reviewer that S4e is too blurry and have removed it entirely as it did not provide substantial information.

13. Genes and drugs are grouped into different pathways in fig 3. What is the overlap in pathway dependencies between the 3 different types of data? Do all different drugs that target the same gene have an effect in a subtype? An overview figure summarizing recurrent dependencies between the 3 data types would be useful. Another suggestion would be to make a figure that relates the RNA/protein levels in TCGA to a dependency to identify potential biomarker candidates for a drug target. Now the results feel confusing and you have to manually look for potential biomarker in for example fig 1.

Authors' reply: We agree with the reviewer that differentially displaying the data between data types could be cumbersome for the reader. Therefore, we have modified Fig. 3 b and c to a heatmap to show differential sensitivity to genetic and pharmacologic dependency between the subtypes. Lower T-values in blue indicate greater differential sensitivity to the genes/agents organized by pathway. The reviewer also recommends adding an overview figure summarizing recurrent dependencies and potential overlap with biomarkers identified in TCGA/CPTAC. Therefore, we have added a figure (Figure 3 below, Fig. 3e in manuscript) summarizing significant genomic alterations (mutation, CN, RNA, protein and phosphoprotein) that are associated with sensitivity to at least two dependency screens (genetic, pharmacologic or PDX screen). This

figure shows several potential biomarkers for the dependencies screen in BL1, BL2 and LAR subtypes. However, while we did observe recurrent dependencies on RAC1/CDC42 and RARA in the mesenchymal subtype, there were no genomic alterations identified in this subtype.

14. Based on figure 4d, there seems to be a very weak correlation between methylation status and RNA expression. What additional value does the methylation status provide when RNA levels

Figure 3 (now Fig. 3e). Recurrent subtype-specific dependencies and associated genomic alterations. Heatmap shows subtype-specific genomic alterations (yellow) in mutations, copy number, RNA protein and phosphoprotein levels. Subtype-specific (T-value <-1), genetic dependencies from RNAi and CRISPR screens are shown in purple. Pharmacological dependencies (T-value <-1) for indicated agents are shown for select inhibitors from cell line (light blue) and PDX explant (dark blue) screens.

is closer to the phenotype? Is the enrichment of “EZH2 targets up” in M subtype also observed with only RNA data? Or is it because of the smaller number of genes with methylation status makes it significant? Why did you chose to validate drugs that target the PRC2 complex when you had all the drug dependency data that could be used to select candidates for further validation? Were the drugs targeting the PRC2 complex part of the screens, and if so, how was the response?

Authors’ reply: The reviewer has concerns about the weak correlation between methylation status and RNA expression in Fig. 4d. However, Fig. 4d is not displaying correlation but rather log fold-change differences between each subtype vs. all other subtypes for each gene expressed and for fold change of methylation probes in near the promoters of corresponding genes. Therefore, we are just showing differential gene expression that correlates with methylation status within promoter regions of the genes on a subtype level.

The reviewer is also curious as to why EZH2 targets were not identified in the transcriptional analysis of the mesenchymal subtype and observed in the methylation analysis. This is not entirely unexpected as differences in the epigenome can be observed despite gene expression is largely unchanged⁴. Furthermore, we anticipate that this is possible because the vast number of genes regulated by the polycomb complex, including many transcription factors (i.e. HOX cluster and forkhead transcription factors) can result in diverse transcriptional changes that can be diluted across many pathways. This is clearly evident when we treat the mesenchymal TNBC cells with EZH2 inhibitors (Fig. 5d), in which we observe 1400-2000 differentially expressed genes. For example, in BT549 cells treated with EZH2i, we observed 2048 differentially expressed genes, of which 360 are transcription factors and of those 96 are HOX genes. Therefore, changes in EZH2 activity can have dramatic changes on transcription that are difficult to identify by transcriptional analyses alone.

The drug dependency screen data largely validated the pathway data (Fig. 2) and genetic dependency (Fig. 3B). The provide further evidence that the LAR subtype is dependent on the AR and PI3K pathway, The BL1 subtype dependent on cell cycle/DNA repair and sensitive to chemotherapy/cell cycle targeted therapies and the BL2 dependent on MAPK. However, the majority of the genetic dependencies specific to the M subtype involved transcription factors, adhesion/motility and epigenetic modifiers (Fig. 3B) and were not validated in the pharmacological screens due to a lack of drugs targeting these proteins. Since identifying agents that can target the immune cold M subtype is of greatest need, we evaluated the epigenetic landscape (Fig. 4) and uncovered a selective repression of immune and EZH2 target genes in the methylation data. Since EZH2 inhibitors were not included in either of the GDSC or PDX drug screens, we evaluated several EZH2 inhibitors across TNBC cell lines.

15. In fig 5, a panel of cell lines are used to compare the protein and RNA expression to M subtype. Which subtypes do the other cell lines reflect? Do they represent all the TNBC subtypes?

Authors' reply: In Figure 5 we included three (HCC1937, HCC1143 and MDA-MB-468) and two BL2 (HCC1806 and MDA-MB-436) cell lines for comparison. Since, we identified and validated many potential therapeutic targets (AR, PI3K, AKT and ERBB2 inhibitors) for the LAR subtype in Fig. 3, we chose to focus on the mesenchymal subtype compared to the other basal subtypes in the remaining analysis.

16. The concentrations of PRC2 complex inhibitors used in fig 5 are all over 1 μ M. The use of this high concentrations suggest that the drugs are low affinity binders to their targets and very likely have multiple off target effects at the 10 μ M concentration used as standard treatment. What is known about the drugs binding specificities?

Authors' reply: The reviewer raises the possibility of off target effects at the 10uM concentration. Tazemetostat is a highly specific EZH2 inhibitor with K_i of 2.5 nM in cell-free systems. Similarly, CPI-1205 is highly specific for EZH2 (2nM) and MAK683 inhibits EED at 59nM. In the literature, cell culture treatments with tazemetostat range from 1uM, 2.7uM⁵ to 10uM⁶. In the Phase I study⁷, the median plasma tazemetostat concentration was 100 ng/mL with the recommended phase 2 dose (800 mg twice daily), which converts to plasma levels of 174uM. Since cells were being treated for up to seven days, and did not substantially alter viability, we decided to use a higher

Figure 4. (now Supplemental Fig. 11a) PRC2 inhibitors decrease H3K27me3 and increase MHC-I protein levels.

Immunoblots show relative H3K27me3, total-H3, MHC-I and GAPDH levels for the indicated cell lines at day 0, 1, 3, 5, 7 after a single dose of 1uM of either tazemetostat, CPI-1205 or MAK-683.

single dose of the EZH2 inhibitors. Given that we used three structurally different inhibitors targeting two separate components of the PRC2 complex and observed similar decreases in H3K27me3 and increases in MHC-I, it is unlikely that these changes can be attributed to off target

effects. However, to mitigate the reviewers concerns for off target effects, we have repeated immunoblot analysis of Fig 5h using 1uM tazemetostat, CPI-1205 and MAK683. Treatment with a single 1uM dose decreased H3K27me3 at days 3 -7, therefore the resulting increase in MHC-I could be attributed to PRC2 inhibition (below). These results are now included in Fig. S11a to accompany the additional ChIP-seq experiments at this concentration in Fig. 6.

17. From results: “We identified 1663, 2048 and 1463 differentially regulated transcripts common to all inhibitors in cell lines CAL51, CAL120 and BT549, respectively (Fig. 5d).” What do you mean by common? What is the overlap in transcript changes between the different cell lines?

Authors’ reply: We apologize for the confusion. In this case “common” is referring to all the significantly differentially expressed transcripts in the 9 PRC2 inhibitor treated cells (3 for each tazemetostat, CPI1205 and MAK683) compared to the 3 DMSO treated samples for each cell line. Therefore, in each of the heatmaps in Fig. 5d there are 1663 for CAL51, 2048 for CAL120 and 1463 for BT549. There is substantial overlap in transcript changes between the cell lines, and can be observed in the Venn diagram below in which 275 genes are common to all of the cell lines (Fig. S9e). Furthermore 50.3% (BT549), 56.2% (CAL120) and 56.1% (CAL51) of the differentially expressed genes are shared with at least one other cell line. We therefore included a Venn diagram of the overlap (Fig. S9e).

Figure 5. (now in Supplemental Fig. 9e): Overlap of genes with increased expression after PRC2 inhibition between TNBC cell lines. Venn diagram shows the overlap of significantly differentially expressed transcripts to all three PRC2 inhibitors from the heatmaps in Fig. 5D

18. I suggest that the authors use at least one person, with expertise in the field that have never read the paper to take substantial time to critically read it, check all statements made in it and provide feedback for updating the manuscript.

Authors’ reply: We thank the review for the suggestion and have had members of the scientific writing and editing for researchers at Vanderbilt (SWERV) core critically edit the manuscript.

19. Proteogenomics is a relatively new term that have become a bite fashionable the last years. I do not fully agree on how the authors use the word proteogenomics. Please see the definition by Nesvizhskii 2014 Nat Methods.

Authors’ reply: We agree that one interpretation of “proteogenomics” by Nesvizhskii et. al defines proteogenomic as the use sequencing and transcriptomics (RNA-Seq, ribosome profiling) data to generate customized protein sequence databases to help interpret proteomics (LC-MS/MS) data. However, increasingly the term “proteogenomics” is used to describe an approach using the intersection/convergence of proteomics and genomics, such as the recent CPTAC breast manuscript, entitled, “Proteogenomic Landscape of Breast Cancer Tumorigenesis and Targeted Therapy”⁸.

20. The abbreviation ICI is commonly used for the drug fulvestrant in BC articles. Considering changing it.

Authors' reply: We agree with the reviewer that the ICI abbreviation could be confused with the chemical name for fulvestrant (ICI 182780). Therefore, we have removed the abbreviation and spelled out “immune checkpoint inhibition” throughout the manuscript.

21. From m&m: TNBC samples were identified by evaluating distribution of ER, PGR and HER2 using RNA, protein (RPPA/mass spec) and DNA copy number annotated with clinical assessment (IHC and FISH) provided by TCGA and CPTAC (Fig. S1). Genomic expression cutoffs were defined by separation of assigned pathology definitions and two lines of genomic evidence required to override a clinical call or infer status when no clinical information was available (Fig. S1a and S1B and Table S1).” How were these cutoffs determined? Further down in the same section: “Similar methods were used to identify 348 primary TNBC patients from METABRIC and 45 metastatic TNBC patients from the MET500.” Similar does not mean the same, so how was it done?

Authors' reply: The empirical cutoffs were manually defined from the bimodal distribution of genomic data (mRNA, protein and copy number for HER2) available for each dataset guided by clinical annotations (IHC for all markers and FISH for HER2) when available. The distributions are shown in Supplemental Fig. 1a for TCGA, Supplemental Fig. 4 a and b for METABRIC and MET500, and Supplemental Fig. 6b for CPTAC. For TCGA and CPTAC, RNA-seq mRNA expression and RPPA data were used to define ER and PR cutoffs, while ERBB2 copy number data was also included to define HER2 cutoffs. For METABRIC and MET500 BRCA samples, mRNA expression was used to define cutoffs for hormone status. The distribution of samples among the subtypes were similar in TCGA, CPTAC and METABRIC, while there was an enrichment of TNBC tumors in the metastatic MET500 dataset, consistent with increased metastatic spread observed with TNBC. We have modified the methods section (Pages 37-38, Lines 971-974, 979-981 and 987-991), “Genomic-guided identification of TNBC specimens” for additional clarity and have specifically described how the analysis were performed in METABRIC and MET500.

22. Under m&m: CPTAC phosphoproteomics data “Similarly, gene-level phosphorylation data were derived by taking the median of phosphorylation levels of the peptides from a given gene, resulting in 4765 unique genes.” Does this mean that the median of all peptides belonging to one gene was used for the analysis? If so, why did you do it this way? Phosphorylation sites on a protein can have independent effects. Similar statement later: For the 22 CPTAC TNBC tumor samples (Fig. 2B) we also used the protein expression and gene-level phosphorylation data derived by taking the median of phosphorylation of the peptides for a given gene. Some refs are given as for example: Vasaikar et al. 2018 PMID: 29136207

Authors' reply: The reviewer is correct that we had summarized the phospho-protein data to the median of all peptides and that phosphorylation of different peptides can have independent effects. Therefore, we have focused on displaying only those phospho-peptides that have known activating/deactivating phosphorylation events. For example, for the PI3K pathway we have

chosen to show activating AKT1/2 (S473, T308) peptides, AKT substrates [GSK3B (S21 and S9), AKT1S1 (T266), BAD (S134) and FOXO3 (S294)], mTOR (S2478 and S2481) and downstream RPS6 (S236 and S240) or EIF4EBP1 (S35 and S65). We have modified Fig. 2b to show a selected analysis of several key pathways and canonical phosphorylated residues.

23. How was the TNBC subtyping done and were there any filtering done? The text in m&m display some different numbers to those in table 1. What test did you use to look for significance in table 1?

Authors' reply: First, TNBC samples were identified from each breast cancer dataset as described in materials and methods section, "Genomic-guided identification of TNBC specimens", using known clinical annotations for ER, PR and HER2. Empirical cutoffs for hormone receptor status and ERBB2 amplification were manually defined from the bimodal distribution of genomic data and distribution of known clinical annotations as shown in Supplemental Fig 1a and 6b. This resulted in 192 (TCGA), 27 (CPTAC) and 348 (METABRIC) TNBC tumors identified. Then TNBC subtyping was performed on normalized mRNA expression of only the TNBC tumors for each dataset using the web-based centroid correlations detailed in the "methods" section entitled "TNBC subtyping". For the TCGA, nine samples were deemed unclassified due to low subtype correlation and were removed from further analysis, thus leaving 183 tumors for subtype analysis of clinical attributes in table 1. Chi-squared testing was used to determine statistical significance of clinical variables in table 1. We have modified m&m and table legend accordingly (Page 41, Line 1069 and legend).

24. In conclusion the manuscript is potentially interesting however the over all impact of the MHC-level and how that is validated could strengthen the paper by consolidating the main conclusions, as well as significant polishing.

Authors' reply: We have performed significant polishing of the manuscript including independent editing by the scientific writing and editing for researchers at Vanderbilt (SWERV) core. We have also added H3K27me3 ChIP-seq demonstrating PRC2 modifications in MHC1 genes. Furthermore, we have added additional in vivo xenograft experiments in syngeneic mouse models (see response to Reviewer #4, comment 2 for detailed explanation of additional experiments).

Reviewer #3, expert in breast cancer subtypes and epigenetics (Remarks to the Author):

Lehman and al. integrate an impressive set of data to interrogate the complexity of TNBC biology. I have several comments regarding the methods and potential limitations of the study:

1/The entire study is based on bulk data analysis - the authors only use single-cell data to perform deconvolution - which is in my view a strong limitation to appreciate the complexity of TNBC. Not only do we expect inter-tumoral heterogeneity, but also intra-tumor heterogeneity. A given tumor could display several subgroups of cells overlapping with several subtypes. Deconvolution is not sufficient to account for such complexity as signatures of some subtypes overlap. At least for

expression studies, single-cell approaches (scRNA-seq) are mandatory to further dissect the complexity of TNBC.

Authors' reply: The reviewer is absolutely correct in that a major limitation of the study is that the data was obtained from bulk analysis and does not reflect the intra-tumor heterogeneity. While this was the focus of a future manuscript, we decided to include additional analysis of existing

Figure 6 (now Supplemental Fig. 3b and c). Analysis on scRNA reveals intra-tumor TNBC heterogeneity. a, scRNA count data were obtained from GSE118390⁹ and normalized in Seruat. b, Umap plots show distinct cell populations colored by the six individual TNBC tumors. Common cell markers were used to identify stroma, monocyte, epithelial, myoepithelial, endothelial and lymphocyte populations. c, Epithelial cells were extracted and renormalized and TNBC subtyped. d, Individual U-map plots show the distribution and subtype calls for individual cells from patient 39 and patient 81. Colorbars above e and f show subtype correlation strength for each subtype of the pseudobulk composite analysis of the integrated expression of all cells for each patient tumor.

single-cell RNA (scRNA) sequencing from TNBC tumors to strengthen the rationale for the methodology used throughout the manuscript. While the samples were binned into four subtypes in figures, all of the differential testing were performed on the correlation strength to each of these subtypes across all samples and therefore we are accounting for tumors correlating to multiple subtypes. To demonstrate the intra-tumor heterogeneity, we processed publicly available scRNA sequencing from six TNBC tumors published⁹. (Figure 6a below and Supplemental Fig. 3a in manuscript). We analyzed all cells with known markers to identify only epithelial cells (Figure 6b). Using only epithelial cells we performed subtyping on individual cells and each tumor by composing a pseudobulk tumor from the integrated values of all cells (Figure 6d). Individual cells within tumors displayed differing subtype composition that was reflected in the pseudobulk analysis correlating to multiple subtypes (Figure 6d. top colorbar) such as PT039 correlating to both BL1 and M subtypes. Patients with pseudobulk analysis reflecting a purer subtype composition such as PT81 were primarily composed of individual BL1 cells (Figure 6d). These data demonstrate that bulk tumors correlating to multiple subtypes likely reflect tumors with greater intratumor heterogeneity and all subtype correlations should be considered when analyzing bulk tumors. We have modified the results accordingly (Page 5, Lines 144-155)

2/ Regarding the integrative view of TNBC (Fig.1), sample order is forced to match known subgroups (BL1, BL2, M and LAR). I wondered why the authors have used this approach rather than a clustering-type of approach to discover co-occurring or exclusive features within this

heterogeneous group of tumors. One objective of the study was to dive into the complexity of TNBC and reveal novel characteristics of these tumors - in this line it would be interesting to integrate data without a priori.

Authors' reply: Reviewer 1 also had similar concerns over subtype determination and therefore we have performed unbiased k-means consensus clustering to derive the rationale for the four subtypes analyzed throughout the manuscript. Please see the Authors' reply to Reviewer #1 comment #8 for detailed explanation. We have also added the analysis as a new Supplemental Fig. S2 and modified the text accordingly. While the samples in Fig.1 were binned by categorical subtype, they were arranged from strongest to lowest correlating sample within each bin.

3/ Throughout the paper, a series of conclusions lack statistical testing, or at least no p-value is mentioned in the text even when the word 'significant' is used.

Authors' reply: We thank the reviewer for identifying this issue and we have performed a manual search to identify any instance where "significant" was used without providing a statistical test and p-value. We included p-values and modified the manuscript accordingly. Below are the modifications that appear in the results section.

(Page 4, Line 114) "The most common histology was invasive ductal carcinoma, however special histological subtypes were significantly enriched in individual subtypes, with medullary carcinomas in BL1 tumors ($p=0.0041$, chi-squared), malignant phyllodes tumors in the M-subtype ($p=0.0026$, chi-squared) and metaplastic carcinomas in the BL2 subtype ($p=0.011$, chi-squared) (Table 1)."

(Page 7, Lines 183-185) "TNBC patients with a higher tumor mutational burden ($>1.5\text{mut/Mb}$) had significantly ($p=0.017$, log-rank) better progression-free interval (PFI) (Fig. S4a)"

(Page 9, Lines 238-240) "While amplifications in PD-L1 occurred across all subtypes, deletions in beta-2-microglobulin (*B2M*) were more frequent (17.8% vs. 3.7%) in the M-subtype compared to other subtypes ($p=0.0061$, Fisher's exact)"

4/ Initially HLA genes are identified as displaying an anti-correlation between DNA methylation and RNA expression. Why did the authors use in a first instance agents targeting EZH2 rather than starting with agents targeting DNA methylation ?

Authors' reply: The reviewer brings up a logical next step of evaluating whether DNA methylation inhibitors could be used to restore MHC-I expression. However, the reason PRC2/EZH2 inhibitors were chosen is based on the observation that EZH2 targets were also methylated in addition to antigen presentation genes. Furthermore, when we performed *in silico* analysis (GSE57343) of 11 TNBC cell lines treated with the demethylating agent 5-azacitidine (5-aza) and observed no consistent increases in MHC-I gene expression at 1, 3, 7 or 10 days¹⁰ (Figure 7, below).

Figure 7. DNA methylation inhibition does not impact MHC-I expression in TNBC cell lines. Heatmap shows relative expression (treated/mock) of HLA-A/B/C in 11 different TNBC cell line models at days 1,3,7 and 10 after treatment with the demethylating agent 5-azacitidine (5-aza).

5/ The study lacks H3K27me3 ChIP-seq analysis of some cell lines to show that indeed the re-expression of HLA genes after EZH2i treatment corresponds to a demethylation of H3K27me3 residues, i.e to 'epigenetic suppression' as indicated in the title of the manuscript.

Authors' reply: The reviewer brings up a good point, also raised by another reviewer and the editor that additional ChIP-seq experiments for H3K27me3 would be necessary to indeed demonstrate that the elevated HLA expression is directly mediated by polycomb inhibition. Therefore, we have performed H3K27me3 ChIP-seq in the three mesenchymal cell lines treated with either DMSO or 1uM of the EZH2 inhibitor tazemetostat. There was a substantial reduction in the number of significantly decreased H3K27me3 peaks with tazemetostat treatment (Figure 8 below, Figure 6 in manuscript). Furthermore, peak intensity within promoter regions of known H3K27me3 targets (BENPORATH_ES_WITH_H3K27ME3) or EZH2 targets (NUYTEN_EZH2_TARGETS_UP) decreased substantially. We then examined how the RNA expression levels changed in relation to decreased peak intensity within promoter regions of genes after EZH2 inhibition (Figure 8b). A substantial portion of HOX genes (known to be repressed by the PRC2) in addition to HLA-A displayed increased in gene expression and decreased H3K27me3 intensity at promoters with EZH2 inhibition, demonstrating that expression of these genes was mediated by a loss of modified H3K27me3 deposition by PRC2.

Figure 8 (now Fig. 6): EZH2 inhibition decreases global H3K27me3 and repressive marks at MHC-I locus. a, Profile plot and heatmap for H3K27me3 ChIP-seq signal for differential peak, H3K27me3 targets (BENPORATH_ES_WITH_H3K27ME3), or EZH2 targets (NUYTEN_EZH2_TARGETS_UP) centered on transcriptional-start sites (TSS) for BT549, CAL-120 and CAL-51 cells treated for 4 days with either DMSO or 1 μ M tazemetostat. Sequencing reads normalized to reads per genomic content. **b**, Scatterplots show differential RNA expression (Log₂ FC, FDR <0.05) and differential H3K27me3 promoter occupancy (FDR <0.05) in tazemetostat treated cells relative to DMSO treatment.

Reviewer #4, expert in pharmacogenomics and in silico vulnerabilities (Remarks to the Author): Lehmann et al provide important insights in potential therapeutic vulnerabilities for triple negative breast cancers (TNBCs) which represents a high clinical need. The strength of the paper is the combination of the classification of TNBCs in relation to the identification of targets for therapy by an integrated approach to link patient data to pharmacogenomic data.

In general the manuscript could improve by putting more focus on the strongest patterns that are observed and to leave all non-relevant relations, including most of the gene ontology analyses, out. The current version provides lists of gene ontologies and leaves the interpretation to the reader which is distracting. Initially, there is relative much emphasis on the classification based on the histology of TNBC tumors and explaining differences between these groups for expressed genes. It is not always clear why certain genes were selected and the patterns are not always convincing. The amount of detail in the text is, although clearly written, is distracting from the overall message of the manuscript.

Authors' reply: We thank the reviewer for their honest, insightful comments and appreciate the recognized strength of the integrated approach to link patient data to pharmacogenetic data. We have substantially revised the manuscript with additional experimentation and improved the text by removing non-relevant distracting information.

1. The not always convincing patterns as provided in Figure 1 become less convincing in Figure 2, also because there is insufficient explanation why certain gene ontology groups are preferred over others. Therefore, given this unclarity, these data might fit better in the supplementary information or, even better, should be selected only for the most convincing part which can be added to Figure 1. Figure 2C is quite difficult to understand because too much visual information (i.e. colors and classes) are provided in the figure which makes it is unclear how to figure should be interpreted. The gene ontology analysis of Figure 4e is also not convincing.

Authors' reply: As per the reviewer's recommendation, we have removed the unconvincing RPA data from Fig. 1 and moved the ssGSEA from Fig 2a. into the supplementary information (Supplemental S6a). In addition, we have only included significantly enriched pathways (FDR<0.003) and have clustered the samples by pathway to show similarly enriched pathways within subtypes. We apologize for the difficulty in understanding Figure 2b and agree that there is substantial visual information. Since we moved Fig 2a to the supplemental, this has allowed us to expand on Figure 2b and include only those phosphorylated residues known to play a role in each pathway. We agree with the reviewer that the gene ontology in Figure 4e is cumbersome, and have removed the other subtypes to supplemental to focus only the mesenchymal subtype.

2. Most importantly, for the manuscript as a whole to become convincing, the therapeutic targets that are identified need evaluation in relevant in vivo models. An obvious focus point could be the therapeutics mentioned for the M-subgroup in Figure 3 in relation to the epigenetic drugs that induce MHCI expression (Figure 5). These in vivo experiments should validate the findings and could be performed independently or as

combinations to provide a rationale for combination therapies. This validation is necessary because the depmap data might be flawed by noise. Since there are many MHC molecules, just the increased expression of MHC expression does not necessarily translate to increased immune activation. Therefore, to show that the results have a translational impact, this *in vivo* experiment is required.

Authors' reply: The reviewer makes an important point about the evaluation of the novel findings *in vivo* models and the disconnect with the agents evaluated in Figure 3. As requested per another reviewer, we have added a new summary Fig. 3e, in which differing genomic alterations are paired with recurrent, genetic and pharmacological dependencies. In each subtype we were able to identify genomic alterations that may be biomarkers for these dependencies with the exception on the M subtype for which there were few effective therapeutic and no markers. Therefore, the remainder of the manuscript was focused on identifying potential targets for the M-subtype, that is characterized by an absence of immune cells and antigen presentation. We uncovered that inhibition of PRC2 specifically in this subtype was able to restore MHC-I expression in M-subtype TNBC cells.

The reviewer recommends that additional *in vivo* experiments are necessary to demonstrate the translational impact of EZH2 inhibition in TNBC. Therefore, we have performed tumor xenograft studies with the murine syngeneic TNBC model 4T1 in immune competent mice. Treatment of 4T1 cells with PRC2 inhibitors were able to decrease H3K27me3 (Figure 9a below and Fig. 7 in manuscript) and increase cell surface MHC-I expression two-fold by flow cytometry (Figure 9b

Figure 9 (now Fig. 7): EZH2 inhibition increases the efficacy of paclitaxel chemotherapy in a syngeneic murine TNBC model. a, Immunoblots show H3K27me3 and MHC-I protein expression at 1, 3, 5, 7 days after a single 10uM treatment with either tazemetostat, CPI-1205 or MAK-683. b, Histograms show distribution and c, quantification of cell-surface MHC-I protein expression 5 days after a 10uM treatment with the indicated PRC2 inhibitors. Results are representative of three experiments. d, Treatment schedule for mice bearing syngeneic 4T1 xenograft tumors. Mice were treated with vehicle, twice daily (BID) with 250 mg/kg tazemetostat, twice a week with 10 mg/kg paclitaxel or the combination of tazemetostat and paclitaxel. Results are representative of ten tumors. e, Graphs show 4T1 tumor volume (mm³) across time of mice treated with vehicle, tazemetostat, paclitaxel or the combination. Error bars represent standard error of the mean. f, Barplot shows distribution of final tumor weight (mg) from mice treated with vehicle, tazemetostat (TAZ), paclitaxel (TAX) or the combination (TAZ + TAX). g, Plot shows IHC quantification of intratumor CD3+ T-cells in 4T1 xenograft tumors by treatment group.

and c). We established 4T1 xenografts in immune competent syngeneic Balb/c mice and treated mice with either vehicle or the the most clinically advanced of the EZH2 inhibitors, tazemetostat, alone or in combination with paclitaxel chemotherapy (Figure 9d). Single agent tazemetostat had minimal and similar effects on tumor growth/weight as paclitaxel, however the combination was more effective than each agent alone (Figure 9 e and f). Treatment with tazemetostat appeared to

increase the levels on intratumor T-cells (Figure 9 g). We have modified the results (Page 20, Lines 507-523) and included the data as Fig. 7 in the manuscript.

Minor issues

3. The link of PRC2 to MHC has been made previously: PMID: 31564637; PMID: 30705065; PMID: 31562203, these references should be mentioned.

Authors’ reply: We thank the reviewer for their suggestion and have acknowledged these studies in the discussion (Page 24, Lines 612-614). “Several recent studies have also demonstrated EZH2 inhibition can enhance tumor cell antigen presentation in head and neck squamous cell carcinoma ¹¹, diffuse large B-cell lymphoma ¹² and melanoma ¹³.”

4. It could be informative to see how the histological classification relates to non-supervised clustering of the expression data and the mutation/cnv profiles, for instance by using K-means clustering. A non-supervised molecular classification could lead to a more refined classification of the tumors, although the mesenchymal subtype will probably still remain distinct given the difference in expression patterns in this subtype.

Authors’ reply: Both Reviewers 1 and 3 had similar concerns regarding subtype determination and therefore we have performed unbiased k-means consensus clustering to derive the rationale for the four subtypes analyzed throughout the manuscript. Please see the Authors’ reply to Reviewer #1 comment #8 for detailed explanation. We have included a new Supplemental Fig. 2a-c showing the K-means consensus clustering of the TCGA tumors and the subtype correlations. Using the cluster identities, we analyzed histological classification and did not observe any meaningful differences (Table 1 below). However, the addition of the analysis along with the scRNA analysis requested by reviewer #3 have strengthened the rationale for using all of the subtype correlations in a regression model to identify differentially expressed genes/proteins rather than a binary approach of binning the tumor into the subtype with the highest correlation, as it is now clear that bulk tumor can be composed of one or more subtypes on an individual cell analysis.

Table 1. Histological classification by unbiased consensus cluster subtypes

Consensus cluster	TNBC subtype composition	Invasive ductal	Invasive lobular	Medullary carcinoma	Metaplastic carcinoma	other
1	M	32	0	0	4	0
2	M/BL1	37	2	0	0	2
3	BL1/IM	45	0	3	0	2
4	BL2	23	0	0	0	0
5	LAR	24	3	0	3	3

References

1. Bareche, Y. *et al.* Unravelling triple-negative breast cancer molecular heterogeneity using an integrative multiomic analysis. *Ann. Oncol.* **29**, 895–902 (2018).
2. Lehmann, B. D. *et al.* Refinement of Triple-Negative Breast Cancer Molecular Subtypes: Implications for Neoadjuvant Chemotherapy Selection. *PLoS One* **11**, e0157368 (2016).
3. Domcke, S., Sinha, R., Levine, D. A., Sander, C. & Schultz, N. Evaluating cell lines as tumour models by comparison of genomic profiles. *Nat. Commun.* **4**, 2126 (2013).
4. Yen, A. & Kellis, M. Systematic chromatin state comparison of epigenomes associated with diverse properties including sex and tissue type. *Nat. Commun.* **6**, 7973 (2015).
5. Knutson, S. K. *et al.* Selective inhibition of EZH2 by EPZ-6438 leads to potent antitumor activity in EZH2-mutant non-Hodgkin lymphoma. *Mol. Cancer Ther.* **13**, 842–854 (2014).
6. Zhang, H. *et al.* EZH2 targeting reduces medulloblastoma growth through epigenetic reactivation of the BAI1/p53 tumor suppressor pathway. *Oncogene* **39**, 1041–1048 (2020).
7. Italiano, A. *et al.* Tazemetostat, an EZH2 inhibitor, in relapsed or refractory B-cell non-Hodgkin lymphoma and advanced solid tumours: a first-in-human, open-label, phase 1 study. *Lancet Oncol.* **19**, 649–659 (2018).
8. Krug, K. *et al.* Proteogenomic landscape of breast cancer tumorigenesis and targeted therapy. *Cell* **183**, 1436–1456.e31 (2020).
9. Karaayvaz, M. *et al.* Unravelling subclonal heterogeneity and aggressive disease states in TNBC through single-cell RNA-seq. *Nat. Commun.* **9**, 3588 (2018).
10. Li, H. *et al.* Immune regulation by low doses of the DNA methyltransferase inhibitor 5-azacitidine in common human epithelial cancers. *Oncotarget* **5**, 587–598 (2014).
11. Zhou, L., Mudianto, T., Ma, X., Riley, R. & Uppaluri, R. Targeting EZH2 Enhances Antigen Presentation, Antitumor Immunity, and Circumvents Anti-PD-1 Resistance in Head and Neck Cancer. *Clin. Cancer Res.* **26**, 290–300 (2020).
12. Ennishi, D. *et al.* Molecular and genetic characterization of MHC deficiency identifies EZH2 as therapeutic target for enhancing immune recognition. *Cancer Discov.* **9**, 546–563 (2019).
13. Burr, M. L. *et al.* An evolutionarily conserved function of polycomb silences the MHC class I antigen presentation pathway and enables immune evasion in cancer. *Cancer Cell* **36**, 385–401.e8 (2019).

REVIEWER COMMENTS

Reviewer #3, expert in breast cancer subtypes/epigenetics (Remarks to the Author):

I thank the author for all the additional experiments and analysis that they performed. I especially appreciated the usage of scRNAseq data to link bulk classification and tumor heterogeneity. The manuscript is now ready for publication.

Reviewer #4, , expert in pharmacogenomics and in silico vulnerabilities (Remarks to the Author):

Major points

In general the manuscript has improved over all. Still, the amount of detail in the text is still distracting from the over-all message of the manuscript. For instance: is it really necessary to explain all clinical characteristics of the patients, especially when considering the relative high consistency and limited relation to the outcome of the work (especially line 98-151)? Also, is it necessary to repeat all the mutations that associate with the subtypes (line 189-197, 219-231)? Each of these points can be addressed in a single sentence, perhaps referring to a table. Only the mutational burden, mutation signatures, epigenetic and immune checkpoint-mutations would be of importance in the context of this work. Of note: The statement that survival/hazard ratio is correlated to the level of immune infiltration (S1D) and subtype (S1E) is not statistically confirmed and should probably not be claimed as such. Figure 3B, C and D have changed and are still a bit difficult to grasp given the high number of possibly true/false positive as well as negative outcomes. It would be advisable to integrate the data and only show the most consistent patterns. BL1 is obviously cell cycle related, then I would advise only to show some genes/targets that are most consistent between Depmap, GDSC and PDTX related to the cell cycle and continue accordingly with the other subtypes. This will provide confidence in the validity of the approach. It could also help to show the underlying data in full detail in the supplements (i.e. showing values for each individual cell line and target gene in a heatmap). Relations that can cause any doubt should be removed. Since this manuscript deals particularly with the M-type, why are the outcomes particularly important for this subtype? Mention this in the title of this section, for example "Analysis of in silico datasets from genetic and pharmacologic screens identify few targetable vulnerabilities in M-subtype TNBCs"

The therapeutic targets that are identified were previously suggested to be evaluated in relevant in vivo models. The in vivo data indeed show that the combined effect of paclitaxel and tazmetostat is stronger than each drug separate which is a positive outcome. Furthermore, I would suggest to add a few more histology quantifications in addition to the CD3 lymphocyte quantification as shown. Also, I would suggest to state whether this model can be considered an M-type tumor. The Western-blot for MHC proteins can be shown here as well (now only FACS data is provided).

Minor issues

The k-means clustering partially confirms the previous TNBC classifications and the single cell data show that non-consistent clustering might be explained by cell-abundancy differences within tumors. Why not use the term "k-means" rather than "CC" in Figure S2C, and why not putting it separate from the other groups on top since this is the reference for comparison?

For the single cell data (Figure S3) it would be advisable to show the UMAPs using the same scale setting for all patients. A clear conclusion to the section "Unsupervised clustering and single cell ..." should be made, since the current statement "This suggests that tumors with multiple correlations are composed of mixed subtypes" is insufficient precise. Make this explicit, does this address the inconsistencies? A summarizing figure in the main manuscript that integrates the classification based on the current standard, k means and single cell data could be provided, most optimally being integrated as part of Figure 1b.

Some remarks of the previous version should be reconsidered: "However, despite both BL1 and M tumors displaying higher mutational burdens, only BL1 tumors were significantly ($p=0.021$ log-rank

test) associated with better survival, suggesting differences in subtype-specific response to standard chemotherapy (Supplementary Fig. 5c).” I think that the underlying assumptions should be reconsidered, i.e. is survival necessarily only linked to the response to chemotherapy. What is probably mentioned here is that the “cell cycle” gene ontology associates with BL1. Implicating that chemotherapy efficacy is causally connected to this is not substantiated though. Of note, the alignment Figure 3d, lowest block, is shifted

Reviewer #5, expert in proteogenomics and breast cancer

Reviewer 5.

Lehmann and colleagues have collected multiple levels of publicly available omics data and performed a multi-omics analysis to increase our understanding of TNBC. They identify the PRC2 complex as important in the mesenchymal subtype and pharmacological intervention of PRC2 increase MHC levels and together with chemotherapy decrease tumor burden in a mouse model.

Overall the study provides new knowledge and is suitable for publication in Nat Comm. However, there are some areas that needs clarification and adjustment first.

After reading the abstract I find it out of sync with the title. The abstract has one sentence that vaguely describes the multi-omics analysis. The rest focus on the mesenchymal subtype and PRC2. An alternative title based on this could be: “Multi-omics analysis of TNBC identify PRC2 as a potential drug target in the mesenchymal subtype”.

The authors use the term proteogenomics in the title. Proteogenomics relates to how genomics can be used to either support identifications of novel peptides or how the DNA and RNA levels impact the proteome. However, the majority of the work in the paper is genomics centered. The first figure that sets the tone for the paper only includes genomics. Figure 2b and S6C could be considered proteogenomics analyses. But is it enough to qualify for proteogenomics in the title? To clearly consider the work as proteogenomics I would expect analysis in fig 4 to unravel how and if the methylation states influence the protein levels? Also, for the paper to be proteogenomics oriented, I would expect to see analyses on how and if copy numbers alterations, and other mutations reach the proteome in TNBC.

The authors also introduce, what is a new term to me, “proteogenomics distribution”. This was at first confusing since it simply refers to RNA and protein levels. This could be confusing for the reader since to my knowledge, it is not a defined concept.

From results: “Primary TNBCs showed similar distributions as previous 107 studies^{3,6}, while metastatic TNBCs had a greater proportion of BL2 tumors (Table 1).” The percentages can vary 10% for other subtypes as well in the primary tumors, so I find this statement unsupported.

In results: “Similar deconvolution methods were used to determine immune cell composition and supported an absence of antigen presenting and effector immune cell classes in the M-subtype²².”

TNBC tumors displayed distinct patterns of gene expression. Most striking was the absence of immune cell markers, immune checkpoint expression, and antigen presentation expression in the mesenchymal subtype (Fig. 1b). Similar patterns of tumor profiles were observed in primary tumors in METABRIC and metastatic tumors in MET500 (Supplementary Fig. S4a-f).” Which tumor profiles are you refereeing to? The different profile groups in fig 1b are different to the groups in S4F, so it is difficult to compare. You could do boxes for groups of genes for easier comparison. You also have the CPTAC protein and RNA data in S6C, which makes it difficult to judge how reproducible the gene groups are between datasets.

The results that TMB is related to different outcome in different subtypes (figure S5C) is very interesting. I would suggest that you consider to lift it to a main figure. Especially if you can see the same results in the other cohorts (Metabric, Met500) as well?

From results: “While there were relatively few activating MAPK pathway mutations (KRAS G12V, HRAS Q61L, HRAS G13R, BRAF L537S, MAP2K1 E203K, MAPK9 S407*), nearly all (5 of 6) occurred in the BL2-subtype (Fig. 1b).” How do you see in fig 1b which are the activating mutations? Amplifications and deletions are shown. Relating to the proteogenomics comment before, for the paper to be more proteogenomics oriented, the question arise if these mutations actually reach the protein level where they can have an effect?

“NOTCH pathway alterations (NOTCH2 and NOTCH3 amplifications and FBXW7 deletions) occurred frequently in BL1- (33%) and M- subtypes (46%) and were mutually exclusive with mutations in the pathway.” What do you mean by mutually exclusive? Both NOTCH2 and NOTCH3 amplifications can occur in the same tumor in fig 1b.

I like the condensed version of the results in of fig 3 in 3e. However I miss the figure text and how it was compiled in materials and methods. And I don’t understand the yellow “genetic alteration” bar when it comes to Protein and phosphoprotein. Show protein levels in each of the subtypes instead?

“These extensive genomic and epigenomic differences suggest that TNBC subtypes could arise from different cells of origin. Supporting this hypothesis is the differential correlation to scRNA signatures derived from normal breast epithelium cells²¹.These data suggest the distinct TNBC subtypes may arise from different cells-of-origin, which likely lead to differential sensitivity to therapeutic agents.”

The data that single cells within a tumor can have different TNBC subtypes (fig S3D) could also indicate a cell plasticity that allows transition between cell states, or part of the tumor evolution. Unless tumors frequently arise from multiple cells. 5 of the 6 tumors have mixed subtypes in fig s3d. It would be interesting to see the proportion between the different single cell TNBC subtype annotations to better assess the effect of the mixing.

Which groups are used for the Kaplan Meier plot in fig S2d? I would like to see the consensus clusters from S2C. The color code is different between c and d, so is it the consensus cluster groups? The mixed group is not defined in S2C.

When describing the differences between the TNBC groups in fig 1B, some differences as the low levels of antigen presentation in mesenchymal tumors are obvious. Other differences are less obvious. A statistical analysis of significant enrichment in one or more groups compared to the others would make the arguments stronger and easier to evaluate.

It is good that the code for data analysis has been uploaded to Github! However, it needs some attention: 1) the code to access the different omics data directs to a dropbox. It should point to a publicly available place so the code can be tested. Now I was unable to test the code. 2) There seems to be missing quite some scripts for a lot of the analysis done, 3) test run all code from the new publicly available place. This way you can also claim the paper as a resource of current publicly available TNBC omics data for other researchers.

A suggestion, which you don't have to add if you don't want to, is a schematic figure showing how PRC2 is molecularly involved in the mesenchymal subtype and how the inhibitors work.

Line 621: "We did not observe substantial cytotoxicity of PRC2 inhibitors in TNBC cells, and therefore PRC2 inhibitors will unlikely be ineffective as a monotherapy in TNBC." Ineffective -> effective?

As a general comment, as a reviewer I highly appreciate when figure, table and reference numbers are correct. There are quite some places where I have to double check and investigate which is the correct number. Please, have someone carefully read the manuscript and check. See below for some examples.

ErBb2 should be ERBB2 on line 538

Line 967 – "In total, we identified 192 (17.5%) TNBC tumors from 1097 patients in TCGA and 28 (23.0%) TNBC from 122 CPTAC BRCA tumors. For METABRIC, mRNA expression distributions for ER, PR, and HER2 with clinical annotations for ER and HER2 were used to infer hormone status (Supplementary Fig. 1a)." Fig S1a only show data for TCGA.

Line 974 – "Using mRNA expression cutoffs for the metastatic MET500 dataset, we identified 66 TNBC samples representing 40 unique patients (Supplementary Fig. S6b)." S4B instead?

I must give credit to all clinical information in the supplementary files. The supplementary tables don't have a number in the review files so I am grateful for the index sheet. However, there are 2 suppl. table nr 3, which made it confusing for me. I was also not able to find TILs in table S2 as pointed to here: "or 4)

immune desert (ID) defined as with TILs absent from the tumor core and surrounding tissue (Supplementary Table 2).”

“To explore this possibility, we evaluated single-cell RNA sequencing 143 (scRNA-seq) from six primary TNBC patients (Supplementary Fig. 3a).15” Ref 15 is: 15. Curtis, C. et al. The genomic and transcriptomic architecture of 2,000 breast tumours 694 reveals novel subgroups. Nature 486, 346–352 (2012). Which is not single cell RNA-seq.

S4f is S4d in the figure text

Fig S5g appear twice in figure text. 2nd S5g should be S5h.

RESPONSE TO REVIEWER COMMENTS

Authors' reply: We appreciate the reviewer's time and thoughtful critique and have provided a point-by-point rebuttal addressing each of the reviewer's concerns below.

REVIEWER 3

Reviewer #3, expert in breast cancer subtypes/epigenetics (Remarks to the Author):

I thank the author for all the additional experiments and analysis that they performed. I especially appreciated the usage of scRNAseq data to link bulk classification and tumor heterogeneity. The manuscript is now ready for publication.

Authors' reply: We appreciate the reviewer's time and suggestions.

REVIEWER 4

Reviewer #4, expert in pharmacogenomics and in silico vulnerabilities (Remarks to the Author):

Major points

1. In general the manuscript has improved over all. Still, the amount of detail in the text is still distracting from the over-all message of the manuscript. For instance: is it really necessary to explain all clinical characteristics of the patients, especially when considering the relative high consistency and limited relation to the outcome of the work (especially line 98-151)? Also, is it necessary to repeat all the mutations that associate with the subtypes (line 189-197, 219-231)? Each of these points can be addressed in a single sentence, perhaps referring to a table. Only the mutational burden, mutation signatures, epigenetic and immune checkpoint-mutations would be of importance in the context of this work.

Authors' reply: We agree with the reviewer that these sections pointed out by the reviewer are not the main focus of the manuscript and could be distracting, therefore we have condensed the clinical section of lines 98-151s from 30 to 20 lines, the mutation analysis in lines 189-197 to 5 lines, and the copy number analysis in lines 219-231 to 7 lines. We kept the "Unsupervised clustering and single cell RNA analysis uncovers intra-tumor heterogeneity" section intact from lines 189-197, as these analyses were specifically requested by other reviewers. We have also streamlined the "In silico analysis of genetic..." section from 76 to 53 lines when revising Fig. 3 as requested by the reviewer in comment #3.

2. Of note: The statement that survival/hazard ratio is correlated to the level of immune infiltration (S1D) and subtype (S1E) is not statistically confirmed and should probably not be claimed as such.

Authors' reply: We agree that the analysis of immune infiltration (S1D) was not statistically significant as presented (tertiles), however there is a clear trend of better survival in the high and medium tertiles. We have performed a comparison between either the upper and lower tertile and see a similar non-significant

trend (below), therefore we have modified the text (Page 3, Lines 53-56), as follows, “TNBCs with lower immune cell estimates¹⁸ trended (log-rank p-value=0.11) towards a shorter progression-free interval (PFI), while tumors with stromal immune cells displayed the lowest risk of recurrence (HR, 0.59) (Supplementary Fig. 1d and e).”

Figure 1. Upper and lower tertile survival analysis by immune estimate

3. Figure 3B, C and D have changed and are still a bit difficult to grasp given the high number of possibly true/false positive as well as negative outcomes. It would be advisable to integrate the data and only show the most consistent patterns. BL1 is obviously cell cycle related, then I would advise only to show some genes/targets that are most consistent between Depmap, GDSC and PDTX related to the cell cycle and continue accordingly with the other subtypes. This will provide confidence in the validity of the approach. It could also help to show the underlying data in full detail in the supplements (i.e. showing values for each individual cell line and target gene in a heatmap).

Authors' reply: We apologize for the changes to figure 3 requested by the prior reviewer that has dropped out. We have taken this reviewer's advice and have modified Fig 3 b, c and d to show some genes/targets integrated by pathways. We have highlighted only those dependencies within cell cycle, DNA repair, AR signaling, PI3K/mTOR, growth factor and adhesion/developmental pathways. As such we have moved the old figure 3 to supplemental Fig. 7.

Figure 2 (Figure 3 in manuscript). *In silico* analyses of datasets from genetic and pharmacologic screens identifies subtype-specific vulnerabilities in TNBC

4. Relations that can cause any doubt should be removed. Since this manuscript deals particularly with the M-type, why are the outcomes particularly important for this subtype? Mention this in the title of this section, for example “Analysis of in silico datasets from genetic and pharmacologic screens identify few targetable vulnerabilities in M-subtype TNBCs”

Authors' reply: We agree with the reviewer and have modified the section title (Page 9, Lines 229-230), to, “In silico analysis of genetic and pharmacologic screens identify few targetable vulnerabilities in M-subtype TNBCs.”

5. The therapeutic targets that are identified were previously suggested to be evaluated in relevant in vivo models. The in vivo data indeed show that the combined effect of paclitaxel and tazmetostat is stronger than each drug separate which is a positive outcome. Furthermore, I would suggest to add a few more histology quantifications in addition to the CD3 lymphocyte quantification as shown. Also, I would suggest to state whether this model can be considered an M-type tumor. The Western-blot for MHC proteins can be shown here as well (now only FACS data is provided).

Authors' reply: We appreciate the reviewer's comment and have added additional IHC staining for H3K27me3, Ki-67 and cleaved caspase-3 (now Figure S12a-d). While we did attempt to quantify each stain individually, it is clear that staining can vary substantially within each tumor. Therefore, we have provided overall tumor images to show gross differences in staining. Tumors treated with vehicle alone are

much larger with proliferating (Ki-67+) cells along the periphery of the tumor and cleaved caspase-3 positive cells confined to the necrotic core (Fig. S12a). H3K27me3+ expressing cells were confined to proliferating cells and staining less intense in tazemetostat-treated tumors. Tumors treated with taxol (S12b) or tazemetostat (S12c) were smaller, especially those treated in combination (S12d). Cleaved caspase-3 occurred outside of the necrotic core in mice treated with drugs and with circular areas of caspase-3 negativity, potentially related to resistance or vascular differences.

While we only provided FACS data for MHC I expression, we have tried both immunoblot and IHC for MHC class I using two different H-2Kd/H-2Dd antibodies (clone# 28-8-6, biogend and 34-1-2-S, eBioscience) and were not able to detect specific staining using a TMA of Balb/c mouse tissues. We only observed staining on monocytic cells on the spleen, lymph node and small bowel. We evaluated prior literature for MHC I staining in mice and nearly all used flow cytometry, thus the epitope may be lost by formalin fixation and cell lysis.

We have analyzed datasets (GSE69006 and GSE104765) and found that individual 4T1 tumors correlated to both the M and BL2 subtypes. Further analysis of individual clones of 4T1 (GSE63180) also were either M or BL2, demonstrating that this model is a mixture between both subtypes.

Fig. S12a

Fig. S12c

Fig. S12b

Fig. S12d

Figure 3 (Supplemental Figure 12a-d in manuscript). Immunohistochemistry of H3K27me3, Ki67 and caspase-3 in xenograft tumors from mice treated with **a**, vehicle **b**, taxol **c**, tazemetstat and **d**, tazemetostat + taxol. Images from serial sections of individual 4T1 tumors (rows) stained for H&E or IHC for H3K27me3, Ki-67 and cleaved caspase-3.

Minor issues

6. The k-means clustering partially confirms the previous TNBC classifications and the single cell data show that non-consistent clustering might be explained by cell-abundancy differences within tumors. Why not use the term “k-means” rather than “CC” in Figure S2C, and why not putting it separate from the other groups on top since this is the reference for comparison?

Authors' reply: We thank the reviewer for this suggestion and have modified Figure S2C as recommended by the reviewer by moving the k-means clustering to the top and changing CC to K-means.

Figure 4. (Figure S2 in manuscript). Unbiased k-means consensus clustering identifies 5 subtypes

7. For the single cell data (Figure S3) it would be advisable to show the UMAPs using the same scale setting for all patients. A clear conclusion to the section “Unsupervised clustering and single cell ...” should be made, since the current statement “This suggests that tumors with multiple correlations are composed of mixed subtypes” is insufficient precise. Make this explicit, does this address the inconsistencies? A summarizing figure in the main manuscript that integrates the classification based on the current standard, k means and single cell data could be provided, most optimally being integrated as part of Figure 1b.

Authors' reply: We have modified the single-cell UMAP plots to have the same scale for all patients. We have modified the conclusion (Page 4, Lines 81-83), “This suggests that tumors with multiple correlations are composed of mixed subtypes” to “These data provide evidence that tumors with multiple correlations are composed of mixed subtypes and may reflect tumor cell plasticity that allows transition between cell states.” While a summarizing figure incorporating all data would be useful, the scRNA data is from a limited number of patients and not from the same tumors as the TCGA and could be confusing. However, we have added the K-means classification to Figure 1B demonstrating mixing of k-means for tumors with decreased subtype correlation.

Figure 5. (Figure S3 in manuscript). Analysis of single-cell RNA-seq (scRNA) reveals intra-tumor TNBC subtype heterogeneity.

Figure 6. (Figure 1b in manuscript). Identification of new TNBC subtype features through integrative genomic analyses

8. Some remarks of the previous version should be reconsidered: “However, despite both BL1 and M tumors displaying higher mutational burdens, only BL1 tumors were significantly ($p=0.021$ log-rank test) associated with better survival, suggesting differences in subtype-specific response to standard chemotherapy (Supplementary Fig. 5c).” I think that the underlying assumptions should be reconsidered, i.e. is survival necessarily only linked to the response to chemotherapy. What is probably mentioned here is that the “cell cycle” gene ontology associates with BL1. Implicating that chemotherapy efficacy is causally connected to this is not substantiated though.

Authors' reply: The reviewer is correct that survival is not only linked to chemotherapy response, however response to chemotherapy is highly associated with better long-term outcomes in TNBC (PMID:18250347). We have therefore modified the sentence as follows, “However, despite both BL1 and M tumors displaying higher mutational burdens, only BL1 tumors were significantly ($p=0.021$ log-rank test) associated with better survival, suggesting differences in subtype-specific long-term outcomes following standard chemotherapy (Supplementary Fig. 5c).”

9. Of note, the alignment Figure 3d, lowest block, is shifted

Authors' reply: We appreciate the reviewer identifying this alignment issue and have made corrections accordingly in the new figure.

REVIEWER 5

Comments for response to Reviewer #1 requests (responses to original referee in red and responses to new referee in blue)

REVIEWER COMMENTS

Reviewer #1, expert in proteogenomics and breast cancer (Remarks to the Author):

Lehmann et al present a potentially interesting manuscript to characterize triple negative breast cancer (TNBC) using publicly available omics data. They go on and validate that PRC2 pharmacological inhibition can partially restore MHC-1 levels in subtype M cell lines. The interesting over-all question is if that is enough to elicit an immune response? However, the reviewing of the content as well as interpretation and conclusions of the work is limited by some shortcomings presented below:

1. There is lots of polishing to do with the manuscript and this partly makes the reviewing the scientific content challenging: In addition, the blurry quality of supplementary figures S1, S4, S5 makes it impossible to accurately review these results.

Authors' reply: We have performed significant polishing of the manuscript including independent editing by the scientific writing and editing for researchers at Vanderbilt (SWERV) core and hope these changes enhance the review of the manuscript. We also apologize for the poor quality of the supplemental figures during pdf conversion in the initial submission and have taken steps to ensure proper image resolution of all supplemental figures.

Response authors' reply (RAR): You should ask for a discount from SWERV since there are still figure numbers pointing to the wrong location and figure text missing as in 3e.

Authors' reply: We have corrected the incorrect figure references (as detailed by reviewer #5 comment 16). We have since remove Figure 3e and the missing text is no longer necessary.

2. Since the majority of the paper revolves around data analysis of publicly available data it is important that the data analysis is clearly explained. The materials and methods section is lacking in many parts. To be transparent and provide a means for other researchers to reproduce the results, all scripts for the analyses should be included with the paper (or preferentially upload to github or similar). Since the authors use a lot of different data sources, the data that goes into the scripts should also be included so the analysis can be easily reproduced (or with alternative methods to load the data from within the scripts).

Authors' reply: We appreciate the reviewer's concern regarding the analysis and therefore we have made all of the data, data acquisition and analysis scripts available on github (https://github.com/TransBioInfoLab/TNBC_analysis) so that analysis can be easily reproduced. We have also modified the methods section to include more detail about identification of TNBC specimens in "Genomic-guided identification of TNBC specimens" (Page 37, Lines 964-967, 972- 974 and 979-984), "TNBC subtype association testing for omics data" (Page 41, Lines 1055- 1056), "RNA expression data analysis (Page 38, Lines 993-1004), "Copy number variant calling" (Page 43, Lines 115-1128) and "Single sample gene expression pathway analysis" (Page 42,Lines 1076-1077).

RAR: Good to see the scripts at Github. The amount of code looks a bite poor. Confirm that all code is there. Also, I could not test any code since the code to access the data were pointing to a dropbox.

Authors' reply: We apologize for the reviewer's inability to test the code due to the scripts reading in from a Dropbox location. We have modified the scripts to read directly from the Github (https://github.com/TransBioInfoLab/TNBC_analysis) data location and we have added all the missing code to reproduce the figures. We also have tested the code on both a mac and pc to ensure compatibility.

Example of above-mentioned problem: The authors present fig 1 as an overview of the TCGA data with RNA, RPPA and copy number alterations. It is unclear in the materials and methods how the different genes were selected for this heatmap. What is the FDR cutoff for inclusion? Other exclusion criteria? Same question goes for fig 2. Some genes that are mentioned in the text reads as they have different levels between subtypes, but do not look significant upon observing the quantitative pattern. Which genes are significantly up in which group? A boxplots of interesting genes is needed to visualize these.

Authors' reply: The reviewer expressed concern over the criteria for the data displayed in Figure 1. The reviewer is correct that we included some non-significant genes in the pathway approach for the RPPA analysis in Figure 1. This was due to the limited number of proteins evaluated with this technology. However, we have taken the reviewer's suggestion and removed the RPPA data from Figure 1, as it does not significantly add to the overall conclusions of the manuscript. The genes selected for the gene expression heatmap were selected based on a biased curated list of known genes in antigen presentation, immune markers and immune checkpoint genes that were significantly (FDR p-value <1E-5) differentially expressed in the M subtype compared to other subtypes. Mutations were hand selected and grouped into similar pathways. Copy number amplifications and deletions were indicated when segment values were > 1 (amplification) or < - 0.7 (deletion) and included for known oncogene and tumor suppressor genes. In Fig. 2 we selected genes with > 1FC and p-value<0.05 to performed unbiased gene ontology analysis and have now indicated this in the figure legend and methods. For Fig 2b-e, we performed a pathway analysis of several pathways and known activating phosphosites.

RAR: This is a biased selection towards the mesenchymal group. This needs to be clearly noted in the text and added to materials and methods. Now I thought fig 1b gave an unbiased overview of TNBC. For this type of exploratory multi-omics analysis, I would expect the first figure to show some unbiased data of TNBC, and then go into directed analyses. Based on reading the text you get the impression that the mesenchymal group appeared based on the “unbiased” discovery in 1b. It would be more informative to first give an unbiased view and then show what specifies the mesenchymal group.

Authors' reply: This first figure is already biased as it has binned TNBC by subtypes. The mesenchymal group has been previously discovered by us and described by several others through unbiased means in independent datasets (PMID: 21633166, PMID: 25208879, PMID: 30853353). The mesenchymal subtype is enriched in mammary stem cell pathways and EMT markers. We feel the inclusion of mesenchymal markers into an already busy figure may distract from the novel finding that focusing on the absence of immune cells and antigen presentation within this subtype.

8. There is an ongoing discussion about the number of subtypes in TNBC. Why did the authors settle with 4 subtypes? The cancer literature suggests that the stroma surrounding the tumor is important for tumor cell response to therapy. Unbiased subtype grouping (for example by consensus clustering) based on the TCGA and other data sets would provide a foundation for the validity of the number of different subtypes in TNBC and the continued characterization of the subtypes in the paper.

Authors' reply: We agree with the reviewer that several other investigators have performed independent analyses and identified 4-6 TNBC subtypes. We chose to focus on the four tumor intrinsic subtypes as the IM and MSL subtypes are likely tumors with high levels of immune and stroma. However, we have performed unbiased consensus clustering on the TCGA RNA-seq data and used the area under the CDF curve to determine that five clusters were the most optimal (Fig.1 below and Supplemental Fig. 2 in manuscript). Annotation of the clusters with both 4 and 6 TNBC subtypes along with correlation strength showed that these five clusters were composed of M-subtype (cluster 1) a mixture of BL1 and M (cluster 2), BL1 subtype (cluster 3), BL2 subtype (cluster 4) and a LAR subtype (cluster 5). Interestingly most of the IM subtype tumors were within the BL1 subtype, however they were also present in BL2 and LAR tumors with lower subtype correlations. These data support that the IM subtype is not a distinct subtype, but rather reflects tumors of varying subtypes that include tumor infiltrating immune cells. While not tumor intrinsic, this classification likely identifies immune reactive tumors that have better prognosis and may be more amenable to immune checkpoint therapy, regardless of subtype. Cluster 3 that is composed of both M and BL1 tumors with correlations to both subtypes likely reflects a transition state between BL1 tumors that are undergoing epithelial to mesenchymal transition. These data support the genomic similarities (mutation, copy number) between BL1 and M tumors, but differ in gene expression and global methylation patterns. Together with the scRNA data, these data suggest that binary subtyping may not accurately reflect the true tumor composition of individual cells of multiple subtypes and support the use of continuous modeling of subtypes using the correlation strength of each subtype. We have added additional text (Page 5-6 Lines 132-154).

RAR: I agree that a tumor is rarely of pure subtype but rather a mixture, or part of a continuum between subtypes. The data and reasoning above also suggests that the number of TNBC subtypes are still up for debate. It depends on if you allow the immune cells to drive the subtype cluster formation and if the immune cells themselves impart changes in the tumor cells that will effect therapy outcome. Since tumor infiltration is prognostic it is of relevance. Consider adding some line about this reasoning.

Authors' reply: We appreciate the reviewer's insight and have added the following text (Page 4, Lines 91-93), “Since TNBC tumors are rarely pure, but rather a mixture or part of a continuum, we performed all

differential testing using subtype correlation strength rather than binary subtype assignment.” We agree that the number of subtypes is debatable and likely influenced by the presence of other cell types in the tumor microenvironment. This was clear from the reclassification of our original six subtype to four in which we reclassified the immunomodulatory (IM) and mesenchymal stem-like (MSL) subtypes due to the overwhelming contribution of signal from infiltrating lymphocytes and tumor-associated stromal cells (PMID: 27310713). However, regardless of the true number of subtypes, there are consistently at least four subtypes appearing in several independent analyses (PMID: 21633166, PMID: 25208879, PMID: 2588748 and PMID: 30853353).

13. Genes and drugs are grouped into different pathways in fig 3. What is the overlap in pathway dependencies between the 3 different types of data? Do all different drugs that target the same gene have an effect in a subtype? An overview figure summarizing recurrent dependencies between the 3 data types would be useful. Another suggestion would be to make a figure that relates the RNA/protein levels in TCGA to a dependency to identify potential biomarker candidates for a drug target. Now the results feels confusing and you have to manually look for potential biomarker in for example fig 1.

Authors' reply: We agree with the reviewer that differentially displaying the data between data types could be cumbersome for the reader. Therefore, we have modified Fig. 3 b and c to a heatmap to show differential sensitivity to genetic and pharmacologic dependency between the subtypes. Lower T-values in blue indicate greater differential sensitivity to the genes/agents organized by pathway. The reviewer also recommends adding an overview figure summarizing recurrent dependencies and potential overlap with biomarkers identified in TCGA/CPTAC. Therefore, we have added a figure (Figure 3 below, Fig. 3e in manuscript) summarizing significant genomic alterations (mutation, CN, RNA, protein and phosphoprotein) that are associated with sensitivity to at least two dependency screens (genetic, pharmacologic or PDX screen). This figure shows several potential biomarkers for the dependencies screen in BL1, BL2 and LAR subtypes. However, while we did observe recurrent dependencies on RAC1/CDC42 and RARA in the mesenchymal subtype, there were no genomic alterations identified in this subtype.

RAR: S3e is informative. RNA and Protein among genetic dependencies, what do you mean?

Authors' reply: We apologize for not including the figure legend for 3e and understand how interpretation of this figure could be confusing. RNA and protein (yellow) are not among the genetic dependencies (purple), but rather part of genomic alterations (yellow) that are significantly elevated RNA and protein expression ($p < 0.05$ and $FC > 1$) identified from subtype-specific differential testing in the TCGA and CPTAC analyses. However, we have since revised figure 3 as per reviewer #4 suggestion and removed this figure (See reviewer #4 comment 3)

15. In fig 5, a panel of cell lines are used to compare the protein and RNA expression to M subtype. Which subtypes do the other cell lines reflect? Do they represent all the TNBC subtypes?

Authors' reply: In Figure 5 we included three (HCC1937, HCC1143 and MDA-MB-468) and two BL2 (HCC1806 and MDA-MB-436) cell lines for comparison. Since, we identified and validated many potential therapeutic targets (AR, PI3K, AKT and ERBB2 inhibitors) for the LAR subtype in Fig. 3, we chose to focus on the mesenchymal subtype compared to the other basal subtypes in the remaining analysis.

RAR: The first 3 (HCC1937, HCC1143 and MDA-MB-468) represent what subtype? Add the other cell line annotations to the text so it becomes clearer what you are comparing to.

Authors' reply: We apologize for leaving out that the first three cell lines were BL1 in the rebuttal. We have modified Figure 5a to include subtype annotations.

Figure 7. (Figure 5a in manuscript).

18. I suggest that the authors use at least one person, with expertise in the field that have never read the paper to take substantial time to critically read it, check all statements made in it and provide feedback for updating the manuscript.

Authors' reply: We thank the review for the suggestion and have had members of the scientific writing and editing for researchers at Vanderbilt (SWERV) core critically edit the manuscript.

RAR: Overall the manuscript is easier to read now. There are some numbers pointing to wrong locations and supplementary tables nrs that don't add up. Go over and check once more after all the figure and table updates.

Authors' reply: We have carefully read over the supplementary tables and figures to ensure fidelity when referred to in the manuscript.

19. Proteogenomics is a relatively new term that have become a bite fashionable the last years. I do not fully agree on how the authors use the word proteogenomics. Please see the definition by Nesvizhskii 2014 Nat Methods.

Authors' reply: We agree that one interpretation of “proteogenomics” by Nesvizhskii et. al defines proteogenomic as the use sequencing and transcriptomics (RNA-Seq, ribosome profiling) data to generate customized protein sequence databases to help interpret proteomics (LCMS/MS) data. However, increasingly the term “proteogenomics” is used to describe an approach using the intersection/convergence of proteomics and genomics, such as the recent CPTAC breast manuscript, entitled, “Proteogenomic Landscape of Breast Cancer Tumorigenesis and Targeted Therapy”.

RAR: I agree that this definition also is valid. But the majority of the paper is genomics centered and only figure 2B and S6C directly comparing different levels. For the paper to be considered a proteogenomics paper I would expect further elucidation about how genome, transcriptome, epigenome convergence at the protein level. You have all the levels of data according to fig 1A, but if CNA, mutations, methylation actually reaches the proteome in TNBC is not analyzed. This type of analysis is need in a systematic way for the paper to be considered a proteogenomics paper.

Authors' reply: We agree with the reviewer that paper is not proteogenomic focused and have modified the title to more accurately reflect t the study. The manuscript title to “Multi-omics analysis identifies therapeutic vulnerabilities in triple-negative breast cancer subtypes”.

REVIEWER 5 ADDITIONAL COMMENTS

Lehmann and colleagues have collected multiple levels of publicly available omics data and performed a multi-omics analysis to increase our understanding of TNBC. They identify the PRC2 complex as important in the mesenchymal subtype and pharmacological intervention of PRC2 increase MHC levels and together with chemotherapy decrease tumor burden in a mouse model. Overall the study provides new knowledge and is suitable for publication in Nat Comm. However, there are some areas that needs clarification and adjustment first.

1. After reading the abstract I find it out of sync with the title. The abstract has one sentence that vaguely describes the multi-omics analysis. The rest focus on the mesenchymal subtype and PRC2. An alternative title based on this could be: "Multi-omics analysis of TNBC identifies PRC2 as a potential drug target in the mesenchymal subtype".

Authors' reply: We agree with the reviewer that the abstract focuses on the novel finding of PRC2 in the mesenchymal subtype. However, we feel multi-omic approach validates prior findings in addition to describing a novel target. Therefore, we have modified the abstract and changed the title to "Multi-omics analysis identifies therapeutic vulnerabilities in triple-negative breast cancer subtypes"

2. The authors use the term proteogenomics in the title. Proteogenomics relates to how genomics can be used to either support identifications of novel peptides or how the DNA and RNA levels impact the proteome. However, the majority of the work in the paper is genomics centered. The first figure that sets the tone for the paper only includes genomics. Figure 2b and S6C could be considered proteogenomics analyses. But is it enough to qualify for proteogenomics in the title? To clearly consider the work as proteogenomics I would expect analysis in fig 4 to unravel how and if the methylation states influence the protein levels? Also, for the paper to be proteogenomics oriented, I would expect to see analyses on how and if copy numbers alterations, and other mutations reach the proteome in TNBC.

Authors' reply: We appreciate the reviewer's suggestion of additional analyses that could render the manuscript more proteogenomic. However, we have decided to refine the title and remove the term "proteogenomics", as the majority of the work in the manuscript is genomics centered.

3. The authors also introduce, what is a new term to me, "proteogenomics distribution". This was at first confusing since it simply refers to RNA and protein levels. This could be confusing for the reader since to my knowledge, it is not a defined concept.

Authors' reply: The reviewer is correct that this statement is referring to RNA and protein levels. We have removed the term "proteogenomic distribution".

4. From results: "Primary TNBCs showed similar distributions as previous 107 studies^{3,6}, while metastatic TNBCs had a greater proportion of BL2 tumors (Table 1)." The percentages can vary 10% for other subtypes as well in the primary tumors, so I find this statement unsupported.

Authors' reply: We agree that the distributions can vary by 10%, especially in the smaller CPTAC dataset. Therefore, we have removed the later part of the statement out of caution. The sentence now reads, "Primary TNBCs showed similar distributions as previous studies and distinct subtype-specific patterns (Table 1)."

5. In results: “Similar deconvolution methods were used to determine immune cell composition and supported an absence of antigen presenting and effector immune cell classes in the M-subtype22. TNBC tumors displayed distinct patterns of gene expression. Most striking was the absence of immune cell markers, immune checkpoint expression, and antigen presentation expression in the mesenchymal subtype (Fig. 1b). Similar patterns of tumor profiles were observed in primary tumors in METABRIC and metastatic tumors in MET500 (Supplementary Fig. S4a-f).” Which tumor profiles are you refereeing to? The different profile groups in fig 1b are different to the groups in S4F, so it is difficult to compare. You could do boxes for groups of genes for easier comparison. You also have the CPTAC protein and RNA data in S6C, which makes it difficult to judge how reproducible the gene groups are between datasets.

Authors' reply: We agree that the profiles the reviewer is referring to are confusing as written. We have modified the results to refer to the correct supplemental figures (Supplementary Fig. S4e-f). In addition, we have reordered figure S4f and figure S6c to better assist with pathway comparison.

Figure 8. (Figures S4f and S6c in manuscript).

6. The results that TMB is related to different outcome in different subtypes (figure S5C) is very interesting. I would suggest that you consider to lift it to a main figure. Especially if you can see the same results in the other cohorts (Metabric, Met500) as well?

Authors' reply: We agree with the reviewer that the differing outcomes by TMB in different subtypes is quite interesting and we have performed further analysis in Metabric. Since Metabric was only profiled with a gene panel of known cancer genes we examined the mutation-count thresholds in the METABRIC dataset that approximated the 90th percentile. While we observed a similar trend ($p=0.17$) for TMB-high associated with increased overall survival for TNBC, the data was not robust enough to perform subtype-specific analysis. Therefore, we have opted to leave Figure S5C in the supplemental figures, as we were not able to independently validate these data in another cohort.

7. From results: "While there were relatively few activating MAPK pathway mutations (KRAS G12V, HRAS Q61L, HRAS G13R, BRAF L537S, MAP2K1 E203K, MAPK9 S407*), nearly all (5 of 6) occurred in the BL2-subtype (Fig. 1b)." How do you see in fig 1b which are the activating mutations? Amplifications and deletions are shown. Relating to the proteogenomics comment before, for the paper to be more proteogenomics oriented, the question arise if these mutations actually reach the protein level where they can have an effect?

Authors' reply: We agree with the reviewer that the activating mutations are not annotated within Figure 1b. Therefore, we have generated a table of the activating MAPK pathway mutations by subtype in supplemental table 4a and performed a Fisher's exact test for significance. The text now reads, "While there were relatively few activating MAPK pathway mutations, they were significantly enriched (p -value = 0.01075, Fisher's exact test) in the BL2 subtype (Supplemental Table. 4a)". We agree with the reviewer that relating the mutations to protein would be interesting and more proteogenomics focused, however this mutation analysis was done with TCGA and the CPTAC prospective samples are from different sources, while the retrospective CPTAC cohort does not span enough samples.

Supplemental Table 4a. MAPK pathway mutations by subtype in TCGA

TCGA_ID	Subtype	Gene	AA change
TCGA-C8-A131	BL2	KRAS	G12V
TCGA-E2-A150	BL2	BRAF	L537S
TCGA-AR-A5QQ	BL2	HRAS	G13R
TCGA-E2-A159	BL2	MAP2K1	E203K
TCGA-D8-A13	BL2	MAPK13	E780*
TCGA-B6-A400	M	HRAS	Q61L

8. "NOTCH pathway alterations (NOTCH2 and NOTCH3 amplifications and FBXW7 deletions) occurred frequently in BL1- (33%) and M- subtypes (46%) and were mutually exclusive with mutations in the pathway." What do you mean by mutually exclusive? Both NOTCH2 and NOTCH3 amplifications can occur in the same tumor in fig 1b.

Authors' reply: We agree that both the term mutually exclusive was confusing as written, we were referring to NOTCH2/3 amplifications exclusive with FBXW7 deletions. However, this sentence was removed per reviewer 4 (See comment #1) requesting that we decrease the size of that corresponding section as it is not a major point of the manuscript.

9. I like the condensed version of the results in of fig 3 in 3e. However I miss the figure text and how it was compiled in materials and methods. And I don't understand the yellow "genetic alteration" bar when it comes to Protein and phosphoprotein. Show protein levels in each of the subtypes instead?

Authors' reply: We apologize for not adding the figure legend to Figure 3e and the confusion generated for this figure. The subtype-specific genomic alterations are in yellow for mutations, copy number, RNA protein and phosphoprotein levels. The RNA, protein and phosphoprotein are colored when they are significantly ($p < 0.05$ and $FC > 1$) expressed higher within a subtype determined from our differentially testing in TCGA and CPTAC. We have since removed this figure and modified Figure 3 as per Reviewer #4 request (comment# 3).

10. "These extensive genomic and epigenomic differences suggest that TNBC subtypes could arise from different cells of origin. Supporting this hypothesis is the differential correlation to scRNA signatures derived from normal breast epithelium cells²¹. These data suggest the distinct TNBC subtypes may arise from different cells-of-origin, which likely lead to differential sensitivity to therapeutic agents." The data that single cells within a tumor can have different TNBC subtypes (fig S3D) could also indicate a cell plasticity that allows transition between cell states, or part of the tumor evolution. Unless tumors frequently arise from multiple cells. 5 of the 6 tumors have mixed subtypes in fig s3d. It would be interesting to see the proportion between the different single cell TNBC subtype annotations to better assess the effect of the mixing.

Authors' reply: The reviewer brings up a valid point that the subtype diversity may not only reflect differential cell origins, but also plasticity and transition states as part of tumor evolution. Therefore, we have modified the results section (Page 4, Lines 81-83) to read as follows, "These data provide evidence that tumors with multiple correlations are composed of mixed subtypes and may reflect tumor cell plasticity that allows transition between cell states." We have also added the proportion of individual cell subtypes as a barplot on Figure S3d, as requested by the reviewer.

Figure 9. (Figure S3d in manuscript).

11. Which groups are used for the Kaplan Meier plot in fig S2d? I would like to see the consensus clusters from S2C. The color code is different between c and d, so is it the consensus cluster groups? The mixed group is not defined in S2C.

Authors' reply: The groups used in Fig. S2d are subtypes when mixed samples are removed. We defined the "mixed group" as those samples that displayed a low consensus clustering correlation (<0.5) in Fig S2d. We have also modified Fig S2c to include annotation for those tumors with "mixed" subtypes. In addition, we have added the survival analysis for the K-means consensus clustering (now Fig. S2e) using the same color code as in fig S2C.

Figure 10. (Figure S2e in manuscript). Overall survival of TNBC stratified by k-means cluster

12. When describing the differences between the TNBC groups in fig 1B, some differences as the low levels of antigen presentation in mesenchymal tumors are obvious. Other differences are less obvious. A statistical analysis of significant enrichment in one or more groups compared to the others would make the arguments stronger and easier to evaluate.

Authors' reply: We thank the reviewer for their suggestion and have provided statistical tests for categorical variables (Chi-squared/Fisher's exact test). We added an asterisk in Fig. 1B for select significant (p<0.05) differences colored by subtype that are highlighted in the results section.

13. It is good that the code for data analysis has been uploaded to Github! However, it needs some attention: 1) the code to access the different omics data directs to a dropbox. It should point to a publicly available place so the code can be tested. Now I was unable to test the code. 2) There seems to be missing quite some scripts for a lot of the analysis done, 3) test run all code from the new publicly available place. This way you can also claim the paper as a resource of current publicly available TNBC omics data for other researchers.

Authors' reply: We apologize that the reviewer was unable to test the code due to the scripts directing to a Dropbox. We have modified the input directories, added missing scripts and have tested the code from both a PC and mac. We really appreciate the detail as to which the reviewer has evaluated the code and are certain other researchers will appreciate this as a TNBC omics resource.

14. A suggestion, which you don't have to add if you don't want to, is a schematic figure showing how PRC2 is molecularly involved in the mesenchymal subtype and how the inhibitors work.

Authors' reply: We thank the reviewer for their suggestion and have added a schematic to Fig. 7 to demonstrate how PRC2 is involved in the mesenchymal subtype and how the inhibitors change the epigenetic landscape.

Figure 11. (Figure 7h in manuscript).

15. Line 621: “We did not observe substantial cytotoxicity of PRC2 inhibitors in TNBC cells, and therefore PRC2 inhibitors will unlikely be ineffective as a monotherapy in TNBC.” Ineffective -> effective?

Authors' reply: In this sentence we are inferring that PRC2 inhibitors will not be effective as a single agent as TNBC cell lines tolerate the inhibitor well. This is opposed to the sensitivity of non-Hodgkin's Lymphoma and melanomas models that carry activating EZH2 mutations to PRC2 inhibition (PMID:26845405). Therefore, we have evaluated EZH2 in combination with paclitaxel chemotherapy in our in vivo experiments.

16. As a general comment, as a reviewer I highly appreciate when figure, table and reference numbers are correct. There are quite some places where I have to double check and investigate which is the correct number. Please, have someone carefully read the manuscript and check. See below for some examples.

Authors' reply: We appreciate the Reviewer's thorough review and identification of several incorrectly referenced figures. We have modified the following according to the reviewer's suggestions.

a) ErBb2 should be ERBB2 on line 538

Authors' reply: We only found one usage of “ErBb2” on line 385 and assume the reviewer is referring to the following text, “Although not amplified, increased ErbB2 RNA and protein expression in LAR cells may identify tumors with sensitivity ErbB2 inhibition.” If so, ErbB2 is typically used when referring to protein, while ERBB2 is used when referring to the gene encoding the protein. In this case we are referring to protein and inhibition of protein with therapeutic compounds. We are happy to modify it if the reviewer feels ERBB2 is more appropriate.

b) Line 967 – “In total, we identified 192 (17.5%) TNBC tumors from 1097 patients in TCGA and 28 (23.0%) TNBC from 122 CPTAC BRCA tumors. For METABRIC, mRNA expression distributions for ER, PR, and HER2 with clinical annotations for ER and HER2 were used to infer hormone status (Supplementary Fig. 1a).” Fig S1a only show data for TCGA.

Authors' reply: We apologize for this error and have modified the text to refer to the correct FigS4a.

c) Line 974 – “Using mRNA expression cutoffs for the metastatic MET500 dataset, we identified 66 TNBC samples representing 40 unique patients (Supplementary Fig. S6b).” S4B instead?

Authors' reply: The reviewer is correct and we have modified the text to reference the correct supplementary figure.

d) I must give credit to all clinical information in the supplementary files. The supplementary tables don't have a number in the review files so I am grateful for the index sheet. However, there are 2 suppl. table nr 3, which made it confusing for me. I was also not able to find TILs in table S2 as pointed to here: “or 4) immune desert (ID) defined as with TILs absent from the tumor core and surrounding tissue (Supplementary Table 2).”

Authors' reply: We appreciate the credit as there is substantial information in these files. We apologize for the two supplemental table 3 files. We have corrected the one table that should have been S4. We also apologize for the difficulty in finding the tumor lymphocyte classification in Supplementary Table 2. We had abbreviated the tumor immune microenvironment classification as “TIME” in the spreadsheet. We have now updated the index sheet for Supplemental Table 2 to include the abbreviation for this scoring.

e) “To explore this possibility, we evaluated single-cell RNA sequencing 143 (scRNA-seq) from six primary TNBC patients (Supplementary Fig. 3a).15” Ref 15 is: 15. Curtis, C. et al. The genomic and transcriptomic architecture of 2,000 breast tumours 694 reveals novel subgroups. Nature 486, 346–352 (2012). Which is not single cell RNA-seq.

Authors' reply: The reviewer is correct and we have corrected the reference to the correct citation (Karaayvaz, M et. al).

f) S4f is S4d in the figure text. Fig S5g appear twice in figure text. 2nd S5g should be S5h.

Authors' reply: We have corrected Figure Legend S4 and S5 to reference the appropriate subpanel.

REVIEWERS' COMMENTS

Reviewer #4 (Remarks to the Author):

NCOMMS-20-36469A Reply to authors reply (third round)

Reviewer #4, expert in pharmacogenomics and in silico vulnerabilities (Remarks to the Author):

Major points

1. In general the manuscript has improved over all. Still, the amount of detail in the text is still distracting from the over-all message of the manuscript. For instance: is it really necessary to explain all clinical characteristics of the patients, especially when considering the relative high consistency and limited relation to the outcome of the work (especially line 98-151)? Also, is it necessary to repeat all the mutations that associate with the subtypes (line 189-197, 219-231)? Each of these points can be addressed in a single sentence, perhaps referring to a table. Only the mutational burden, mutation signatures, epigenetic and immune checkpoint-mutations would be of importance in the context of this work.

Authors' reply: We agree with the reviewer that these sections pointed out by the reviewer are not the main focus of the manuscript and could be distracting, therefore we have condensed the clinical section of lines 98-151s from 30 to 20 lines, the mutation analysis in lines 189-197 to 5 lines, and the copy number analysis in lines 219-231 to 7 lines. We kept the "Unsupervised clustering and single cell RNA analysis uncovers intra-tumor heterogeneity" section intact from lines 189-197, as these analyses were specifically requested by other reviewers. We have also streamlined the "In silico analysis of genetic..." section from 76 to 53 lines when revising Fig. 3 as requested by the reviewer in comment #3.

Reviewers reply: The manuscript has now been greatly improved and this point is sufficiently addressed. In general I would reconsider to make the publication title more specific since it now very general.

2. Of note: The statement that survival/hazard ratio is correlated to the level of immune infiltration (S1D) and subtype (S1E) is not statistically confirmed and should probably not be claimed as such.

Authors' reply: We agree that the analysis of immune infiltration (S1D) was not statistically significant as presented (tertiles), however there is a clear trend of better survival in the high and medium tertiles. We have performed a comparison between either the upper and lower tertile and see a similar non-significant 2 trend (below), therefore we have modified the text (Page 3, Lines 53-56), as follows, "TNBCs with lower immune cell estimates trended (log-rank p-value=0.11) towards a shorter progression-free interval (PFI), while tumors with stromal immune cells displayed the lowest risk of recurrence (HR, 0.59) (Supplementary Fig. 1d and e)."

Reviewers reply: This point is now sufficiently addressed.

3. Figure 3B, C and D have changed and are still a bit difficult to grasp given the high number of possibly true/false positive as well as negative outcomes. It would be advisable to integrate the data and only show the most consistent patterns. BL1 is obviously cell cycle related, then I would advise only to show some genes/targets that are most consistent between Depmap, GDSC and PDTX related to the cell cycle and continue accordingly with the other subtypes. This will provide confidence in the validity of the approach. It could also help to show the underlying data in full detail in the supplements (i.e. showing values for each individual cell line and target gene in a heatmap).

Authors' reply: We apologize for the changes to figure 3 requested by the prior reviewer that has dropped out. We have taken this reviewer's advice and have modified Fig 3 b, c and d to show some genes/targets integrated by pathways. We have highlighted only those dependencies within cell cycle, DNA repair, AR signaling, PI3K/mTOR, growth factor and adhesion/developmental pathways. As such we have moved the old figure 3 to supplemental Fig. 7.

Reviewers reply: This point is now sufficiently addressed.

4. Relations that can cause any doubt should be removed. Since this manuscript deals particularly with the M-type, why are the outcomes particularly important for this subtype? Mention this in the title of this section, for example "Analysis of in silico datasets from genetic and pharmacologic screens identify few targetable vulnerabilities in M-subtype TNBCs"

Authors' reply: We agree with the reviewer and have modified the section title (Page 9, Lines 229-230), to, "In silico analysis of genetic and pharmacologic screens identify few targetable vulnerabilities in M-subtype TNBCs."

Reviewers reply: This point is now sufficiently addressed.

5. The therapeutic targets that are identified were previously suggested to be evaluated in relevant in vivo models. The in vivo data indeed show that the combined effect of paclitaxel and tazemetostat is stronger than each drug separate which is a positive outcome. Furthermore, I would suggest to add a few more histology quantifications in addition to the CD3 lymphocyte quantification as shown. Also, I would suggest to state whether this model can be considered an M-type tumor. The Western-blot for MHC proteins can be shown here as well (now only FACS data is provided).

Authors' reply: We appreciate the reviewer's comment and have added additional IHC staining for H3K27me3, Ki-67 and cleaved caspase-3 (now Figure S12a-d). While we did attempt to quantify each stain individually, it is clear that staining can vary substantially within each tumor. Therefore, we have provided overall tumor images to show gross differences in staining. Tumors treated with vehicle alone are much larger with proliferating (Ki-67+) cells along the periphery of the tumor and cleaved caspase-3 positive cells confined to the necrotic core (Fig. S12a). H3K27me3+ expressing cells were confined to proliferating cells and staining less intense in tazemetostat-treated tumors. Tumors treated with taxol (S12b) or tazemetostat (S12c) were smaller, especially those treated in combination (S12d). Cleaved caspase-3 occurred outside of the necrotic core in mice treated with drugs and with circular areas of caspase-3 negativity, potentially related to resistance or vascular differences.

While we only provided FACS data for MHC1 expression, we have tried both immunoblot and IHC for MHC class I using two different H-2Kd/H-2Dd antibodies (clone# 28-8-6, biolegend and 34-1-2-S, eBioscience) and were not able to detect specific staining using a TMA of Balb/c mouse tissues. We only observed staining on monocytic cells on the spleen, lymph node and small bowel. We evaluated prior literature for MHC1 staining in mice and nearly all used flow cytometry, thus the epitope may be lost by formalin fixation and cell lysis.

We have analyzed datasets (GSE69006 and GSE104765) and found that individual 4T1 tumors correlated to both the M and BL2 subtypes. Further analysis of individual clones of 4T1 (GSE63180) also were either M or BL2, demonstrating that this model is a mixture between both subtypes.

Reviewers reply: This point is now sufficiently addressed.

Minor issues

6. The k-means clustering partially confirms the previous TNBC classifications and the single cell data show that non-consistent clustering might be explained by cell-abundancy differences within tumors. Why not use the term "k-means" rather than "CC" in Figure S2C, and why not putting it separate from the other groups on top since this is the reference for comparison?

Authors' reply: We thank the reviewer for this suggestion and have modified Figure S2C as recommended by the reviewer by moving the k-means clustering to the top and changing CC to K-means.

Reviewers reply: This point is now sufficiently addressed.

7. For the single cell data (Figure S3) it would be advisable to show the UMAPs using the same scale setting for all patients. A clear conclusion to the section "Unsupervised clustering and single cell ..." should be made, since the current statement "This suggests that tumors with multiple correlations are composed of mixed subtypes" is insufficient precise. Make this explicit, does this address the inconsistencies? A summarizing figure in the main manuscript that integrates the classification based on the current standard, k means and single cell data could be provided, most optimally being

integrated as part of Figure 1b.

Authors' reply: We have modified the single-cell UMAP plots to have the same scale for all patients. We have modified the conclusion (Page 4, Lines 81-83), "This suggests that tumors with multiple correlations are composed of mixed subtypes" to "These data provide evidence that tumors with multiple correlations are composed of mixed subtypes and may reflect tumor cell plasticity that allows transition between cell states." While a summarizing figure incorporating all data would be useful, the scRNA data is from a limited number of patients and not from the same tumors as the TCGA and could be confusing. However, we have added the K-means classification to Figure 1B demonstrating mixing of k-means for tumors with decreased subtype correlation.

Reviewers reply: This point is now sufficiently addressed.

8. Some remarks of the previous version should be reconsidered: "However, despite both BL1 and M tumors displaying higher mutational burdens, only BL1 tumors were significantly ($p=0.021$ log-rank test) associated with better survival, suggesting differences in subtype-specific response to standard chemotherapy (Supplementary Fig. 5c)." I think that the underlying assumptions should be reconsidered, i.e. is survival necessarily only linked to the response to chemotherapy. What is probably mentioned here is that the "cell cycle" gene ontology associates with BL1. Implicating that chemotherapy efficacy is causally connected to this is not substantiated though.

Authors' reply: The reviewer is correct that survival is not only linked to chemotherapy response, however response to chemotherapy is highly associated with better long-term outcomes in TNBC (PMID:18250347). We have therefore modified the sentence as follows, "However, despite both BL1 and M tumors displaying higher mutational burdens, only BL1 tumors were significantly ($p=0.021$ log-rank test) associated with better survival, suggesting differences in subtype-specific long-term outcomes following standard chemotherapy (Supplementary Fig. 5c)."

Reviewers reply: This point is now sufficiently addressed.

9. Of note, the alignment Figure 3d, lowest block, is shifted

Authors' reply: We appreciate the reviewer identifying this alignment issue and have made corrections accordingly in the new figure.

Reviewers reply: This point is now sufficiently addressed.

REVIEWER 5

The authors have now addressed the majority of my concerns and with some minor changes/clarifications noted below, the paper is ready for publication.

Comments for response to Reviewer #1 requests (responses to original referee in red and responses to new referee in blue)

REVIEWER COMMENTS

Reviewer #1, expert in proteogenomics and breast cancer (Remarks to the Author):

Lehmann et al present a potentially interesting manuscript to characterize triple negative breast cancer (TNBC) using publicly available omics data. They go on and validate that PRC2 pharmacological inhibition

can partially restore MHC-1 levels in subtype M cell lines. The interesting over-all question is if that is enough to elicit an immune response? However, the reviewing of the content as well as interpretation and conclusions of the work is limited by some shortcomings presented below:

1. There is lots of polishing to do with the manuscript and this partly makes the reviewing the scientific content challenging: In addition, the blurry quality of supplementary figures S1, S4, S5 makes it impossible to accurately review these results.

Authors' reply: We have performed significant polishing of the manuscript including independent editing by the scientific writing and editing for researchers at Vanderbilt (SWERV) core and hope these changes enhance the review of the manuscript. We also apologize for the poor quality of the supplemental figures during pdf conversion in the initial submission and have taken steps to ensure proper image resolution of all supplemental figures.

Response authors' reply (RAR): You should ask for a discount from SWERV since there are still figure numbers pointing to the wrong location and figure text missing as in 3e.

Authors' reply: We have corrected the incorrect figure references (as detailed by reviewer #5 comment 16). We have since remove Figure 3e and the missing text is no longer necessary.

Response authors' reply 2 (RAR2): ok

2. Since the majority of the paper revolves around data analysis of publicly available data it is important that the data analysis is clearly explained. The materials and methods section is lacking in many parts. To be transparent and provide a means for other researchers to reproduce the results, all scripts for the analyses should be included with the paper (or preferentially upload to github or similar).

Since the authors use a lot of different data sources, the data that goes into the scripts should also be included so the analysis can be easily reproduced (or with alternative methods to load the data from within the scripts).

Authors' reply: We appreciate the reviewer's concern regarding the analysis and therefore we have made all of the data, data acquisition and analysis scripts available on github

(https://github.com/TransBioInfoLab/TNBC_analysis) so that analysis can be easily reproduced. We have also modified the methods section to include more detail about identification of TNBC specimens in "Genomic-guided identification of TNBC specimens" (Page 37, Lines 964-967, 972- 974 and 979-984), "TNBC subtype association testing for omics data" (Page 41, Lines 1055- 1056), "RNA expression data analysis (Page 38, Lines 993-1004), "Copy number variant calling" (Page 43, Lines 115-1128) and "Single sample gene expression pathway analysis" (Page 42, Lines 1076-1077).

RAR: Good to see the scripts at Github. The amount of code looks a bite poor. Confirm that all code is there. Also, I could not test any code since the code to access the data were pointing to a dropbox.

Authors' reply: We apologize for the reviewer's inability to test the code due to the scripts reading in from a Dropbox location. We have modified the scripts to read directly from the Github (https://github.com/TransBioInfoLab/TNBC_analysis) data location and we have added all the missing code to reproduce the figures. We also have tested the code on both a mac and pc to ensure compatibility.

RAR2: Good!

Example of above-mentioned problem: The authors present fig 1 as an overview of the TCGA data with RNA, RPPA and copy number alterations. It is unclear in the materials and methods how the different genes were selected for this heatmap. What is the FDR cutoff for inclusion? Other exclusion criteria? Same question goes for fig 2. Some genes that are mentioned in the text reads as they have different levels between subtypes, but do not look significant upon observing the quantitative pattern. Which genes are significantly up in which group? A boxplots of interesting genes is needed to visualize these.

Authors' reply: The reviewer expressed concern over the criteria for the data displayed in Figure 1. The reviewer is correct that we included some non-significant genes in the pathway approach for the RPPA analysis in Figure 1. This was due to the limited number of proteins evaluated with this technology. However, we have taken the reviewer's suggestion and removed the RPPA data from Figure 1, as it does not significantly add to the overall conclusions of the manuscript. The genes selected for the gene expression heatmap were selected based on a biased curated list of known genes in antigen presentation, immune markers and immune checkpoint genes that were significantly (FDR p-value <1E-5) differentially expressed in the M subtype compared to other subtypes. Mutations were hand selected and grouped into similar pathways. Copy number amplifications and deletions were indicated when segment values were > 1 (amplification) or < - 0.7 (deletion) and included for known oncogene and tumor suppressor genes. In Fig. 2 we selected genes with > 1FC and p-value<0.05 to performed unbiased gene ontology analysis and have now indicated this in the figure legend and methods. For Fig 2b-e, we performed a pathway analysis of several pathways and known activating phosphosites.

RAR: This is a biased selection towards the mesenchymal group. This needs to be clearly noted in the text and added to materials and methods. Now I thought fig 1b gave an unbiased overview of TNBC. For this type of exploratory multi-omics analysis, I would expect the first figure to show some unbiased data of

TNBC, and then go into directed analyses. Based on reading the text you get the impression that the mesenchymal group appeared based on the “unbiased” discovery in 1b. It would be more informative to first give an unbiased view and then show what specify the mesenchymal group.

Authors' reply: This first figure is already biased as it has binned TNBC by subtypes. The mesenchymal group has been previously discovered by us and described by several others through unbiased means in independent datasets (PMID: 21633166, PMID: 25208879, PMID: 30853353). The mesenchymal subtype is enriched in enriched in mammary stem cell pathways and EMT markers. We feel the inclusion of mesenchymal markers into an already busy figure may distract from the novel finding that focusing on the absence of immune cells and antigen presentation within this subtype.

RAR2: You can have a biased selection of genes. However, when I read the paper, it is not clear how the genes were selected. Please include a sentence to clarify this.

8. There is an ongoing discussion about the number of subtypes in TNBC. Why did the authors settle with 4 subtypes? The cancer literature suggests that the stroma surrounding the tumor is important for tumor cell response to therapy. Unbiased subtype grouping (for example by consensus clustering) based on the TCGA and other data sets would provide a foundation for the validity of the number of different subtypes in TNBC and the continued characterization of the subtypes in the paper.

Authors' reply: We agree with the reviewer that several other investigators have performed independent analyses and identified 4-6 TNBC subtypes. We chose to focus on the four tumor intrinsic subtypes as the IM and MSL subtypes are likely tumors with high levels of immune and stroma. However, we have performed unbiased consensus clustering on the TCGA RNA-seq data and used the area under the CDF curve to determine that five clusters were the most optimal (Fig.1 below and Supplemental Fig. 2 in manuscript). Annotation of the clusters with both 4 and 6 TNBC subtypes along with correlation strength showed that these five clusters were composed of M-subtype (cluster 1) a mixture of BL1 and M (cluster 2), BL1 subtype (cluster 3), BL2 subtype (cluster 4) and a LAR subtype (cluster 5). Interestingly most of the IM subtype tumors were within the BL1 subtype, however they were also present in BL2 and LAR tumors with lower subtype correlations. These data support that the IM subtype is not a distinct subtype, but rather reflects tumors of varying subtypes that include tumor infiltrating immune cells. While not tumor intrinsic, this classification likely identifies immune reactive tumors that have better prognosis and may be more amenable to immune checkpoint therapy, regardless of subtype. Cluster 3 that is composed of both M and BL1 tumors with correlations to both subtypes likely reflects a transition state between BL1 tumors that are undergoing epithelial to mesenchymal transition. These data support the genomic similarities (mutation, copy number) between BL1 and M tumors, but differ in gene expression and global methylation patterns. Together with the scRNA data, these data suggest that binary subtyping may not accurately reflect the true tumor composition of individual cells of multiple subtypes and support the use of continuous modeling of subtypes using the correlation strength of each subtype. We have added additional text (Page 5-6 Lines 132-154).

RAR: I agree that a tumor is rarely of pure subtype but rather a mixture, or part of a continuum between subtypes. The data and reasoning above also suggests that the number of TNBC subtypes are still up for debate. It depends on if you allow the immune cells to drive the subtype cluster formation and if the immune cells themselves impart changes in the tumor cells that will effect therapy outcome. Since tumor infiltration is prognostic it is of relevance. Consider adding some line about this reasoning.

Authors' reply: We appreciate the reviewer's insight and have added the following text (Page 4, Lines 9193), "Since TNBC tumors are rarely pure, but rather a mixture or part of a continuum, we performed all differential testing using subtype correlation strength rather than binary subtype assignment." We agree that the number of subtypes is debatable and likely influenced by the presence of other cell types in the tumor microenvironment. This was clear from the reclassification of our original six subtype to four in which we reclassified the immunomodulatory (IM) and mesenchymal stem-like (MSL) subtypes due to the overwhelming contribution of signal from infiltrating lymphocytes and tumor-associated stromal cells (PMID: 27310713). However, regardless of the true number of subtypes, there are consistently at least four subtypes appearing in several independent analyses (PMID: 21633166, PMID: 25208879, PMID: 2588748 and PMID: 30853353).

RAR2: Ok

13. Genes and drugs are grouped into different pathways in fig 3. What is the overlap in pathway dependencies between the 3 different types of data? Do all different drugs that target the same gene have an effect in a subtype? An overview figure summarizing recurrent dependencies between the 3 data types would be useful. Another suggestion would be to make a figure that relates the RNA/protein levels in TCGA to a dependency to identify potential biomarker candidates for a drug target. Now the results feels confusing and you have to manually look for potential biomarker in for example fig 1.

Authors' reply: We agree with the reviewer that differentially displaying the data between data types could be cumbersome for the reader. Therefore, we have modified Fig. 3 b and c to a heatmap to show differential sensitivity to genetic and pharmacologic dependency between the subtypes. Lower T-values in blue indicate greater differential sensitivity to the genes/agents organized by pathway. The reviewer also recommends adding an overview figure summarizing recurrent dependencies and potential overlap with biomarkers identified in TCGA/CPTAC. Therefore, we have added a figure (Figure 3 below, Fig. 3e in manuscript) summarizing significant genomic alterations (mutation, CN, RNA, protein and phosphoprotein) that are associated with sensitivity to at least two dependency screens (genetic, pharmacologic or PDX screen). This figure shows several potential biomarkers for the dependencies screen in BL1, BL2 and LAR subtypes. However, while we did observe recurrent dependencies on RAC1/CDC42 and RARA in the mesenchymal subtype, there were no genomic alterations identified in this subtype.

RAR: S3e is informative. RNA and Protein among genetic dependencies, what do you mean?

Authors' reply: We apologize for not including the figure legend for 3e and understand how interpretation of this figure could be confusing. RNA and protein (yellow) are not among the genetic dependencies (purple), but rather part of genomic alterations (yellow) that are significantly elevated RNA and protein expression ($p < 0.05$ and $FC > 1$) identified from subtype-specific differential testing in the TCGA and CPTAC analyses. However, we have since revised figure 3 as per reviewer #4 suggestion and removed this figure (See reviewer #4 comment 3)

RAR2: Ok, fig 3 is much easier to follow now.

15. In fig 5, a panel of cell lines are used to compare the protein and RNA expression to M subtype. Which subtypes do the other cell lines reflect? Do they represent all the TNBC subtypes?

Authors' reply: In Figure 5 we included three (HCC1937, HCC1143 and MDA-MB-468) and two BL2 (HCC1806 and MDA-MB-436) cell lines for comparison. Since, we identified and validated many potential therapeutic targets (AR, PI3K, AKT and ERBB2 inhibitors) for the LAR subtype in Fig. 3, we chose to focus on the mesenchymal subtype compared to the other basal subtypes in the remaining analysis.

RAR: The first 3 (HCC1937, HCC1143 and MDA-MB-468) represent what subtype? Add the other cell line annotations to the text so it becomes clearer what you are comparing to.

Authors' reply: We apologize for leaving out that the first three cell lines were BL1 in the rebuttal. We have modified Figure 5a to include subtype annotations.

Figure 7. (Figure 5a in manuscript).

RAR2: ok!

18. I suggest that the authors use at least one person, with expertise in the field that have never read the paper to take substantial time to critically read it, check all statements made in it and provide feedback for updating the manuscript.

Authors' reply: We thank the review for the suggestion and have had members of the scientific writing and editing for researchers at Vanderbilt (SWERV) core critically edit the manuscript.

RAR: Overall the manuscript is easier to read now. There are some numbers pointing to wrong locations and supplementary tables nrs that don't add up. Go over and check once more after all the figure and table updates.

Authors' reply: We have carefully read over the supplementary tables and figures to ensure fidelity when referred to in the manuscript.

RAR2: ok

19. Proteogenomics is a relatively new term that have become a bite fashionable the last years. I do not fully agree on how the authors use the word proteogenomics. Please see the definition by Nesvizhskii 2014 Nat Methods.

Authors' reply: We agree that one interpretation of "proteogenomics" by Nesvizhskii et. al defines proteogenomic as the use sequencing and transcriptomics (RNA-Seq, ribosome profiling) data to generate customized protein sequence databases to help interpret proteomics (LCMS/MS) data.

However, increasingly the term “proteogenomics” is used to describe an approach using the intersection/convergence of proteomics and genomics, such as the recent CPTAC breast manuscript, entitled, “Proteogenomic Landscape of Breast Cancer Tumorigenesis and Targeted Therapy”.

RAR: I agree that this definition also is valid. But the majority of the paper is genomics centered and only figure 2B and S6C directly comparing different levels. For the paper to be considered a proteogenomics paper I would expect further elucidation about how genome, transcriptome, epigenome convergence at the protein level. You have all the levels of data according to fig 1A, but if CNA, mutations, methylation actually reaches the proteome in TNBC is not analyzed. This type of analysis is need in a systematic way for the paper to be considered a proteogenomics paper.

Authors' reply: We agree with the reviewer that paper is not proteogenomic focused and have modified the title to more accurately reflect t the study. The manuscript title to “Multi-omics analysis identifies therapeutic vulnerabilities in triple-negative breast cancer subtypes”.

REVIEWER 5 ADDITIONAL COMMENTS

Lehmann and colleagues have collected multiple levels of publicly available omics data and performed a multi-omics analysis to increase our understanding of TNBC. They identify the PRC2 complex as important in the mesenchymal subtype and pharmacological intervention of PRC2 increase MHC levels and together with chemotherapy decrease tumor burden in a mouse model. Overall the study provides new knowledge and is suitable for publication in Nat Comm. However, there are some areas that needs clarification and adjustment first.

1. After reading the abstract I find it out of sync with the title. The abstract has one sentence that vaguely describes the multi-omics analysis. The rest focus on the mesenchymal subtype and PRC2. An alternative title based on this could be: “Multi-omics analysis of TNBC identifies PRC2 as a potential drug target in the mesenchymal subtype”.

Authors' reply: We agree with the reviewer that the abstract focuses on the novel finding of PRC2 in the mesenchymal subtype. However, we feel multi-omic approach validates prior findings in addition to describing a novel target. Therefore, we have modified the abstract and changed the title to “Multi-omics analysis identifies therapeutic vulnerabilities in triple-negative breast cancer subtypes”

Response to authors' reply (RAR): ok

2. The authors use the term proteogenomics in the title. Proteogenomics relates to how genomics can be used to either support identifications of novel peptides or how the DNA and RNA levels impact the proteome. However, the majority of the work in the paper is genomics centered. The first figure that sets the tone for the paper only includes genomics. Figure 2b and S6C could be considered proteogenomics analyses. But is it enough to qualify for proteogenomics in the title? To clearly consider the work as proteogenomics I would expect analysis in fig 4 to unravel how and if the methylation states influence the protein levels? Also, for the paper to be proteogenomics oriented, I would expect to see analyses on how and if copy numbers alterations, and other mutations reach the proteome in TNBC.

Authors' reply: We appreciate the reviewer's suggestion of additional analyses that could render the manuscript more proteogenomic. However, we have decided to refine the title and remove the term "proteogenomics", as the majority of the work in the manuscript is genomics centered.

RAR: ok, the term proteogenomics is still used in multiple locations throughout the papers. Consider changing them to multi-omics.

3. The authors also introduce, what is a new term to me, "proteogenomics distribution". This was at first confusing since it simply refers to RNA and protein levels. This could be confusing for the reader since to my knowledge, it is not a defined concept.

Authors' reply: The reviewer is correct that this statement is referring to RNA and protein levels. We have removed the term "proteogenomic distribution".

RAR: good, however check fig S1 and text.

4. From results: "Primary TNBCs showed similar distributions as previous 107 studies^{3,6}, while metastatic TNBCs had a greater proportion of BL2 tumors (Table 1)." The percentages can vary 10% for other subtypes as well in the primary tumors, so I find this statement unsupported.

Authors' reply: We agree that the distributions can vary by 10%, especially in the smaller CPTAC dataset. Therefore, we have removed the later part of the statement out of caution. The sentence now reads, "Primary TNBCs showed similar distributions as previous studies and distinct subtype-specific patterns (Table 1)."

RAR: ok

5. In results: "Similar deconvolution methods were used to determine immune cell composition and supported an absence of antigen presenting and effector immune cell classes in the M-subtype²². TNBC tumors displayed distinct patterns of gene expression. Most striking was the absence of immune cell markers, immune checkpoint expression, and antigen presentation expression in the mesenchymal subtype (Fig. 1b). Similar patterns of tumor profiles were observed in primary tumors in METABRIC and metastatic tumors in MET500 (Supplementary Fig. S4a-f)." Which tumor profiles are you referring to? The different profile groups in fig 1b are different to the groups in S4F, so it is difficult to compare. You could do boxes for groups of genes for easier comparison. You also have the CPTAC protein and RNA data in S6C, which makes it difficult to judge how reproducible the gene groups are between datasets.

Authors' reply: We agree that the profiles the reviewer is referring to are confusing as written. We have modified the results to refer to the correct supplemental figures (Supplementary Fig. S4e-f). In addition, we have reordered figure S4f and figure S6c to better assist with pathway comparison.

Figure 8. (Figures S4f and S6c in manuscript).

RAR: ok

6. The results that TMB is related to different outcome in different subtypes (figure S5C) is very interesting. I would suggest that you consider to lift it to a main figure. Especially if you can see the same results in the other cohorts (Metabric, Met500) as well?

Authors' reply: We agree with the reviewer that the differing outcomes by TMB in different subtypes is quite interesting and we have performed further analysis in Metabric. Since Metabric was only profiled with a gene panel of known cancer genes we examined the mutation-count thresholds in the METABRIC dataset that approximated the 90th percentile. While we observed a similar trend ($p=0.17$) for TMB-high associated with increased overall survival for TNBC, the data was not robust enough to perform subtype-

specific analysis. Therefore, we have opted to leave Figure S5C in the supplemental figures, as we were not able to independently validate these data in another cohort.

RAR: ok, will be interesting to see if this finding can be reproduced in other datasets in the future!

7. From results: “While there were relatively few activating MAPK pathway mutations (KRAS G12V, HRAS Q61L, HRAS G13R, BRAF L537S, MAP2K1 E203K, MAPK9 S407*), nearly all (5 of 6) occurred in the BL2- subtype (Fig. 1b).” How do you see in fig 1b which are the activating mutations? Amplifications and deletions are shown. Relating to the proteogenomics comment before, for the paper to be more proteogenomics oriented, the question arise if these mutations actually reach the protein level where they can have an effect?

Authors' reply: We agree with the reviewer that the activating mutations are not annotated within Figure 1b. Therefore, we have generated a table of the activating MAPK pathway mutations by subtype in supplemental table 4a and performed a Fisher's exact test for significance. The text now reads, “While there were relatively few activating MAPK pathway mutations, they were significantly enriched (p-value = 0.01075, Fisher's exact test) in the BL2 subtype (Supplemental Table. 4a)”. We agree with the reviewer that relating the mutations to protein would be interesting and more proteogenomics focused, however this mutation analysis was done with TCGA and the CPTAC prospective samples are from different sources, while the retrospective CPTAC cohort does not span enough samples.

Supplemental Table 4a. MAPK pathway mutations by subtype in TCGA

TCGA_ID	Subtype	Gene	AA change
TCGA-C8-A131	BL2	KRAS	G12V
TCGA-E2-A150	BL2	BRAF	L537S
TCGA-AR-A5QQ	BL2	HRAS	G13R
TCGA-E2-A159	BL2	MAP2K1	E203K
TCGA-D8-A13	BL2	MAPK13	E780*
TCGA-B6-A400	M	HRAS	Q61L

RAR: ok

8. "NOTCH pathway alterations (NOTCH2 and NOTCH3 amplifications and FBXW7 deletions) occurred frequently in BL1- (33%) and M- subtypes (46%) and were mutually exclusive with mutations in the pathway." What do you mean by mutually exclusive? Both NOTCH2 and NOTCH3 amplifications can occur in the same tumor in fig 1b.

Authors' reply: We agree that both the term mutually exclusive was confusing as written, we were referring to NOTCH2/3 amplifications exclusive with FBXW7 deletions. However, this sentence was removed per reviewer 4 (See comment #1) requesting that we decrease the size of that corresponding section as it is not a major point of the manuscript.

RAR: ok

9. I like the condensed version of the results in of fig 3 in 3e. However I miss the figure text and how it was compiled in materials and methods. And I don't understand the yellow "genetic alteration" bar when it comes to Protein and phosphoprotein. Show protein levels in each of the subtypes instead?

Authors' reply: We apologize for not adding the figure legend to Figure 3e and the confusion generated for this figure. The subtype-specific genomic alterations are in yellow for mutations, copy number, RNA protein and phosphoprotein levels. The RNA, protein and phosphoprotein are colored when they are significantly ($p < 0.05$ and $FC > 1$) expressed higher within a subtype determined from our differentially testing in TCGA and CPTAC. We have since removed this figure and modified Figure 3 as per Reviewer #4 request (comment# 3).

RAR: ok

10. "These extensive genomic and epigenomic differences suggest that TNBC subtypes could arise from different cells of origin. Supporting this hypothesis is the differential correlation to scRNA signatures derived from normal breast epithelium cells²¹. These data suggest the distinct TNBC subtypes may arise from different cells-of-origin, which likely lead to differential sensitivity to therapeutic agents." The data that single cells within a tumor can have different TNBC subtypes (fig S3D) could also indicate a cell plasticity that allows transition between cell states, or part of the tumor evolution. Unless tumors frequently arise from multiple cells. 5 of the 6 tumors have mixed subtypes in fig s3d. It would be interesting to see the proportion between the different single cell TNBC subtype annotations to better assess the effect of the mixing.

Authors' reply: The reviewer brings up a valid point that the subtype diversity may not only reflect differential cell origins, but also plasticity and transition states as part of tumor evolution. Therefore, we have modified the results section (Page 4, Lines 81-83) to read as follows, "These data provide evidence that tumors with multiple correlations are composed of mixed subtypes and may reflect tumor cell plasticity that allows transition between cell states." We have also added the proportion of individual cell subtypes as a barplot on Figure S3d, as requested by the reviewer.

Figure 9. (Figure S3d in manuscript).

RAR: ok, good!

In discussion, the summary sentence reads: “These data suggest the distinct TNBC subtypes may arise from different cells-of-origin, which likely lead to differential sensitivity to therapeutic agents.” Consider changing to: “These data suggest the distinct TNBC subtypes may arise from different cells-of-origin or transitions between cell states, which likely lead to differential sensitivity to therapeutic agents.”

11. Which groups are used for the Kaplan Meier plot in fig S2d? I would like to see the consensus clusters from S2C. The color code is different between c and d, so is it the consensus cluster groups? The mixed group is not defined in S2C.

Authors’ reply: The groups used in Fig. S2d are subtypes when mixed samples are removed. We defined the “mixed group” as those samples that displayed a low consensus clustering correlation (<0.5) in Fig S2d. We have also modified Fig S2c to include annotation for those tumors with “mixed” subtypes. In addition, we have added the survival analysis for the K-means consensus clustering (now Fig. S2e) using the same color code as in fig S2C.

Figure 10. (Figure S2e in manuscript). Overall survival of TNBC stratified by k-means cluster

RAR: ok, good! However, the color codes for mixed are different between S2d and S2e, which is a bit confusing.

12. When describing the differences between the TNBC groups in fig 1B, some differences as the low levels of antigen presentation in mesenchymal tumors are obvious. Other differences are less obvious. A statistical analysis of significant enrichment in one or more groups compared to the others would make the arguments stronger and easier to evaluate.

Authors' reply: We thank the reviewer for their suggestion and have provided statistical tests for categorical variables (Chi-squared/Fisher's exact test). We added an asterisk in Fig. 1B for select significant ($p < 0.05$) differences colored by subtype that are highlighted in the results section.

RAR: ok, good!

13. It is good that the code for data analysis has been uploaded to Github! However, it needs some attention: 1) the code to access the different omics data directs to a dropbox. It should point to a publicly available place so the code can be tested. Now I was unable to test the code. 2) There seems to be missing quite some scripts for a lot of the analysis done, 3) test run all code from the new publicly available place. This way you can also claim the paper as a resource of current publicly available TNBC omics data for other researchers.

Authors' reply: We apologize that the reviewer was unable to test the code due to the scripts directing to a Dropbox. We have modified the input directories, added missing scripts and have tested the code from both a PC and mac. We really appreciate the detail as to which the reviewer has evaluated the code and are certain other researchers will appreciate this as a TNBC omics resource.

RAR: ok, good!

14. A suggestion, which you don't have to add if you don't want to, is a schematic figure showing how PRC2 is molecularly involved in the mesenchymal subtype and how the inhibitors work.

Authors' reply: We thank the reviewer for their suggestion and have added a schematic to Fig. 7 to demonstrate how PRC2 is involved in the mesenchymal subtype and how the inhibitors change the epigenetic landscape.

Figure 11. (Figure 7h in manuscript).

RAR: Nice!

15. Line 621: “We did not observe substantial cytotoxicity of PRC2 inhibitors in TNBC cells, and therefore PRC2 inhibitors will unlikely be ineffective as a monotherapy in TNBC.” Ineffective -> effective?

Authors' reply: In this sentence we are inferring that PRC2 inhibitors will not be effective as a single agent as TNBC cell lines tolerate the inhibitor well. This is opposed to the sensitivity of non-Hodgkin's lymphoma and melanomas models that carry activating EZH2 mutations to PRC2 inhibition (PMID:26845405). Therefore, we have evaluated EZH2 in combination with paclitaxel chemotherapy in our in vivo experiments.

RAR: ok, it is the two negating words after each other that makes it difficult to read. Unlikely ineffective... can you phrase it easier way.

16. As a general comment, as a reviewer I highly appreciate when figure, table and reference numbers are correct. There are quite some places where I have to double check and investigate which is the correct number. Please, have someone carefully read the manuscript and check. See below for some examples.

Authors' reply: We appreciate the Reviewer's thorough review and identification of several incorrectly referenced figures. We have modified the following according to the reviewer's suggestions.

a) ErBb2 should be ERBB2 on line 538

Authors' reply: We only found one usage of “ErBb2” on line 385 and assume the reviewer is referring to the following text, “Although not amplified, increased ErbB2 RNA and protein expression in LAR cells may identify tumors with sensitivity ErbB2 inhibition.” If so, ErbB2 is typically used when referring to protein, while ERBB2 is used when referring to the gene encoding the protein. In this case we are referring to protein and inhibition of protein with therapeutic compounds. We are happy to modify it if the reviewer feels ERBB2 is more appropriate.

b) Line 967 – “In total, we identified 192 (17.5%) TNBC tumors from 1097 patients in TCGA and 28 (23.0%) TNBC from 122 CPTAC BRCA tumors. For METABRIC, mRNA expression distributions for

ER, PR, and HER2 with clinical annotations for ER and HER2 were used to infer hormone status (Supplementary Fig. 1a).” Fig S1a only show data for TCGA.

Authors' reply: We apologize for this error and have modified the text to refer to the correct FigS4a.

c) Line 974 – “Using mRNA expression cutoffs for the metastatic MET500 dataset, we identified 66 TNBC samples representing 40 unique patients (Supplementary Fig. S6b).” S4B instead?

Authors' reply: The reviewer is correct and we have modified the text to reference the correct supplementary figure.

d) I must give credit to all clinical information in the supplementary files. The supplementary tables don't have a number in the review files so I am grateful for the index sheet. However, there are 2 suppl. table nr 3, which made it confusing for me. I was also not able to find TILs in table S2 as pointed to here: “or 4) immune desert (ID) defined as with TILs absent from the tumor core and surrounding tissue (Supplementary Table 2).”

Authors' reply: We appreciate the credit as there is substantial information in these files. We apologize for the two supplemental table 3 files. We have corrected the one table that should have been S4. We also apologize for the difficulty in finding the tumor lymphocyte classification in Supplementary Table 2. We had abbreviated the tumor immune microenvironment classification as “TIME” in the spreadsheet. We have now updated the index sheet for Supplemental Table 2 to include the abbreviation for this scoring.

e) “To explore this possibility, we evaluated single-cell RNA sequencing 143 (scRNA-seq) from six primary TNBC patients (Supplementary Fig. 3a).15” Ref 15 is: 15. Curtis, C. et al. The genomic and transcriptomic architecture of 2,000 breast tumours 694 reveals novel subgroups. Nature 486, 346–352 (2012). Which is not single cell RNA-seq.

Authors' reply: The reviewer is correct and we have corrected the reference to the correct citation (Karaayvaz, M et. al).

f) S4f is S4d in the figure text. Fig S5g appear twice in figure text. 2nd S5g should be S5h. Authors'

reply: We have corrected Figure Legend S4 and S5 to reference the appropriate subpanel.

RAR: Ok, good!

17. on line 352: “We identified 1663, 1463 and 2048 differentially regulated transcripts common to all inhibitors in cell lines CAL51, CAL120 and BT549, respectively (Fig. 5f).” common? Do you mean overlapping 2 or more of the cell lines? Further down: “common to all of the cell lines (n=275, union)...” This is not what you mean, right?

Next sentence: “The vast majority of differentially expressed transcripts increased in expression with PRC2 inhibitor treatment (CAL51, 96.6%; CAL120, 91.2%; BT549, 93.8%) and were shared between each

of the cell lines (Supplementary Fig. 9e).” what does the percentages mean? The shared numbers between cell lines do not look that high in S9e